# Towards a mathematics formalisation assistant using large language models

## Abstract

Mathematics formalisation is the task of writing mathematics (i.e., definitions, theorem statements, proofs) in natural language, as found in books and papers, into a formal language that can then be checked for correctness by a program. It is a thriving activity today, however formalisation remains cumbersome. In this paper, we explore the abilities of a large language model (Codex) to help with formalisation in the Lean theorem prover. We find that with careful input-dependent prompt selection and postprocessing, Codex is able to formalise short mathematical statements at undergrad level with nearly 75% accuracy for 120 theorem statements. For proofs quantitative analysis is infeasible and we undertake a detailed case study. We choose a diverse set of 13 theorems at undergrad level with proofs that fit in two-three paragraphs. We show that with a new prompting strategy Codex can formalise these proofs in natural language with at least one out of twelve Codex completion being easy to repair into a complete proof. This is surprising as essentially no aligned data exists for formalised mathematics, particularly for proofs. These results suggest that large language models are a promising avenue towards fully or partially automating formalisation.

## 1 Introduction

Mathematics (definitions, theorems, proofs, remarks) as found in books and papers is written in a semi-formal style combining natural language with formal language in specialized notation. We refer to the language of this style of writing mathematics as *natural language* or NL. *Formalisation of mathematics* consists of writing mathematics in a *formal language* that can then be checked and manipulated by a computer. NL mathematics writing, while being more rigorous than writing in most other domains, falls far short of the standard of detail and rigour required for full formalisation. Formalisation is done with the help of *proof assistants*. A proof assistant consists of a formal language in which mathematical statements can be encoded along with a piece of software that assists in writing and checking proofs in the formal language up to the foundational axioms. See under Prompt in Figure 1 for some examples. Formalisation is an old endeavour that is thriving with several actively developed libraries of formalised mathematics for major proof assistants including Coq, Isabelle, Lean and Mizar. A major use of proof assistants is in software and hardware verification but here we are concerned with their applications in mathematics: checking formalised mathematics automatically results in a much higher degree of confidence in the correctness of proofs. Formalisation promises to open up new possibilities in mathematical exposition, teaching, research and collaboration (Massot, 2021; Buzzard, 2022); in addition, it can facilitate automated proof discovery, e.g. (Lample et al., 2022).

Formalisation of mathematics today poses a barrier to entry because of the need to learn to use proof assistants; it is also notoriously labour-intensive because many details normally taken for granted in the language of mathematics must be supplied when formalising. *Autoformalisation* Wang et al. (2018) is the task of (semi-)automatically turning a piece of mathematics in natural language into a formalised one. An autoformalisation tool that speeds-up formalisation or fully automates it would be of great value by enabling the above advantages of formalisation and opening up new ones Szegedy (2020).

Autoformalisation is challenging. It is a natural language understanding problem for the language of mathematics. While the language of mathematics is stylized compared to natural language in

other domains and deals with relatively narrow subject matter, it retains much of the complexity in addition to presenting new challenges for autoformalisation of its own, including supplying missing details and assumptions that are taken for granted by humans, and semantically mapping concepts in the informal description to those in the formal corpus (Ganesalingam, 2013; Massot, 2021).

Autoformalisation also presents practical challenges in the application of modern deep learning-based methods: the amount of formalised mathematics available is much smaller than code in major programming languages. Furthermore, there is very little aligned data between informal and formal mathematics. Autoformalisation implicitly includes semantic search in the formalised library. Autoformalisation of proofs is much more than independent autoformalisation of each statement in the proof: one needs to maintain context across the proof and find correspondence between NL constructs and tactics in formal proofs.

In this paper we worked with **Lean**, a popular proof assistant with two actively used versions: Lean 3 (de Moura et al., 2015) and Lean4 (de Moura & Ullrich, 2021). The rapidly evolving Lean mathematical library (abbreviated mathlib) is one of the largest libraries of formal mathematics. mathlib is currently 226MB in size. mathlib is monolithic by design, ensuring that formalisations of different parts of mathematics can be combined easily. The resulting standardization of terminology in mathlib and its good coverage make Lean an attractive target for autoformalisation.

To our knowledge, the only form in which aligned data occurs in mathlib is as docstrings for definitions and theorem statements. Furthermore, there is a complete lack of aligned data for proofs: while some examples of natural language proofs together with their corresponding formal proofs occur in the blueprints of some Lean formalisation projects, e.g. the Liquid Tensor Experiment, these are only a handful and highly specialised.

**Our contributions.** In this paper we apply a large language model (specifically, Codex) to the problem of autoformalisation. We focused on two different tasks: (1) translating theorem statements of a form similar to docstrings of mathlib to theorems (in Lean 4), and (2) translating (outlines of) NL proofs to Lean proofs (in Lean 3). The latest version of Lean, Lean 4, is (in addition to an interactive theorem prover) a full-fledged programming language with a fast runtime. This allows a seamless integration of proofs, programs and meta-programs. We use Lean 4 for one set of experiments and Lean 3 for the other because, at the time of writing, mathlib was only partially available in Lean 4 (via a partial binary port). Hence we use Lean 4 where its additional capabilities are important and Lean 3 where these are not used and the larger library is of greater value. More details on Lean are in Appendix A.

*Theorem statement autoformalisation.* For the evaluation dataset, we chose 120 theorem statements at the undergrad level so that the relevant concepts (background theory and definitions) were mostly already in mathlib. Since mathlib is substantial (it has a significant fraction of undergrad mathematics curriculum apart from many advanced results), this is not a restriction. We focused on theorem statements at the undergrad and more advanced level from various areas of mathematics. These statements tend to be more challenging for autoformalisation compared to mathematics competition problems studied in prior work (Wu et al., 2022) as they often assume more in terms of implicit context and draw from a much larger background (Wu et al., 2022).

We experimented with using input-dependent prompting, with mathlib as a database. Specifically, we chose our few-shot prompts to consist of theorem-docstring pairs from mathlib where the docstring is close in a sentence similarity metric to the statement to be formalised. We also experimented with filtering outputs generated at high temperatures by checking validity in Lean 4 and some other post-processing.

Our results showed that *there is a strong effect of both prompt engineering and selection, and even more when used in combination* and that *a reasonably large fraction are elaborated when both prompt engineering and selection is done* (the results improve further when more prompts are used).

In the context of autoformalisation, we are the first to use input-dependent prompting. Our use of elaboration for postprocessing is novel. Both of these are greatly facilitated by the availability of mathlib, and the nature of Lean 4, which gives easy access to its internals and in Lean 4 itself – the latter allowing shared code and avoiding context switching.

*Autoformalisation of proofs.* We chose an evaluation dataset of 13 NL theorems and their proofs. (Due to the lack of data for proofs and need for manual inspection of the outputs, a larger scale

study was infeasible.) The theorem statements are at the undergrad level with a proof fitting in two-three paragraphs. They are diverse across several axes: (1) proof techniques such as proof by contradiction, induction, algebraic manipulations etc., (2) domains such as topology, analysis, group theory, linear algebra, representation theory, and algebraic number theory, (3) difficulty level.

Our chosen proofs are much longer than a typical theorem statement and we didn't observe Codex outputting a completely correct proof. We instead relaxed the requirement of autoformalisation to produce a (faulty) proof that is easy to repair for humans, saving time and effort compared to formalising from scratch. We experimented with several output formal proof formats depending on the level of detail, and with or without NL comments interspersed in the proofs. We designed fixed few-shot prompts for each of these formats. We undertook a detailed manual study of the outputs. We found that *proofs with comments work better*; for about half of the formal proofs Codex output would save significant effort and for the rest it would save some effort. Proofs with comments is a new kind of prompting strategy in line with other recent prompting strategies such as chain-of-thought prompting Wei et al. (2022). Presumably, interleaving of NL comments helps Codex align its output with the NL proof.

All of our datasets were carefully controlled for the possibility of overlap with the training data of Codex as we discuss in more detail later in the paper. We make all of our data available as supplementary material. In summary, our contributions are

- Design of a postprocessing technique for theorem statement autoformalisation resulting in significantly improved performance when combined with prompt engineering.
- First study of proof autoformalisation and design of a prompting technique for proof auto-formalisation. With this technique, Codex is able to produce *useful* partially correct formal proofs in at least one out of thirteen completions.
- A detailed case study of proof autoformalisation which may be useful for future work.

**Organisation.** After discussion of related work in Section 2, we discuss in detail autoformalisation of theorem statements in Section 3 and of proofs in Section 4. We conclude in Section 5.

## 2 BACKGROUND AND RELATED WORK

Natural language understanding has a long history in AI; here we can only briefly touch upon the most relevant subfields of this large field.

**Semantic parsing and program synthesis from natural language specification.** Semantic parsing is the task of translating a natural language utterance into a logical form. Tasks are normally restricted to specific domains and the logical forms come from a domain-specific language ranging from first-order logic to regular expressions, e.g. (Kamath & Das, 2019; Hahn et al., 2022).

**Large Language Models and mathematics.** The advent of transformer-based large language models (LLMs) for natural languages, e.g. (Devlin et al., 2018; Brown et al., 2020), has brought about a sea change in natural language processing. This is largely fueled by the remarkable ability of LLMs to achieve good performance on a diverse set of tasks, ranging from translation to solving math word problems, via few-shot demonstrations in the prompt even though the LLMs are only trained on the language modelling objective. With careful prompt design, these latent abilities can be further teased out, e.g., (Liu et al., 2021; Wei et al., 2022). The input-dependent prompting we use has precedent in prior work, e.g., Jain et al. (2022). Prompt design can be combined with postprocessing to select the best among many answers generated at higher temperatures, e.g., (Jain et al., 2022; Li et al., 2022; Wang et al., 2022) based on their performance on unit tests and other metrics.

Specifically, LLMs applied to code, e.g. (Chen et al., 2021; Fried et al., 2022), have led to new advances in program synthesis from natural language specification. In this paper, we will be using a Codex (Chen et al., 2021) version code-davinci-002. LLMs and related methods have been used for solving mathematical problems (Lewkowycz et al., 2022) with natural language solutions, for proving theorems in natural language (Welleck et al., 2022), and for proof search, e.g. (Lample et al., 2022).

**Autoformalisation.** While the term *autoformalisation* was coined in Wang et al. (2018), the problem itself has a long history; see Wang et al. (2020). Autoformalisation can be thought of as semantic

parsing for the domain of mathematics. Mathematics is a far larger and sophisticated domain than most domains considered in semantic parsing.

Wang et al. (2020) applied deep learning-based methods to autoformalisation by treating it as a language translation problem. They construct datasets for supervised and unsupervised neural machine translation and evaluate the syntactic distance of the output from the gold output by metrics such as BLEU but do not provide data for correctness. The recent work Wu et al. (2022) is closest to ours and stimulated our work. They considered statement autoformalisation in Isabelle/HOL using LLMs. For their quantitative results, their statements were from middle school to undergrad mathematical competitions (Zheng et al., 2022). These problems use only elementary concepts. Their quantitative studies are for fixed few-shot prompts. While a direct comparison with their results is not possible due to the use of different proof assistants and datasets, our method compares favourably with their method (fixed few-shot prompting with greedy decoding) as shown in the next section. Our input-dependent prompting is not applicable on their dataset due to the lack of availability of aligned data at the elementary level of statements in their datasets. Lean Chat is a fixed-prompt autoformalisation tool for Lean 3 based on Codex.

## 3 AUTOFORMALISING THEOREM STATEMENTS

Here we discuss autoformalisation of theorem statements.

### 3.1 EVALUATION DATASETS

We used three test sets with 40 natural language statements each. The natural language statements were of the same form as typical doctrings in mathlib: single sentences often with Lean code fragments (including names and formulas not in LATEX but in unicode) enclosed in backticks. We call such strings **docstring-style** strings.

Our first set consisted of mathematical theorems (some were conjectures as well) in areas well-represented by mathlib, such as undergraduate-level number theory, group theory and topology.

The other two sets were designed to minimize contamination due to similar results being in the training of Codex. Our second set consisted of what we called *silly statements*, such as *every vector space with dimension 2 is finite dimensional*. While being true, these were easy and/or absurdly specific, so unlikely to appear in this precise form anywhere else. We created this set by looking at theorems in mathlib and modifying them.

The third set consisted of *false statements*: these obviously cannot appear in any library. The statements in this set were closely related to those in mathlib or our first dataset: for example, while our first dataset had the statement *every field is a ring* our third dataset had its (false) converse *every ring is a field*.

### 3.2 TECHNIQUES

We used Codex to translate an input text in natural language to code in Lean 4. Codex takes as input a prompt and returns completion(s). We generated a prompt from the input text and post-processed completions as described below. Figure 1 is an example of a prompt, the initial result (with one completion shown) and the result after post-processing. We remark that this example needs prompt engineering, as we see in Section 3.3.

**Prompt engineering.** Given an input text to be translated, we chose example prompts from mathlib whose docstrings are similar to the input text. We used two notions of similarity: proximity in sentence embeddings and keyword matching. This style of prompt design appears in the previous work, e.g., Jain et al. (2022). The docstrings and the corresponding Lean code were extracted from mathlib documentation.

**Post processing.** Lean 4 code is compiled in two phases: a **parser** converts a string into a *Syntax* object, and an **elaborator** converts a *Syntax* object into a type-correct expression. The elaboration step is a much stricter analogue of type-checking in a strongly-typed language. It is roughly a formal analogue of supplying all the implicit details in an NL theorem statement. Lean 4 is unique among

---

**Input text:** "If a vector space has dimension '2' then it is finite dimensional."

---

**Prompt:**

```
/−− If a vector space has a finite basis, then it is finite−dimensional. −/
theorem {K : Type u} {V : Type v} [division_ring K] [add_comm_group V] [module K V] {ι : Type w} [fintype ι] (h : basis
    ι K V) : finite_dimensional K V :=

...

/−− A vector space has dimension at most '1' if and only if there is a single vector of which all vectors are multiples. −/
theorem {K : Type u} {V : Type v} [division_ring K] [add_comm_group V] [module K V] : module.rank K V ≤ 1 ↔ ∃ (v₀ :
    V), ∀ (v : V), ∃ (r : K), r · v₀ = v :=

/−− If a vector space has dimension '2' then it is finite dimensional. −/
theorem
```

---

**Codex Completion:**

```
{K : Type u} {V : Type v} [division_ring K] [add_comm_group V] [module K V] (h : module.rank K V = 2) :
    finite_dimensional K V
```

---

**Post-processed code in Lean 4:**

```
theorem ∀ {K : Type u} {V : Type v} [inst : DivisionRing K] [inst_1 : AddCommGroup V] [inst_2 : Module K V],
  Module.rank K V = 2 → FiniteDimensional K V
```

---

Figure 1: Example of a prompt, the initial result and the result after post-processing. Part of the prompt was elided to save space; full prompt appears in Appendix B.1.

proof assistants in being implemented in Lean 4 and providing an interpreter API, which facilitates our implementation.

We parsed the Codex completions, translated from Lean 3 to Lean 4 and auto-corrected (as described in Section B.2) to obtain Syntax objects corresponding to (syntactically valid) completions. We attempted to elaborate each of these. Thus, restriction to completions which are successfully parsed and elaborated gives a **strong filter**.

### 3.3 RESULTS

We tested the effects of the prompt engineering and post-processing as well as the final quality of translations for the datasets described in Section 3.1.

**Success rates for the Elaborater** We begin with quantitative results showing the utility of both prompt engineering and elaboration filtering for the datasets described in Section 3.1.

We summarize the number of statements that were elaborated for each of the three sets of statements in Table 1. For each set, we considered results with 4 fixed prompts (those used by Lean Chat) and 4 prompts chosen by sentence similarity. For each of these cases we considered answers chosen greedily (i.e., temperature 0 and 1 completion) and those obtained by choosing several completions at temperature $0.8$ with filtering and selection. We made three runs for each configuration, and the result reported is the *median*. We also ran a configuration with the Codex recommended default temperature $0.2$ and with fixed prompts. The results of this are included in parentheses in the entries for the greedy case. As 11 of the theorem statements were present in mathlib we also ran all the configurations excluding these and obtained similar results as above: in particular 23 of the 29 statements were elaborated with prompt engineering and selection.

We see in the next section that elaboration is a good proxy measure for accuracy. Thus, we can justify the claims made in 1.

The example in Figure 1 illustrates the effect of prompt engineering. None of the 15 completions were elaborated in all the three runs with the fixed (Lean Chat) prompts. The completions often used the wrong name from mathlib or assumed a definition was at a different level of abstraction (e.g., modules versus vector spaces) from that of mathlib. We also saw that a larger number of examples did lead to more sentences being elaborated, but the effect was not strong enough to quantify robustly.

| | Theorems | | Silly Statements | | False Statements | |
|---|---|---|---|---|---|---|
| | Fixed | Input-dependent | Fixed | Input-dependent | Fixed | Input-dependent |
| Greedy | 20 (18) | 21 | 19 (21) | 28 | 15 (16) | 23 |
| Filtered | 25 | 29 | 29 | 34 | 24 | 30 |

Table 1: Numbers of elaborated statements; numbers in parenthesis are for temperature $0.2$ (instead of $0$) with one completion

| | false statements | silly statements | theorem statements |
|---|---|---|---|
| **Elaborated** | **32** | **34** | **33** |
| Correct | 21 | 26 | 30 |
| Some correct | 28 | 32 | 30 |
| All wrong | 4 | 2 | 3 |

Table 2: Correctness of elaborated statements

**Correctness of elaboration.** Next, we analysed how often completions that were successfully elaborated were correct. In the case where more than one completion was elaborated, we considered both whether the chosen completion was correct and whether any of the elaborated completions were correct.

For each of the three sets, we considered a configuration with high temperature and prompt engineering – specifically, we considered the configuration with the highest number of elaborated statements, as our goal was to test elaboration as a proxy measure for correctness. We manually checked the correctness of the selected completion for the elaborated completions, as reported in Table 2.

Further, the statements where all completions were wrong involved some concept for which we had very few prompts available, in part due to the incomplete state of the binary port of mathlib, also suggesting that elaboration is a good proxy measure.

## 4    AUTOFORMALISATION OF PROOFS

As discussed in Sec. 1, this task presents new difficulties on top of autoformalisation of theorem statements. Input-dependent prompting, which was an important ingredient in the previous section, is presently infeasible for proofs due the the lack of aligned data for proofs. Elaboration, another important ingredient for theorem statements, is also infeasible for proofs since it is very rare for the language model to output a completely correct proof. Therefore, instead of aiming for completely correct formalised proofs, we aim for *useful* formalised proof outputs: those that can be easily repaired to construct a correct formalised proof, saving time and effort compared to formalisation from scratch. With this relaxation, we see that LLMs show promise.

### 4.1    METHODOLOGY

**Evaluation dataset.** We collected 13 natural language theorems and their proofs from various sources such as ProofWiki, university courses etc., of varying proof technique, domain and difficulty level. We carefully checked if a similar proof is already formalised in Lean (in mathlib or elsewhere on the internet). While in some cases a similar proof does appear, in all cases the structure of our NL proof was significantly different or different formalisms were used (we provide details for each theorem in Appendix D). Since we measure autoformalisation performance according to the faithfulness of the output proof to our NL proof, we believe there is minimal risk that our output were memorized by Codex from its training data. We also used a few hand-written natural language proofs. Some of these are listed below (the full list is in Section C.1)

1. *Absolute Value Function is Convex* (`abs_convex`): Let $f : \mathbb{R} \to \mathbb{R}$ be the absolute value function on the real numbers. Then $f$ is convex.
2. *Schur's Lemma* (`schur_lemma`): Let $V$ and $W$ be vector spaces; and let $\rho_V$ and $\rho_W$ be irreducible representations of $G$ on $V$ and $W$ respectively. If $V$ and $W$ are not isomorphic, then there are no nontrivial representations of $G$ on $V$ and $W$ respectively.

3. *Schur's Inequality* (`schur_ineq`): Let $x, y, z \in \mathbb{R}_{\geq 0}$ be positive real numbers such that $x \geq y \geq z \geq 0$. Let $t \in \mathbb{R}, t > 0$ be a (strictly) positive real number. Then: $x^t(x-y)(x-z) + y^t(y-z)(y-x) + z^t(z-x)(z-y) \geq 0$

4. *Contraction Mapping theorem* (`contraction_mapping`): Let $B$ be a Banach space, $M$ a closed subset of $B$, and $\Phi$ a mapping from $M$ to $M$ such that for some $k \in [0, 1)$, $\|\Phi(x) - \Phi(y)\| \leq k\|x - y\|$ for any two points $x$ and $y$ in $M$. Then there is a unique point $z$ in $M$ such that $\Phi(z) = z$.

```
/--`theorem`
Power Set is Closed under Intersection
Let $S$ be a set.
Let $\powerset S$ be the power set of $S$.
Then:
:$\forall A, B \in \powerset S: A \cap B \in \powerset S$
`proof`
Let $A, B \in \powerset S$.
Then by the definition of power set, $A \subseteq S$ and $B \subseteq S$.
From Intersection is Subset we have that $A \cap B \subseteq A$.
It follows from Subset Relation is Transitive that $A \cap B \subseteq S$.
Thus $A \cap B \in \powerset S$ and closure is proved.
{{qed}}
--/
theorem power_set_intersection_closed {α : Type*} (S : set α) : ∀ A B ∈ 𝒫 S, (A ∩ B) ∈ 𝒫 S :=
begin
  -- $A$ and $B$ are sets. $A$ and $B$ belong to power set of $S$
  assume (A : set α) (hA : A ∈ 𝒫 S) (B : set α) (hB : B ∈ 𝒫 S),
  -- Then $A ⊆ S$ and $B ⊆ S$, by power set definition
  have h1 : (A ⊆ S) ∧ (B ⊆ S), from by {split,apply set.subset_of_mem_powerset,exact hA,apply
    set.subset_of_mem_powerset,exact hB},
  -- Then $(A ∩ B) ⊆ A$, by intersection of set is a subset
  have h2 : (A ∩ B) ⊆ A, from by apply set.inter_subset_left,
  -- Then $(A ∩ B) ⊆ S$, by subset relation is transitive
  have h3 : (A ∩ B) ⊆ S, from by {apply set.subset.trans h2 h1.left},
  -- Hence $(A ∩ B) ∈ 𝒫 S$, by power set definition
  show (A ∩ B) ∈ 𝒫 S, from by {apply set.mem_powerset h3},
end
```

Figure 2: One of the examples in the prompt for the full-proof-with-comments format

**Proof formats.** We experimented with several *formal proof formats*; all formats use forward reasoning (see Appendix A for more on forward vs. backward reasoning). Proof formats differ in their ease of translation by a language model and in their utility to a user. Proof formats vary across two axes: the level of detail and whether the formal proof has comments. The three levels of detail are the following.

*Full proof.* This corresponds to the complete proof.

*Proof outline.* This consists of the main steps of the proof listed in order. In Lean code, an outline is given by a series of `have` statements with `sorry` as a placeholder for the intermediate proofs. Although an outline contains far less information than a full proof, a tool that is capable of producing good outlines could still be valuable to a user since one could, in principle, iteratively produce outlines of the main proof and all its steps, until one is left with trivial steps that can handled by automation.

*Proof outline with premises.* This format is at an intermediate level of detail: each proof step is listed along with a list of *premises* from which it can be deduced. This is done by introducing a fictitious new Lean tactic `auto` that takes as arguments the list of theorems that go into a proof along with an optional list of Lean tactics that may be helpful.

Proofs at each level of detail can be used as is or combined with comments (each step preceded by a comment explaining that step). This results in a total of *six formats*. For an example, see the formal proof in Figure 2.

**Prompts.** We designed few-shot prompts (one for each format) for our chosen set of theorems. The prompts consist of three theorems with corresponding proofs; we illustrate one such theorem for full proof with comments in Figure 2 and the full prompt can be found in Section B.1.

**Hyperparameters.** We initially considered four temperatures 0, 0.2, 0.4 and 0.8. We sampled three outputs for each of the latter three, and a single output for the former.

**Evaluation.** To generate proofs in different formats, we queried Codex with a prompt consisting of example natural language proofs and the corresponding step-wise Lean proofs in the appropriate format, followed by the natural language proof to be translated. For evaluation, we manually inspected the generated outputs via the following grading scheme.

*Theorem statement formalisation:* 0 if the output is incorrect; 1 if the output is somewhat correct; 2 if the output is fully correct.

*Proof formalization.* 0 if the output does not help with formalising the complete proof; 1 if the output slightly decreases the effort needed to formalise the complete proof; 2 if the output makes it substantially easier to formalise the complete proof; 3 if the output only needs few minor corrections; 4 if the output is fully correct.

As manual grading of proof output by Codex is time-consuming, after a preliminary analysis we focused on three formats, namely those with comments, and on temperatures 0.4 and 0.8, as the results were better in these cases. Outputting proof formats with comments might help Codex relate the natural language proof with the formalised Lean proof at a more granular level. Hence, for each theorem, we analysed 18 Codex completions (6 per proof format). The model was initially given the task of formalising the theorem statement as well as the proof. Later, we also prompted the model with correct Lean statements for some theorems and assigned it the task of the formalisation of the respective proofs.

### 4.2 Results

Overall, we found that the generated proofs were well aligned with the natural language proofs and also well-structured as per Lean style. These proofs could therefore be used as a good starting points for formalisation assistance as illustrated in Figure 3. We summarise the scores given to the Codex completions after manual inspection in Figure 6. No completion received a perfect score of 4. For 8 of the 13 theorems, at least one Codex completion (out of 18), was marked 3. The other 5 theorems got a maximum of 2. The main sources of lower scores were errors related to incorrect natural language statement translations that could be mathematically incorrect, irrelevant or invalid Lean code. Some lower scores were also due to step repetitions. These results are given and errors analysed in detail in Section C; here we present a synopsis.

The best proof format depended on the nature of the proof, with more detailed formats often better (i.e., with more intermediate details) for harder to formalise proofs, while the results were good for all formats for easy to formalise statements. Including the correct Lean theorem statements did not show a clear improvement. However, in the case of the lowest scoring theorem without the correct statements, including the correct statement improved the score from 1 to 2.

**Capabilities shown by Codex.** In `schur_ineq`, the natural language proof had statements that simply mentioned that all the terms are non-negative and concluded the proof. Interestingly, Codex completions had these details formalised as intermediate steps.

Codex sometimes expanded a definition instead of using the mathlib definition directly; for example, $A = A^T$ instead of `is_symm A`. Codex also generated plausible theorem names, which were in line with mathlib style.

**Errors in completions.** Occasionally the completion had invalid Lean syntax, e.g., `x ≥ y ≥ z ≥ 0` (copied directly from the natural proof) instead of the valid Lean syntax `x ≥ y ∧ y ≥ z ∧ z ≥ 0`. There was an instance where Codex generated a proof in what seemed like a different language. In some cases, we observed that the completion used an undeclared variable in the proof. For example, it declared `t > 0` without introducing `t: ℝ`.

There were a few instances where proof-steps were syntactically correct but mathematically incorrect, for instance stating `|α*x + β*y| = α*|x| + β*|y|` instead of the triangle inequality.

Sometimes natural language translations were wrong, although they were mathematically valid statements and valid Lean code. For example, in `schur_lemma`, Codex confuses a homomorphism being "non-zero" with a homomorphism being "nowhere zero". In a proof of the

```
theorem abs_convex : ∀ (x y : ℝ), ∀ (α β : ℝ), α ≥ 0 ∧ β ≥ 0 ∧ α + β = 1 →
  abs (α * x + β * y) ≤ (α * abs x + β * abs y) :=
begin
  assume (x y : ℝ) (α β : ℝ) (h1 : α ≥ 0 ∧ β ≥ 0 ∧ α + β = 1),
  -- expand the absolute value function
  have h2 : abs (α * x + β * y) = (α * x + β * y), from by rw abs_of_nonneg (add_nonneg (mul_nonneg h1.left (abs_nonneg x)) (mul_nonneg h1.right
    (abs_nonneg y))),
  -- apply the triangle inequality
  have h3 : abs (α * x + β * y) ≤ abs (α * x) + abs (β * y), from by apply abs_add,
  -- apply the absolute value of product
  have h4 : abs (α * x + β * y) ≤ abs α * abs x + abs β * abs y, from by { rw [ ← abs_mul α x, ←
    abs_mul β y] at h3, exact h3 },
  -- apply the absolute value definition
  have h5 : abs (α * x + β * y) ≤ α * abs x + β * abs y, from by { rw abs_of_nonneg h1.left at h4,
    rw abs_of_nonneg h1.right.left at h4, exact h4 },
  -- apply the definition of absolute value function
  have h6 : abs (α * x + β * y) ≤ α * abs x + β * abs y, from by rw abs_of_nonneg h1.right at h5,
  show abs (α * x + β * y) ≤ α * abs x + β * abs y, from h6 h5.
end
```

Figure 3: Correction of a Codex completion of *Absolute Value Function is Convex*. The text highlighted in red is to be deleted and the text highlighted in green and underlined is to be added.

contraction_mapping, Codex defines a sequence $x$ to be $x(i) := \phi(x(i))$, instead of the inductive definition: $x(0) := x\_0$ and $x(i + 1) := \phi(x(i))$.

Sometimes the Codex completion contained contradicting proof steps (even in cases not involving proof of contradiction). As an example, in a generated proof of schur_lemma, there were two statements, one stating that the kernel of a homomorphism is trivial, and the next one stating the opposite.

**Hallucinations.** Codex sometimes output names that looked realistic but are not present in mathlib, e.g. convex_function instead of convex_on.

**Deviation from Natural Language Proof.** Occasionally the output proof formats deviated from the natural language proofs, with Codex merging different proofs from distinct sources, leading to conflicting notation.

## 5 LIMITATIONS AND FUTURE WORK

With our techniques, Codex shows promising performance for autoformalisation of docstring-style theorem statements and for proofs There are many avenues for future work.

Using docstrings from mathlib in the present form does not give adequate examples of complex LATEX formulas and of some mathematical idioms. An additional database of prompts targeting these could address this. Further, we can make use of Lean's easily extensible syntax to incorporate more mathematical notation. One way to improve selection is to reverse the translation to obtain NL text from Lean code and use a similarity measure with the original text to select the best completion. While preliminary experiments show this is useful, presently it is too slow to be practical.

Better equality testing for theorem statements will also result in better filtering. Unlike program synthesis, for theorem autoformalisation, there is no obvious counterpart of unit tests. Better equality testing with the correct Lean formal statement, however, can serve the role of unit tests.

Outputs generated by our framework can be a useful starting point for formalisation, potentially saving considerable time and effort. Presently about one or two out of up to 18 completions tend to be useful; recognizing these automatically will reduce effort. We did not experiment with interactive formalisation as evaluation becomes harder. It would be interesting to combine our framework with automatic proof search or repair ideas: partial proofs, being close to complete proofs, can serve as a good starting point for proof search. This could result in an autoformalisation system that is closer to being autonomous.

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

# A    THE LEAN PROVER AND REASONING STYLES

We use the Lean interactive theorem prover. The Lean mathematical library (mathlib) includes most of the standard undergraduate curriculum. Many advanced results have also been formalised building on mathlib. mathlib is monolithic by design, ensuring that formalisations of different parts of mathematics can be combined easily. The presence of this library to use and also the resulting standardization of terminology makes Lean an attractive target for autoformalisation.

The latest version of Lean, Lean 4, is a full-fledged programming language with a fast runtime in addition to being an interactive theorem prover. This allows a seamless integration of proofs, programs and meta-programs. We use Lean 3 for one set of experiments and Lean 4 for the other because, at the time of writing, mathlib was only partially available in Lean 4 (via a partial binary port). Hence we use Lean 4 where its additional capabilities are important and Lean 3 where these are not used and the larger library is of greater value.

Mathematical proofs in the literature usually use *forward reasoning*, where a series of conclusions are deduced starting with the hypotheses from previous conclusions and known results. A notable exception is proof by induction, where we begin with the goal and reduce to sub-goals for the base case and the induction step. Reasoning starting with goals is called *backward reasoning*.

In Lean (and similar systems), proofs can use both forward and backward reasoning. However, backward reasoning allows for much more powerful automation within the *tactic mode*, at the cost of readability. Lean has powerful tactics like `rw` (applies an equation or if and only if statement) and `apply`(tries to match the goal against the conclusion of the lemma being used) to deal with these.

We illustrate the different styles of reasoning by proving in Lean in various ways the result $3 < 7$ using only the results $\forall n \in \mathbb{N}, 0 < n + 1$ and $\forall n, m \in \mathbb{N}, n \leq m \implies n + 1 \leq m + 1$. In Lean these are the theorems `Nat.zero_lt_succ` and `Nat.succ_lt_succ`. Four proofs (in Lean 4, which also work in Lean 3) are shown in Figure 4.

```
theorem three_lt_seven₁ : 3 < 7 :=
    have l₁ : 0 < 4 :=
    Nat.zero_lt_succ 3
    have l₂ : 1 < 5 :=
    Nat.succ_lt_succ l₁
    have l₃ : 2 < 6 :=
    Nat.succ_lt_succ l₂
    Nat.succ_lt_succ l₃
```

```
theorem three_lt_seven₂ : 3 < 7 :=
Nat.succ_lt_succ (
Nat.succ_lt_succ (Nat.succ_lt_succ
    (Nat.zero_lt_succ 3)))

theorem three_lt_seven₃ : 3 < 7 :=
by  repeat (apply Nat.succ_lt_succ)
    apply Nat.zero_lt_succ

theorem three_lt_seven₄ : 3 < 7 :=
    by decide
```

Figure 4: Four proofs of $3 < 7$ in Lean

The first proof is a typical forward reasoning proof making deductions from known results and previous deductions. The second proof is simply this in a more concise form. The third and fourth proof use backward reasoning in tactic mode. Tactics are powerful algorithms for finding proofs. It is evident that the above backward proofs are more concise and will be easier for a user to produce. However, in practice a complex mathematical proof in Lean will have *mixed* forward and backward reasoning, with forward reasoning taking the form of a sequence of lemmas (in the form of `have` statements) leading to the main theorem and backward reasoning used in the proof of each lemma.

## B  Theorem Statement Translation : Further Details

We sketch more details of the various steps in translating sentences.

### B.1  Full Example prompt

The full prompt for Figure 1 is in Figure 5. As mentioned earlier, no completion elaborated when we used fixed prompts.

---

**Input text:** "If a vector space has dimension '2' then it is finite dimensional."

---

**Prompt:**

```
/−− If a vector space has a finite basis, then it is finite−dimensional. −/
theorem {K : Type u} {V : Type v} [division_ring K] [add_comm_group V] [module
    K V] {ι : Type w} [fintype ι] (h : basis ι K V) : finite_dimensional K V :=

/−− A finite dimensional space is nontrivial if it has positive 'finrank'. −/
theorem {K : Type u} {V : Type v} [division_ring K] [add_comm_group V] [module
    K V] (h : 0 < finite_dimensional.finrank K V) : nontrivial V :=

/−− A finite dimensional space that is a subsingleton has zero 'finrank'. −/
theorem {K : Type u} {V : Type v} [division_ring K] [add_comm_group V] [module
    K V] [h : subsingleton V] : finite_dimensional.finrank K V = 0 :=

/−− A vector space has dimension at most '1' if and only if there is a single vector of which all
    vectors are multiples. −/
theorem {K : Type u} {V : Type v} [division_ring K] [add_comm_group V] [module
    K V] : module.rank K V ≤ 1 ↔ ∃ (v₀ : V), ∀ (v : V), ∃ (r : K), r · v₀ = v :=

/−− If a vector space has dimension '2' then it is finite dimensional. −/
theorem
```

---

**Codex Completion:**

```
{K : Type u} {V : Type v} [division_ring K] [add_comm_group V] [module K V] (h
    : module.rank K V = 2) : finite_dimensional K V
```

---

**Post-processed code in Lean 4:**

```
theorem ∀ {K : Type u} {V : Type v} [inst : DivisionRing K] [inst_1 :
    AddCommGroup V] [inst_2 : Module K V],
  Module.rank K V = 2 → FiniteDimensional K V
```

---

Figure 5: Complete prompt used in Figure 1.

### B.2  Parsing, translation and auto-correction

Given a Codex completion, we first (attempted to) parse this and extracted *identifiers* from the syntax. These were *translated* and *auto-corrected* before re-parsing. The translation step is necessary as the prompt data we had available was in Lean 3, as is most of the data in GitHub on which Codex is trained. Thus the completions usually use Lean 3/mathlib terminology. Using a prebuilt dictionary, we translated the Lean 3/mathlib identifiers to those used by the binary port (binport) of mathlib, with auto-correction attempted for those that did not have valid translations. Both the dictionary and auto-correction are based on transformations of two forms: case transformations (for example camel-case versus snake-case) and dropping or adding segments of the form is or has.

### B.3 SELECTION

If more than one completion was correctly elaborated (which is typical when at least one completion is elaborated), we selected the best completion by *voting*. Namely, we first group elaborated completions together into groups whose members can be proved to be equal using a certain tactic. The tactic we used is one that slightly extends *reflexivity* (i.e., *definitional equality*). The chosen answer was the first member of the largest group. In practice, as the present tactic for proving equality is weak, in most cases this simply picked the first completion of Codex that was valid (i.e., that elaborated).

### B.4 SENTENCE SIMILARITY

We used Sentence-Similarity library (Reimers & Gurevych, 2019) for calculating the sentence embeddings of the doc-strings. We used all-mpnet-base-v2 model, a pretrained transformer model finetuned over 1 Billion sentence pairs from multiple datasets. This model provided the best quality embeddings among the hosted models on the library at the time of writing this paper. We computed the cosine similarity of the sentence-embeddings generated from the input docstring with the collection of mathlib docstrings and selected Top-K similar docstrings based on the similarity scores and their corresponding Lean statements for the example prompts.

### B.5 KEYWORD-BASED PROMPTING

The purpose of input-dependent prompting for theorem statements is to retrieve a collection of examples that contain all the relevant details to formalise a given statement, and this is achieved to a large extent using sentence similarity. However, in optimising for overall similarity, this approach may leave out smaller details that are nevertheless crucial for formalising the statement correctly. To address this, we introduce a method of prompting based on keywords that complements sentence-similarity based retrieval. We used the YAKE keyword extraction tool (Campos et al., 2018) to extract the keywords from mathlib and store them in a convenient format. When preparing the prompt for formalising a sentence, we extract the keywords and retrieve a few examples for each keyword, in addition to using sentence similarity.

### B.6 MORE COMPLEX PROMPTS

While our translations were often successful with the docstring-style prompts we considered, the performance was poor with prompts that contained:

- Complex formulas in LaTeX.
- Certain idioms such as "... the intersection of every $d + 1$ of these sets ..."

It appears that a major reason for this was the lack of good examples in our database. Indeed, in the case of proof translation, our prompts did have complex LaTeX formulas and Codex successfully generated translated input text with (similar) complex LaTeX formulas.

## C   PROOF TRANSLATION RESULTS AND ANALYSIS

### C.1   EVALUATION DATASET

The following is the full list of theorems.

1. *Absolute Value Function is Convex* (`abs_convex`): Let $f : \mathbb{R} \to \mathbb{R}$ be the absolute value function on the real numbers. Then $f$ is convex.
2. *Symmetric Real Matrices have Real Eigenvalues* (`symm_real_mat_real_eigenvalue`): Every real symmetric matrix has real eigenvalues.
3. *Schur's Lemma* (`schur_lemma`): Let $V$ and $W$ be vector spaces; and let $\rho_V$ and $\rho_W$ be irreducible representations of $G$ on $V$ and $W$ respectively. If $V$ and $W$ are not isomorphic, then there are no nontrivial representations of $G$ on $V$ and $W$ respectively.
4. $\mathbb{R}^n$ *is paracompact* (`rn_paracompact`): $\mathbb{R}^n$ is paracompact for all $n$.
5. *Schur's Inequality* (`schur_ineq`): Let $x, y, z \in \mathbb{R}_{\geq 0}$ be positive real numbers such that $x \geq y \geq z \geq 0$. Let $t \in \mathbb{R}, t > 0$ be a (strictly) positive real number. Then: $x^t(x-y)(x-z) + y^t(y-z)(y-x) + z^t(z-x)(z-y) \geq 0$.
6. *Overflow theorem* (`overflow`): Let $F$ be a set of first-order formulas which has finite models of arbitrarily large size. Then $F$ has an infinite model.
7. *Density of Irrational Orbit* (`density_irr_orbit`): The fractional parts of the integer multiples of an irrational number form a dense subset of the unit interval.
8. *Contraction Mapping theorem* (`contraction_mapping`): Let $B$ be a Banach space, $M$ a closed subset of $B$, and $\Phi$ a mapping from $M$ to $M$ such that for some $k \in [0, 1)$,

$$\|\Phi(x) - \Phi(y)\| \leq k\|x - y\|$$

   for any two points $x$ and $y$ in $M$. Then there is a unique point $z$ in $M$ such that $\Phi(z) = z$.
9. *Class number of a PID* (`class_num_pid`): The class number of a number field $K$ is 1 if and only if the ring of integers is a PID.
10. *Bipartite Graph is two colorable* (`bipartite_iff_two_colorable`): Let $G$ be a graph. Then $G$ is 2-colorable if and only if $G$ is bipartite.
11. *p-adic Units* (`padic_units`): Given a prime number $p$ and a natural number $x$, if $x$ is coprime to $p$, then $x$ is a unit in the $p$-adic integers.
12. *Nesbitt's Inequality* (`nesbitt_ineq`): Let $a$, $b$ and $c$ be positive real numbers. Then:
    $$\frac{a}{b+c} + \frac{b}{a+c} + \frac{c}{a+b} \geq \frac{3}{2}$$
13. *Bernoulli's Polynomial Evaluation* (`bernoulli_polynomial_eval`): Given a natural number $n$ and a rational $x$, let $B_n(x)$ denote the $n$-th Bernoulli polynomial evaluated at $x$. Then,
    $$B_n(1 + x) = B_n(x) + nx^{n-1}$$

These theorems involve different types of proving techniques such as proof by contradiction, induction, algebraic manipulations etc. They belong to different domains such as topology, algebraic number theory, analysis, group theory, linear algebra, representation theory and graph theory. They are also of varying difficulty levels. For the first 11 theorems, we asked Codex for both Lean theorem statements and proofs. For the remaining 2, we included the correct Lean theorem statement in the example prompt and asked Codex for the Lean proof format.

### C.2   RESULTS

This section summarises our main findings in the proof autoformalisation experiments. The results of evaluation of proof translation are summarised in Figure 6. Out of the 288 completions, 22 (i.e., about one in thirteen) were scored as 3.

### C.2.1   CAPABILITIES SHOWN BY CODEX

The output proof formats were well structured and Codex was able to break the full proof into its main steps. For example, in `bernoulli_polynomial_eval`, the natural language proof had two major proof statements in a single sentence and Codex broke it down into the two relevant pieces. Also, for `schur_ineq`, we observed that the natural language proof had statements that

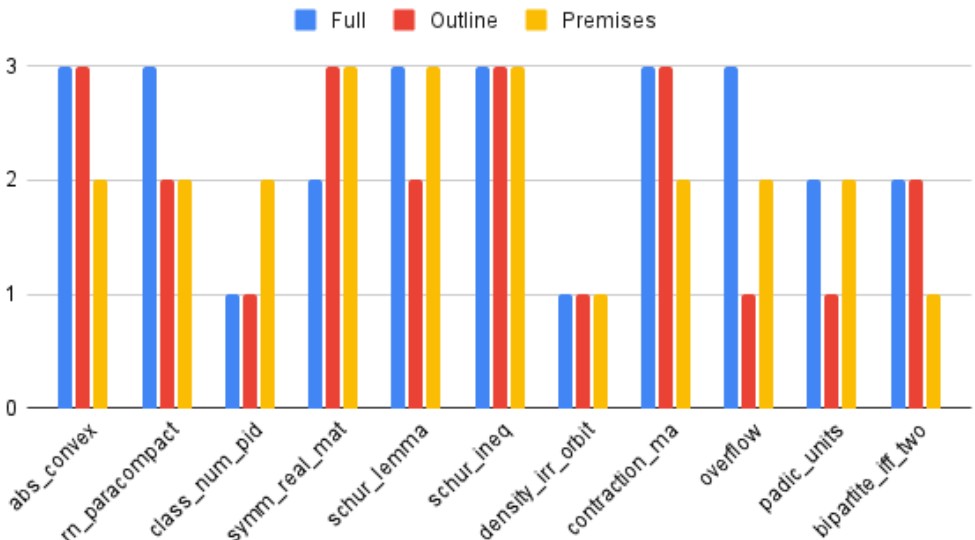

Figure 6: Max scores of the 11 theorems, for the other 2 theorems we included correct Lean statements in the prompt

simply mentioned that all the terms are non-negative and concluded the proof. Interestingly, Codex completions had these details formalised as intermediate steps.

While formalising statements, Codex sometimes expanded a definition instead of using the mathlib definition directly. Interestingly, in a completion for `abs_convex`, Codex directly outputted the precise simplified statement that needed to be proved, possibly taking cues from the natural language proof provided – it didn't use the `convex_on` definition from mathlib and directly outputted $\forall$ `(x y : ` $\mathbb{R}$`),` $\forall$ `(`$\alpha$ $\beta$ ` : ` $\mathbb{R}$`),` $\alpha \geq 0 \wedge \beta \geq 0 \wedge \alpha + \beta = 1 \rightarrow$ `abs (`$\alpha$ `* x +` $\beta$ `* y)` $\leq$ `(`$\alpha$ `* abs x +` $\beta$ `* abs y)` as the theorem statement to be proved. Also, while formalising the `symm_real_mat_real_eigenvalue` lemma, some completions had the hypothesis that $A = A^T$ instead of `is_symm A`. Maybe input-dependent prompting for proofs could direct Codex to output relevant mathlib definitions more frequently instead of using their expanded forms.

Codex also utilized contextual mathematical knowledge that was missing in the theorem statements but present in natural language proofs. For example, while formalising the proof of `symm_real_mat_real_eigenvalue`, the statement of the natural language prompt simply said that every real symmetric matrix has real eigenvalues. One Codex completion started with a complex eigenvalue, possibly getting the context right from the natural language proof.

We also observed that Codex generated plausible theorem names, which were in line with mathlib style.

### C.2.2 ERRORS

There were very few instances where the outputs did not follow valid Lean syntax. For instance, In `schur_ineq`, Codex completion contained incorrect Lean expression `x` $\geq$ `y` $\geq$ `z` $\geq$ `0` (copying directly from the natural proof) instead of the correct Lean expression `x` $\geq$ `y` $\wedge$ `y` $\geq$ `z` $\wedge$ `z` $\geq$ `0`. There was one instance where Codex generated a proof format in a different language.

In some cases, we observed that the completion used an undeclared variable in the proof. For example, it declared `t > 0` without introducing `t:` $\mathbb{R}$.

There were a few instances where proof-steps were mathematically incorrect. For instance, in `abs_convex`, Codex didn't assume $\alpha$ and $\beta$ as positive numbers, however it used this condition

in the proof. It also wrongly stated the triangle inequality as equality: $|\alpha * x + \beta * y| = \alpha * |x| + \beta * |y|$.

There were also instances where natural language translations were wrong, although they were mathematically valid statements and valid Lean code. For example, in `schur_lemma`, Codex confuses a homomorphism being "nonzero" with a homomorphism being "nowhere" zero . Also, in some completions, the definition of "dense" in the `density_irr_orbit` was used incorrectly. In a proof of the `contraction_mapping`, Codex defines a function `x` to be `x(i) := φ (x (i))`. This is wrong since it is using `x(i)` to define `x(i)`. The correct definition is inductive : `x(0) := x` and `x(i) := φ (x (i - 1))` for $i \geq 1$, and might have been difficult for Codex to deduce since induction was not explicitly mentioned. In a few instances, it also mistook absolute value for norm.

There were also cases when Codex got stuck at a particular step and looped until it completed the 2000 tokens. These repetitions occured in different forms – frequently applying the same lemma or writing a set of `have` statements. At times, the proof was looped in an inductive fashion. For example, in the `contraction_mapping`, Codex defined `φ (x n) = x n - 1` followed by `have h1 : φ (x 1) = x 0`, `have h1 : φ (x 2) = x 1`, and so on.

Sometimes Codex outputted contradicting proof steps (we report cases not involving proof of contradiction). As an example, in the generated proof of `schur_lemma`, there were two statements, one stating that the kernel is trivial, and the next one stating the opposite.

### C.2.3 OTHER OBSERVATIONS

**Hallucinations.** We say that Codex hallucinates if the definition/theorem name generated is not present in mathlib even though the correct definition/theorem is present in the mathlib. We have observed that hallucination is common. For example, in few instances while formalising `abs_convex`,Codex outputs `convex_function` instead of `convex_on` and similar made up definitions/theorem names that looked realistic but are not present in mathlib.

**Deviation from Natural Language Proof.** There were very few instances where the output proof formats deviated from the natural language proofs. When we included the correct formalised statement to Codex for the `rn_paracompact`, Codex merged different proofs from distinct sources, leading to conflicting notation. Also, Codex formalised the contra-positive version of theorem statement of `schur_lemma` given in the natural language proof.

## D COMPLETE DATA AND ANALYSIS FOR PROOF TRANSLATIONS

### D.1 FEW-SHOT PROMPTS FOR DIFFERENT PROOF FORMATS

In this section, we show the example fixed few-shot prompts for the different proof formats that we provided to Codex. We designed these prompts using three theorems – *Power set is closed under intersection*, *Square of sum, and Identity of Group is Unique.* For each of these theorems, we used the NL theorem statements and proofs in raw formats and combined them with the corresponding Lean proof format. These Lean proof formats have interleaved NL comments picked from the NL proof. We concatenated the NL theorem statement and proof that has to be formalised with these example prompts and provided them to Codex for output.

#### D.1.1 PROOF OUTLINE

Following are the example few-shot prompts for the proof outline format.

```
/−−'theorem'
Power Set is Closed under Intersection
Let $S$ be a set.
Let $\powerset S$ be the power set of $S$.
Then:
:$\forall A, B \in \powerset S: A \cap B \in \powerset S$
'proof'
Let $A, B \in \powerset S$.
Then by the definition of power set, $A \subseteq S$ and $B \subseteq S$.
From Intersection is Subset we have that $A \cap B \subseteq A$.
It follows from Subset Relation is Transitive that $A \cap B \subseteq S$.
Thus $A \cap B \in \powerset S$ and closure is proved.
{{qed}}
−/
theorem power_set_intersection_closed {α : Type*} (S : set α) : ∀ A B ∈ 𝒫 S, (A ∩ B) ∈
    𝒫 S :=
begin
  −− $A$ and $B$ are sets. $A$ and $B$ belong to power set of $S$
  assume (A : set α) (hA : A ∈ 𝒫 S) (B : set α) (hB : B ∈ 𝒫 S),
  −− Then $A ⊆ S$ and $B ⊆ S$, by power set definition
  have h1 : (A ⊆ S) ∧ (B ⊆ S), from sorry,
  −− Then $(A ∩ B) ⊆ A$, by intersection of set is a subset
  have h2 : (A ∩ B) ⊆ A, from sorry,
  −− Then $(A ∩ B) ⊆ S$, by subset relation is transitive
  have h3 : (A ∩ B) ⊆ S, from sorry,
  −− Hence $(A ∩ B) ∈ 𝒫 S$, by power set definition
  show (A ∩ B) ∈ 𝒫 S, from sorry,
end

/−−'theorem'
Square of Sum
:$\forall x, y \in \R: \paren {x + y}^2 = x^2 + 2 x y + y^2$
'proof'
Follows from the distribution of multiplication over addition:
{{begin−eqn}}
{{eqn | l = \left({x + y}\right)^2
     | r = \left({x + y}\right) \cdot \left({x + y}\right)
}}
{{eqn | r = x \cdot \left({x + y}\right) + y \cdot \left({x + y}\right)
     | c = Real Multiplication Distributes over Addition
}}
{{eqn | r = x \cdot x + x \cdot y + y \cdot x + y \cdot y
     | c = Real Multiplication Distributes over Addition
}}
{{eqn | r = x^2 + 2xy + y^2
     | c =
}}
{{end−eqn}}
```

```
{{qed}}
−/
theorem square_of_sum (x y : ℝ) : (x + y)^2 = (x^2 + 2*x*y + y^2)
begin
  −− expand the power
  calc (x + y)^2 = (x+y)*(x+y) : by sorry
  −− distributive property of multiplication over addition gives:
  ... = x*(x+y) + y*(x+y) : by sorry
  −− applying the above property further gives:
  ... = x*x + x*y + y*x + y*y : by sorry
  −− rearranging the terms using commutativity and adding gives:
  ... = x^2 + 2*x*y + y^2 : by sorry,
end
```

```
/−−`theorem`
Identity of Group is Unique
Let $\struct {G, \circ}$ be a group. Then there is a unique identity element $e \in G$.
`proof`
From Group has Latin Square Property, there exists a unique $x \in G$ such that:
:$a x = b$
and there exists a unique $y \in G$ such that:
:$y a = b$
Setting $b = a$, this becomes:
There exists a unique $x \in G$ such that:
:$a x = a$
and there exists a unique $y \in G$ such that:
:$y a = a$
These $x$ and $y$ are both $e$, by definition of identity element.
{{qed}}
−/
theorem group_identity_unique {G : Type*} [group G] : ∃! e : G, ∀ a : G, e * a = a ∧ a * e =
    a :=
begin
  −− Group has Latin Square Property
  have h1 : ∀ a b : G, ∃! x : G, a * x = b, from sorry,
  have h2 : ∀ a b : G, ∃! y : G, y * a = b, from sorry,

  −− Setting $b = a$, this becomes:
  have h3 : ∀ a : G, ∃! x : G, a * x = a, from sorry,
  have h4 : ∀ a : G, ∃! y : G, y * a = a, from sorry,

  −− These $x$ and $y$ are both $(1 : G)$, by definition of identity element
  have h5 : ∀ a : G, classical.some (h3 a) = (1 : G), from sorry,
  have h6 : ∀ a : G, classical.some (h4 a) = (1 : G), from sorry,

  show ∃! e : G, ∀ a : G, e * a = a ∧ a * e = a, from by {
    use (1 : G),
    have h7 : ∀ e : G, (∀ a : G, e * a = a ∧ a * e = a) → e = 1, from by {
      assume (e : G) (h7 : ∀ a : G, e * a = a ∧ a * e = a),
      have h8 : ∀ a : G, e = classical.some (h3 a), from sorry,
      have h9 : ∀ a : G, e = classical.some (h4 a), from sorry,
      show e = (1 : G), from sorry,
    },
    sorry,
  }
end
```

### D.1.2 PROOF WITH PREMISES

Following are the example few-shot prompts for the proof outline with premises format.

```
/−−'theorem'
Power Set is Closed under Intersection
Let $S$ be a set.
Let $\powerset S$ be the power set of $S$.
Then:
:$\forall A, B \in \powerset S: A \cap B \in \powerset S$
'proof'
Let $A, B \in \powerset S$.
Then by the definition of power set, $A \subseteq S$ and $B \subseteq S$.
From Intersection is Subset we have that $A \cap B \subseteq A$.
It follows from Subset Relation is Transitive that $A \cap B \subseteq S$.
Thus $A \cap B \in \powerset S$ and closure is proved.
{{qed}}
−/
theorem power_set_intersection_closed {α : Type*} (S : set α) : ∀ A B ∈ 𝒫 S, (A ∩ B) ∈ 𝒫
    S :=
begin
  −− $A$ and $B$ are sets. $A$ and $B$ belong to power set of $S$
  assume (A : set α) (hA : A ∈ 𝒫 S) (B : set α) (hB : B ∈ 𝒫 S),
  −− Then $A ⊆ S$ and $B ⊆ S$, by power set definition
  have h1 : (A ⊆ S) ∧ (B ⊆ S), from by auto [set.subset_of_mem_powerset,
    set.subset_of_mem_powerset],
  −− Then $(A ∩ B) ⊆ A$, by intersection of set is a subset
  have h2 : (A ∩ B) ⊆ A, from by auto [set.inter_subset_left],
  −− Then $(A ∩ B) ⊆ S$, by subset relation is transitive
  have h3 : (A ∩ B) ⊆ S, from by auto [set.subset.trans],
  −− Hence $(A ∩ B) ∈ 𝒫 S$, by power set definition
  show (A ∩ B) ∈ 𝒫 S, from by auto [set.mem_powerset],
end

/−−'theorem'
Square of Sum
:$\forall x, y \in \R: \paren {x + y}^2 = x^2 + 2 x y + y^2$
'proof'
Follows from the distribution of multiplication over addition:
{{begin−eqn}}
{{eqn | l = \left({x + y}\right)^2
    | r = \left({x + y}\right) \cdot \left({x + y}\right)
}}
{{eqn | r = x \cdot \left({x + y}\right) + y \cdot \left({x + y}\right)
    | c = Real Multiplication Distributes over Addition
}}
{{eqn | r = x \cdot x + x \cdot y + y \cdot x + y \cdot y
    | c = Real Multiplication Distributes over Addition
}}
{{eqn | r = x^2 + 2xy + y^2
    | c =
}}
{{end−eqn}}
{{qed}}
−/
theorem square_of_sum (x y : ℝ) : (x + y)^2 = (x^2 + 2*x*y + y^2)
begin
  −− expand the power
  calc (x + y)^2 = (x+y)*(x+y) : by auto [sq]
  −− distributive property of multiplication over addition gives:
  ... = x*(x+y) + y*(x+y) : by auto [add_mul]
  −− applying the above property further gives:
  ... = x*x + x*y + y*x + y*y : by auto [mul_comm, add_mul] using [ring]
  −− rearranging the terms using commutativity and adding gives:
  ... = x^2 + 2*x*y + y^2 : by auto [sq, mul_comm] using [ring]
```

```
end
```

/――'theorem'
Identity of Group is Unique
Let $\struct {G, \circ}$ be a group. Then there is a unique identity element $e \in G$.
'proof'
From Group has Latin Square Property, there exists a unique $x \in G$ such that:
:$a x = b$
and there exists a unique $y \in G$ such that:
:$y a = b$
Setting $b = a$, this becomes:
There exists a unique $x \in G$ such that:
:$a x = a$
and there exists a unique $y \in G$ such that:
:$y a = a$
These $x$ and $y$ are both $e$, by definition of identity element.
{{qed}}
―/

```
theorem group_identity_unique {G : Type*} [group G] : ∃! e : G, ∀ a : G, e * a = a ∧ a * e =
    a :=
begin
  -- Group has Latin Square Property
  have h1 : ∀ a b : G, ∃! x : G, a * x = b, from by auto using [use (a⁻¹ * b)],
  have h2 : ∀ a b : G, ∃! y : G, y * a = b, from by auto using [use b * a⁻¹],

  -- Setting $b = a$, this becomes:
  have h3 : ∀ a : G, ∃! x : G, a * x = a, from by auto [h1],
  have h4 : ∀ a : G, ∃! y : G, y * a = a, from by auto [h2],

  -- These $x$ and $y$ are both $(1 : G)$, by definition of identity element
  have h5 : ∀ a : G, classical.some (h3 a).exists = (1 : G), from by auto
    [exists_unique.unique, h3, classical.some_spec, exists_unique.exists,
    mul_one],
  have h6 : ∀ a : G, classical.some (h4 a).exists = (1 : G), from by auto
    [exists_unique.unique, h4, classical.some_spec, exists_unique.exists,
    one_mul],

  show ∃! e : G, ∀ a : G, e * a = a ∧ a * e = a, from by auto [h3, h4, exists_unique.unique,
    classical.some_spec, exists_unique.exists] using [use (1 : G)],
end
```

### D.1.3 FULL PROOFS

Following are the example few-shot prompts for the full proof format.

```
/——'theorem'
Power Set is Closed under Intersection
Let $S$ be a set.
Let $\powerset S$ be the power set of $S$.
Then:
:$\forall A, B \in \powerset S: A \cap B \in \powerset S$
'proof'
Let $A, B \in \powerset S$.
Then by the definition of power set, $A \subseteq S$ and $B \subseteq S$.
From Intersection is Subset we have that $A \cap B \subseteq A$.
It follows from Subset Relation is Transitive that $A \cap B \subseteq S$.
Thus $A \cap B \in \powerset S$ and closure is proved.
{{qed}}
—/
theorem power_set_intersection_closed {α : Type*} (S : set α) : ∀ A B ∈ 𝒫 S, (A ∩ B) ∈ 𝒫
    S :=
begin
 —— $A$ and $B$ are sets. $A$ and $B$ belong to power set of $S$
 assume (A : set α)(hA : A ∈ 𝒫 S)(B : set α)(hB : B ∈ 𝒫 S),
 —— Then $A ⊆ S$ and $B ⊆ S$, by power set definition
 have h1 : (A ⊆ S) ∧ (B ⊆ S), from by {split,apply set.subset_of_mem_powerset,exact
    hA,apply set.subset_of_mem_powerset,exact hB},
 —— Then $(A ∩ B) ⊆ A$, by intersection of set is a subset
 have h2 : (A ∩ B) ⊆ A, from by apply set.inter_subset_left,
 —— Then $(A ∩ B) ⊆ S$, by subset relation is transitive
 have h3 : (A ∩ B) ⊆ S, from by {apply set.subset.trans h2 h1.left},
 —— Hence $(A ∩ B) ∈ 𝒫 S$, by power set definition
 show (A ∩ B) ∈ 𝒫 S, from by {apply set.mem_powerset h3},
end

/——'theorem'
Square of Sum
 :$\forall x, y \in \R: \paren {x + y}^2 = x^2 + 2 x y + y^2$
'proof'
Follows from the distribution of multiplication over addition:
{{begin−eqn}}
{{eqn | l = \left({x + y}\right)^2
    | r = \left({x + y}\right) \cdot \left({x + y}\right)
}}
{{eqn | r = x \cdot \left({x + y}\right) + y \cdot \left({x + y}\right)
    | c = Real Multiplication Distributes over Addition
}}
{{eqn | r = x \cdot x + x \cdot y + y \cdot x + y \cdot y
    | c = Real Multiplication Distributes over Addition
}}
{{eqn | r = x^2 + 2xy + y^2
    | c =
}}
{{end−eqn}}
{{qed}}
—/
theorem square_of_sum (x y : ℝ) : (x + y)^2 = (x^2 + 2*x*y + y^2)
begin
 —— expand the power
 calc (x + y)^2 = (x+y)*(x+y) : by rw sq
 —— distributive property of multiplication over addition gives:
 ... = x*(x+y) + y*(x+y) : by rw add_mul
 —— applying the above property further gives:
 ... = x*x + x*y + y*x + y*y : by {rw [mul_comm x (x+y),mul_comm y (x+y)], rw
    [add_mul,add_mul], ring}
 —— rearranging the terms using commutativity and adding gives:
```

```
...= x^2 + 2*x*y + y^2 : by {repeat {rw ← sq}, rw mul_comm y x, ring}
end
```

```
/——'theorem'
Identity of Group is Unique
Let $\struct {G, \circ}$ be a group. Then there is a unique identity element $e \in G$.
'proof'
From Group has Latin Square Property, there exists a unique $x \in G$ such that:
:$a x = b$
and there exists a unique $y \in G$ such that:
:$y a = b$
Setting $b = a$, this becomes:
There exists a unique $x \in G$ such that:
:$a x = a$
and there exists a unique $y \in G$ such that:
:$y a = a$
These $x$ and $y$ are both $e$, by definition of identity element.
{{qed}}
−/
theorem group_identity_unique {G : Type*} [group G] : ∃! e : G, ∀ a : G, e * a = a ∧ a * e =
    a :=
begin
 —— Group has Latin Square Property
 have h1 : ∀ a b : G, ∃! x : G, a * x = b, from by {
  assume a b : G, use a⁻¹ * b, obviously, },
 have h2 : ∀ a b : G, ∃! y : G, y * a = b, from by {
  assume a b : G, use b * a⁻¹, obviously, },

 —— Setting $b = a$, this becomes:
 have h3 : ∀ a : G, ∃! x : G, a * x = a, from
  assume a : G, h1 a a,
 have h4 : ∀ a : G, ∃! y : G, y * a = a, from
  assume a : G, h2 a a,

 —— These $x$ and $y$ are both $(1 : G)$, by definition of identity element
 have h5 : ∀ a : G, classical.some (h3 a).exists = (1 : G), from assume a :G,
  exists_unique.unique (h3 a) (classical.some_spec (exists_unique.exists (h3 a)))
  (mul_one a),
 have h6 : ∀ a : G, classical.some (h4 a).exists = (1 : G), from assume a : G,
  exists_unique.unique (h4 a) (classical.some_spec (exists_unique.exists (h4 a)))
    (one_mul a),

 show ∃! e : G, ∀ a : G, e * a = a ∧ a * e = a, from by {
  use (1 : G),
  have h7 : ∀ e : G, (∀ a : G, e * a = a ∧ a * e = a) → e = 1, from by {
   assume (e : G) (hident : ∀ a : G, e * a = a ∧ a * e = a),
   have h8 : ∀ a : G, e = classical.some (h3 a).exists, from assume (a : G),
    exists_unique.unique (h3 a) (hident a).right
    (classical.some_spec (exists_unique.exists (h3 a))),
   have h9 : ∀ a : G, e = classical.some (h4 a).exists, from assume (a : G),
    exists_unique.unique (h4 a) (hident a).left
    (classical.some_spec (exists_unique.exists (h4 a))),
   show e = (1 : G), from eq.trans (h9 e) (h6 _),
  },
  exact ⟨by obviously, h7⟩,
 }
end
```

## D.2 EVALUATION SCORES

The following tables represent the scores given to the proof format outputs produced by Codex. The tables on the right represent the scores given to the output statement (max score = 2). The tables on the left represent the scores given to the output proof format (max score = 4) (ref. Tables 3 – 13).

In the tables, **Full** corresponds to *full proof*, **Outline** corresponds to *proof outline*, and **Premises** corresponds to *proof outline with premises*. **O1**, **O2** and **O3** are the three outputs sampled at each temperature.

We also include tables to show evaluation of the outputs when we included correct Lean theorem statement in the prompt (ref. Tables 14 – 18).

There are 18 scores in total, 6 for each proof format, out of which there are 3 each for temperatures 0.4 and 0.8 respectively.

| Format | T=0.4 | | | T=0.8 | | | Max |
|---|---|---|---|---|---|---|---|
| | O1 | O2 | O3 | O1 | O2 | O3 | |
| Full | 3 | 2 | 1 | 1 | 1 | 1 | 3 |
| Outline | 2 | 3 | 2 | 3 | 3 | 3 | 3 |
| Premises | 2 | 1 | 1 | 2 | 2 | 1 | 2 |
| Max | 3 | | | 3 | | | |

| Format | T=0.4 | | | T=0.8 | | | Max |
|---|---|---|---|---|---|---|---|
| | O1 | O2 | O3 | O1 | O2 | O3 | |
| Full | 2 | 1 | 1 | 1 | 0 | 1 | 2 |
| Outline | 1 | 1 | 0 | 1 | 1 | 0 | 1 |
| Premises | 1 | 0 | 0 | 0 | 0 | 0 | 1 |
| Max | 2 | | | 1 | | | |

Table 3: `abs_convex`

| Format | T=0.4 | | | T=0.8 | | | Max |
|---|---|---|---|---|---|---|---|
| | O1 | O2 | O3 | O1 | O2 | O3 | |
| Full | 3 | 3 | 2 | 2 | 1 | 2 | 3 |
| Outline | 3 | 3 | 3 | 2 | 2 | 1 | 3 |
| Premises | 2 | 2 | 2 | 3 | 2 | 1 | 3 |
| Max | 3 | | | 3 | | | |

| Format | T=0.4 | | | T=0.8 | | | Max |
|---|---|---|---|---|---|---|---|
| | O1 | O2 | O3 | O1 | O2 | O3 | |
| Full | 2 | 2 | 1 | 1 | 1 | 2 | 2 |
| Outline | 2 | 2 | 2 | 1 | 1 | 2 | 2 |
| Premises | 2 | 2 | 2 | 2 | 1 | 2 | 2 |
| Max | 2 | | | 2 | | | |

Table 4: `schur_ineq`

| Format | T=0.4 | | | T=0.8 | | | Max |
|---|---|---|---|---|---|---|---|
| | O1 | O2 | O3 | O1 | O2 | O3 | |
| Full | 0 | 1 | 1 | 2 | 0 | 0 | 2 |
| Outline | 1 | 2 | 3 | 1 | 0 | 1 | 3 |
| Premises | 3 | 1 | 0 | 0 | 2 | 0 | 3 |
| Max | 3 | | | 2 | | | |

| Format | T=0.4 | | | T=0.8 | | | Max |
|---|---|---|---|---|---|---|---|
| | O1 | O2 | O3 | O1 | O2 | O3 | |
| Full | 0 | 0 | 1 | 0 | 1 | 0 | 1 |
| Outline | 0 | 1 | 1 | 0 | 0 | 1 | 1 |
| Premises | 1 | 1 | 0 | 0 | 1 | 0 | 1 |
| Max | 1 | | | 1 | | | |

Table 5: `symm_real_mat_real_eigenvalue`

| Format | T=0.4 | | | T=0.8 | | | Max |
|---|---|---|---|---|---|---|---|
| | O1 | O2 | O3 | O1 | O2 | O3 | |
| Full | 0 | 1 | 0 | 1 | 1 | 0 | 1 |
| Outline | 0 | 1 | 1 | 0 | 0 | 0 | 1 |
| Premises | 1 | 1 | 0 | 0 | 1 | 1 | 1 |
| Max | 1 | | | 1 | | | |

| Format | T=0.4 | | | T=0.8 | | | Max |
|---|---|---|---|---|---|---|---|
| | O1 | O2 | O3 | O1 | O2 | O3 | |
| Full | 0 | 1 | 0 | 1 | 1 | 0 | 1 |
| Outline | 1 | 0 | 1 | 0 | 1 | 0 | 1 |
| Premises | 1 | 1 | 1 | 1 | 0 | 0 | 1 |
| Max | 1 | | | 1 | | | |

Table 6: `density_irr_orbit`

| Format | T=0.4 | | | T=0.8 | | | Max |
|---|---|---|---|---|---|---|---|
| | O1 | O2 | O3 | O1 | O2 | O3 | |
| Full | 2 | 3 | 2 | 0 | 2 | 0 | 3 |
| Outline | 0 | 3 | 2 | 0 | 0 | 3 | 3 |
| Premises | 1 | 1 | 2 | 1 | 1 | 2 | 2 |
| Max | 3 | | | 3 | | | |

| Format | T=0.4 | | | T=0.8 | | | Max |
|---|---|---|---|---|---|---|---|
| | O1 | O2 | O3 | O1 | O2 | O3 | |
| Full | 1 | 1 | 1 | 1 | 1 | 1 | 1 |
| Outline | 1 | 1 | 1 | 1 | 1 | 1 | 1 |
| Premises | 1 | 1 | 1 | 1 | 1 | 1 | 1 |
| Max | 1 | | | 1 | | | |

Table 7: `contraction_mapping`

| Format | T=0.4 | | | T=0.8 | | | Max |
|---|---|---|---|---|---|---|---|
| | O1 | O2 | O3 | O1 | O2 | O3 | |
| Full | 2 | 2 | 2 | 0 | 3 | 0 | 3 |
| Outline | 0 | 0 | 0 | 2 | 2 | 0 | 2 |
| Premises | 3 | 1 | 0 | 0 | 0 | 2 | 3 |
| Max | 3 | | | 3 | | | |

| Format | T=0.4 | | | T=0.8 | | | Max |
|---|---|---|---|---|---|---|---|
| | O1 | O2 | O3 | O1 | O2 | O3 | |
| Full | 1 | 1 | 1 | 1 | 1 | 1 | 1 |
| Outline | 1 | 2 | 2 | 2 | 1 | 0 | 2 |
| Premises | 1 | 1 | 1 | 1 | 1 | 1 | 1 |
| Max | 2 | | | 2 | | | |

Table 8: `schur_lemma`

| Format | T=0.4 | | | T=0.8 | | | Max |
|---|---|---|---|---|---|---|---|
| | O1 | O2 | O3 | O1 | O2 | O3 | |
| Full | 0 | 2 | 0 | 3 | 0 | 1 | 3 |
| Outline | 0 | 1 | 0 | 0 | 0 | 1 | 1 |
| Premises | 0 | 1 | 1 | 2 | 1 | 2 | 2 |
| Max | 2 | | | 3 | | | |

| Format | T=0.4 | | | T=0.8 | | | Max |
|---|---|---|---|---|---|---|---|
| | O1 | O2 | O3 | O1 | O2 | O3 | |
| Full | 2 | 2 | 1 | 1 | 1 | 0 | 2 |
| Outline | 2 | 1 | 0 | 0 | 0 | 2 | 2 |
| Premises | 2 | 0 | 1 | 0 | 0 | 0 | 2 |
| Max | 2 | | | 2 | | | |

Table 9: `overflow`

| Format | T=0.4 | | | T=0.8 | | | Max |
|---|---|---|---|---|---|---|---|
| | O1 | O2 | O3 | O1 | O2 | O3 | |
| Full | 0 | 0 | 1 | 0 | 1 | 0 | 1 |
| Outline | 0 | 1 | 1 | 1 | 0 | 0 | 1 |
| Premises | 1 | 0 | 1 | 2 | 0 | 1 | 2 |
| Max | 1 | | | 2 | | | |

| Format | T=0.4 | | | T=0.8 | | | Max |
|---|---|---|---|---|---|---|---|
| | O1 | O2 | O3 | O1 | O2 | O3 | |
| Full | 0 | 0 | 0 | 0 | 1 | 0 | 1 |
| Outline | 1 | 2 | 1 | 1 | 1 | 0 | 2 |
| Premises | 1 | 1 | 1 | 1 | 0 | 1 | 1 |
| Max | 2 | | | 1 | | | |

Table 10: `class_num_pid`

| Format | T=0.4 | | | T=0.8 | | | Max |
|---|---|---|---|---|---|---|---|
| | O1 | O2 | O3 | O1 | O2 | O3 | |
| Full | 1 | 3 | 0 | 2 | 1 | 1 | 3 |
| Outline | 1 | 1 | 1 | 2 | 0 | 1 | 2 |
| Premises | 1 | 2 | 1 | 2 | 1 | 1 | 2 |
| Max | 3 | | | 2 | | | |

| Format | T=0.4 | | | T=0.8 | | | Max |
|---|---|---|---|---|---|---|---|
| | O1 | O2 | O3 | O1 | O2 | O3 | |
| Full | 0 | 0 | 0 | 0 | 0 | 0 | 0 |
| Outline | 0 | 0 | 0 | 0 | 0 | 0 | 0 |
| Premises | 0 | 0 | 0 | 0 | 0 | 0 | 0 |
| Max | 0 | | | 0 | | | |

Table 11: `rn_paracompact`

### D.3 INTERESTING CODEX PROOF OUTPUTS

This section contains a few examples that demonstrate some of Codex's abilities like inventing realistic theorem names and performing algebraic manipulations by analogy, as well as the influence of the example prompts on the output.

#### D.3.1 PROOF OUTPUT WHOSE FOUNDATIONAL CONCEPTS ARE NOT DEFINED IN MATHLIB

As an experiment, we prompted Codex with a theorem for which the concepts required to state it were not in mathlib. Codex was forced to invent its own names for theorems and concepts, and we observed that Codex hints plausible and realistic names for concepts in synthetic Euclidean geometry that can be included in mathlib.

*Theorem name*

Diagonals of Rhombus Bisect Each Other at Right Angles

*Theorem statement*

Let $ABCD$ be a rhombus. The diagonals $AC$ and $BD$ of $ABCD$ bisect each other at right angles.

*Natural Language proof*

By the definition of a rhombus, $AB = AD = BC = DC$.

| Format | T=0.4 | | | T=0.8 | | | Max | Format | T=0.4 | | | T=0.8 | | | Max |
|---|---|---|---|---|---|---|---|---|---|---|---|---|---|---|---|
| | O1 | O2 | O3 | O1 | O2 | O3 | | | O1 | O2 | O3 | O1 | O2 | O3 | |
| Full | 1 | 0 | 1 | 2 | 0 | 1 | 2 | Full | 0 | 0 | 0 | 1 | 0 | 0 | 1 |
| Outline | 1 | 0 | 1 | 2 | 0 | 0 | 2 | Outline | 0 | 0 | 0 | 1 | 0 | 0 | 1 |
| Premises | 0 | 1 | 1 | 1 | 0 | 0 | 1 | Premises | 0 | 0 | 0 | 0 | 0 | 0 | 0 |
| Max | 1 | | | 2 | | | | Max | 0 | | | 1 | | | |

Table 12: `bipartite_iff_two_colorable`

| Format | T=0.4 | | | T=0.8 | | | Max | Format | T=0.4 | | | T=0.8 | | | Max |
|---|---|---|---|---|---|---|---|---|---|---|---|---|---|---|---|
| | O1 | O2 | O3 | O1 | O2 | O3 | | | O1 | O2 | O3 | O1 | O2 | O3 | |
| Full | 0 | 1 | 2 | 0 | 0 | 0 | 2 | Full | 0 | 1 | 2 | 1 | 1 | 0 | 2 |
| Outline | 1 | 1 | 1 | 1 | 1 | 0 | 1 | Outline | 2 | 2 | 1 | 1 | 1 | 0 | 2 |
| Premises | 2 | 2 | 2 | 0 | 1 | 0 | 2 | Premises | 1 | 1 | 1 | 0 | 1 | 1 | 1 |
| Max | 2 | | | 1 | | | | Max | 2 | | | 1 | | | |

Table 13: `padic_units`

| Format | T=0.4 | | | T=0.8 | | | Max |
|---|---|---|---|---|---|---|---|
| | O1 | O2 | O3 | O1 | O2 | O3 | |
| Full | 0 | 1 | 1 | 1 | 0 | 1 | 1 |
| Outline | 1 | 3 | 0 | 2 | 0 | 1 | 3 |
| Premises | 1 | 0 | 1 | 0 | 0 | 1 | 1 |
| Max | 3 | | | 2 | | | |

Table 14: `nesbitt_ineq` (Correct Lean theorem statement included in the prompt)

| Format | T=0.4 | | | T=0.8 | | | Max |
|---|---|---|---|---|---|---|---|
| | O1 | O2 | O3 | O1 | O2 | O3 | |
| Full | 1 | 1 | 0 | 1 | 0 | 2 | 2 |
| Outline | 0 | 1 | 1 | 0 | 3 | 1 | 3 |
| Premises | 1 | 1 | 1 | 2 | 1 | 1 | 2 |
| Max | 1 | | | 3 | | | |

Table 15: `bernoulli_polynomial_eval` (Correct Lean theorem statement included in the prompt)

| Format | T=0.4 | | | T=0.8 | | | Max |
|---|---|---|---|---|---|---|---|
| | O1 | O2 | O3 | O1 | O2 | O3 | |
| Full | 1 | 1 | 1 | 0 | 0 | 1 | 1 |
| Outline | 0 | 1 | 2 | 1 | 1 | 0 | 2 |
| Premises | 1 | 2 | 0 | 0 | 0 | 1 | 2 |
| Max | 2 | | | 1 | | | |

Table 16: `rn_paracompact` (Correct Lean theorem statement included in the prompt)

| Format | T=0.4 | | | T=0.8 | | | Max |
|---|---|---|---|---|---|---|---|
| | O1 | O2 | O3 | O1 | O2 | O3 | |
| Full | 0 | 1 | 1 | 1 | 1 | 1 | 1 |
| Outline | 1 | 0 | 2 | 1 | 1 | 0 | 2 |
| Premises | 0 | 2 | 0 | 1 | 2 | 2 | 2 |
| Max | 2 | | | 2 | | | |

Table 17: `overflow` (Correct Lean theorem statement included in the prompt)

Without loss of generality, consider the diagonal $BD$.

Thus:

| Format | T=0.4 | | | T=0.8 | | | Max |
|---|---|---|---|---|---|---|---|
| | O1 | O2 | O3 | O1 | O2 | O3 | |
| Full | 1 | 0 | 2 | 1 | 2 | 0 | 2 |
| Outline | 1 | 2 | 0 | 1 | 0 | 2 | 2 |
| Premises | 1 | 1 | 2 | 2 | 1 | 1 | 2 |
| Max | | 2 | | | 2 | | |

Table 18: `density_irr_orbit` (Correct Lean theorem statement included in the prompt)

$\triangle ABD$ is an isosceles triangle whose apex is $A$ and whose base is $BD$.

By Diagonals of Rhombus Bisect Angles, $AC$ bisects $\angle BAD$.

From Bisector of Apex of Isosceles Triangle also Bisects Base, $AC$ bisects $BD$.

From Bisector of Apex of Isosceles Triangle is Perpendicular to Base, $AC$ bisects $BD$ at right angles.

Hence the result.

$Q.E.D.$

Codex outputs are shown in Figure 7 and Figure 8.

```
Output 1:
theorem diagonals_of_rhombus_bisect_each_other_at_right_angles (ABCD:
    rhombus):
∀ AC BD ∈ diagonals ABCD, AC.bisects BD ∧ BD.bisects AC ∧ AC.perpendicular
    BD :=
begin
  assume (AC : segment) (hAC : AC ∈ diagonals ABCD) (BD : segment) (hBD : BD ∈
    diagonals ABCD),
  have : AB = AD ∧ AB = BC ∧ AB = DC, from sorry,
  have : ∀ A B C, isosceles_triangle A B C, from sorry,
  have : isosceles_triangle A B D, from sorry,
  have : AC.bisects ∠ BAD, from sorry,
  have : AC.bisects BD, from sorry,
  have : AC.perpendicular BD, from sorry,
show AC.bisects BD ∧ BD.bisects AC ∧ AC.perpendicular BD, from sorry,
end
```

Figure 7: Codex generates ABCD as a type "rhombus" that is not present in mathlib. It also generates definitions like "bisects" and "perpendicular" that are also not formalised in mathlib and uses them plausibly. Here Codex shows capabilities of composing proofs in line with type theory.

### D.3.2 EFFECT OF EXAMPLE FEW-SHOT PROMPTS ON CODEX OUTPUT

We show the effect of few-shot prompts with an example of the formalisation of theorem *Cube of sum*. In one case, we included the theorem *square of sum* in the few shot prompts and in the other case we did not. In both of these cases, we provided the NL proof of *Cube of sum* using *Binomial Theorem*. We observe that, in the first case, Codex directly took cues from the *square of sum* theorem and deviated from the NL proof provided; while in the second case, Codex translated the NL proof of *Cube of sum* with important intermediate steps.

*Theorem statement*:

$(x + y)^3 = x^3 + 3x^2y + 3xy^2 + y^3$

*Natural language proof*:

Follows directly from the Binomial Theorem:

```
Completion 2:
theorem diagonals_of_rhombus_bisect_each_other_at_right_angles {A B C
    D : point} (h_rhombus : is_rhombus A B C D):
 bisects (segment.mk A C) (segment.mk B D) ∧ bisects (segment.mk B D)
    (segment.mk A C) :=
begin
  -- By the definition of a rhombus, $AB = AD = BC = DC$.
  have AB_eq_AD_and_BC_eq_DC : (segment.mk A B = segment.mk A D) ∧ (segment.mk
    B C = segment.mk D C), from sorry,
  -- WLOG, consider the diagonal $BD$.
  have BD_eq_DC : segment.mk B D = segment.mk D C, from sorry,
  -- Thus: $\triangle ABD$ is an isosceles triangle whose apex is $A$ and whose base is $BD$.
  have isosceles_triangle_ABD : is_isosceles_triangle A B D, from sorry,
  -- By Diagonals of Rhombus Bisect Angles, $AC$ bisects $\angle BAD$.
  have AC_bisects_BAD : bisects (segment.mk A C) (angle.mk B A D), from sorry,
  -- From Bisector of Apex of Isosceles Triangle also Bisects Base, $AC$ bisects $BD$.
  have AC_bisects_BD : bisects (segment.mk A C) (segment.mk B D), from sorry,
  -- From Bisector of Apex of Isosceles Triangle is Perpendicular to Base, $AC$ bisects $BD$ at
    right angles.
  have AC_bisects_BD_at_right_angles : bisects_at_right_angles
    (segment.mk A C) (segment.mk B D), from sorry,
  -- Hence the result.
  show bisects (segment.mk A C) (segment.mk B D) ∧ bisects (segment.mk B D)
    (segment.mk A C), from sorry,
end
```

Figure 8: Codex "suggested" designing *line segment* as a structure *segment* and treated *point* as a type, for defining 4 vertices.

```
Completion 1 (including "Square of Sum" in few shot prompts):
theorem cube_of_sum (x y : ℝ) : (x + y)^3 = (x^3 + 3*x^2*y + 3*x*y^2 + y^3) :=
begin
  calc (x + y)^3 = (x+y)*(x+y)*(x+y) : by sorry
  ... = (x+y)*(x^2 + 2*x*y + y^2) : by sorry
  ... = x*(x^2 + 2*x*y + y^2) + y*(x^2 + 2*x*y + y^2) : by sorry
  ... = x*x^2 + x*2*x*y + x*y^2 + y*x^2 + y*2*x*y + y*y^2 : by sorry
  ... = x^3 + 2*x^2*y + x*y^2 + x^2*y + 2*x*y^2 + y^3 : by sorry
  ... = x^3 + 3*x^2*y + 3*x*y^2 + y^3 : by sorry,
end
```

Figure 9: Codex deviated from the natural language proof and instead outputted the proof outline involving expanding the terms and manipulating the expressions, possibly taking cues from the *Square of Sum* prompt.

$$\forall n \in Z_{\geq 0} : (x + y)^n = \sum_{k=0}^{n} \binom{n}{k} x^{n-k} y^k$$

putting $n = 3$.

*Q.E.D.*

Codex outputs for the two cases are shown in Figure 9 and Figure 10 respectively.

```
Completion 2 (not including "Square of Sum" in few shot prompts):
theorem cube_of_sum : ∀ x y : ℝ, (x + y)^3 = x^3 + 3 * x^2 * y + 3 * x * y^2 + y^3 :=
begin
  assume (x y : ℝ),
  have : (x + y)^3 = Σ k in finset.range 3, (3 choose k) * x^(3 − k) * y^k, from sorry,
  have : (x + y)^3 = (3 choose 0) * x^(3 − 0) * y^0 + (3 choose 1) * x^(3 − 1) * y^1 + (3
      choose 2) * x^(3 − 2) * y^2 + (3 choose 3) * x^(3 − 3) * y^3, from sorry,
  have : (x + y)^3 = 1 * x^3 * y^0 + 3 * x^2 * y^1 + 3 * x * y^2 + 1 * x^0 * y^3, from sorry,
  have : (x + y)^3 = x^3 + 3 * x^2 * y + 3 * x * y^2 + y^3, from sorry,
  show (x + y)^3 = x^3 + 3 * x^2 * y + 3 * x * y^2 + y^3, from sorry,
end
```

Figure 10: Codex stuck to the natural language proof and also added some more intermediate steps important for formalisation. Although, there is a minor Lean error in "choose" definition – it should be `3.choose 2` instead of `3 choose 2`.

### D.3.3 SIMPLE ALGEBRAIC MANIPULATIONS

We experimented with Codex's ability to perform algebraic manipulations by analogy. With a few example prompts, we appended the *square of sum* NL proof, its Lean formalisation and the phrase *4th power of Sum*. Codex then outputted the 4th, 8th, ..16th power of sum. We observed that for the 4th power and 8th power of the term `(x + y)`, algebraic manipulations and final results were accurate, however, from 16th power onwards, Codex outputted partial expressions and some of these terms were incorrect. We provided the following prompt to Codex:

```
/--`theorem`
Euclid's Theorem
For any [[Definition:Finite Set|finite set]] of [[Definition:Prime
Number|prime numbers]], there exists a [[Definition:Prime
Number|prime number]] not in that [[Definition:Set|set]].
`proof`
Let $\mathbb P$ be a [[Definition:Finite Set|finite set]] of
[[Definition:Prime Number|prime numbers]].
Consider the number:
:$\displaystyle n_p = \paren {\prod_{p \mathop \in \mathbb P} p} + 1$
Take any $p_j \in \mathbb P$.
We have that:
: $\displaystyle p_j \divides \prod_{p \mathop \in \mathbb P} p$
Hence:
:$\displaystyle \exists q \in \Z: \prod_{p \mathop \in \mathbb P} p =
 q p_j$
So:
{{begin-eqn}}
{{eqn | l = n_p
      | r = q p_j + 1
      | c = [[Division Theorem]]
}}
{{eqn | ll= \leadsto
      | l = n_p
      | o = \perp
      | r = p_j
      | c = {{Defof|Coprime Integers}}
}}
{{end-eqn}}
So $p_j \nmid n_p$.
There are two possibilities:
:$(1): \quad n_p$ is [[Definition:Prime Number|prime]], which is not
in $\mathbb P$.
:$(2): \quad n_p$ is [[Definition:Composite Number|composite]].
```

```
 But from [[Positive Integer Greater than 1 has Prime Divisor]], it
must be divisible by ''some'' prime.
 That means it is divisible by a prime which is not in $\mathbb P$.
 So, in either case, there exists at least one [[Definition:Prime
Number|prime]] which is not in the original set $\mathbb P$ we
created.
{{qed}}
-/
euclid_infinitude_primes : {p : ℕ | prime p}.infinite :=
begin
  assume (primes : {p | prime p}.finite),
  let ℙ := set.finite.to_finset primes,
  let np := ℙ.val.prod + 1,
  have hprime_div_prod : ∀ p_j ∈ ℙ, p_j | ℙ.val.prod , from sorry,
  have prime_div_rem : ∀ p_j ∈ ℙ, ∃ q, ℙ.val.prod = p_j * q, from
sorry,
  have prime_not_div_n_p : ∀ p_j ∈ ℙ, ¬(p_j | np), from sorry,
  have prime_not_in_ℙ : ∃ p, prime p ∧ p ∉ ℙ, from sorry,
  show false, by {
  rcases prime_not_in_ℙ with ⟨p, hprime : prime p, hnot_in_ℙ : p ∉ ℙ⟩,
  have : p ∈ ℙ, by {apply set.mem_to_finset.mpr, exact hprime,},
  contradiction, }
end

/--`theorem`
Power Set is Closed under Intersection
Let $S$ be a [[Definition:Set|set]].
Let $\powerset S$ be the [[Definition:Power Set|power set]] of $S$.
Then:
:$\forall A, B \in \powerset S: A \cap B \in \powerset S$
`proof`
Let $A, B \in \powerset S$.
Then by the definition of [[Definition:Power Set|power set]], $A
\subseteq S$ and $B \subseteq S$.
From [[Intersection is Subset]] we have that $A \cap B \subseteq A$.
It follows from [[Subset Relation is Transitive]] that $A \cap B
\subseteq S$.
Thus $A \cap B \in \powerset S$ and [[Definition:Closure (Abstract
Algebra)|closure]] is proved.
{{qed}}
-/
power_set_intersection_closed {α : Type*} (S : set α) : ∀ A B ∈ 𝒫
S, (A ∩ B) ∈ 𝒫 S :=
begin
  assume (A : set α) (hA : A ∈ 𝒫 S) (B : set α) (hB : B ∈ 𝒫 S),
  have : (A ⊆ S) ∧ (B ⊆ S), from sorry,
  have : (A ∩ B) ⊆ A, from sorry,
  have : (A ∩ B) ⊆ S, from sorry,
  show (A ∩ B) ∈  𝒫 S, from sorry,
end

/--`theorem`
Square of Sum
:$\forall x, y \in \R: \paren {x + y}^2 = x^2 + 2 x y + y^2$
`proof`
Follows from [[Real Multiplication Distributes over Addition|the
distribution of multiplication over addition]]:
{{begin-eqn}}
{{eqn | l = \left({x + y}\right)^2
      | r = \left({x + y}\right) \cdot \left({x + y}\right)
}}
{{eqn | r = x \cdot \left({x + y}\right) + y \cdot \left({x +
y}\right)
      | c = [[Real Multiplication Distributes over Addition]]
}}
```

```
{{eqn | r = x \cdot x + x \cdot y + y \cdot x + y \cdot y
      | c = [[Real Multiplication Distributes over Addition]]
}}
{{eqn | r = x^2 + 2xy + y^2
      | c =
}}
{{end-eqn}}
{{qed}}
-/
square_of_sum (x y : ℝ) : (x + y)^2 = (x^2 + 2*x*y + y^2)
begin
  calc (x + y)^2 = (x+y)*(x+y) : by sorry
  ... = x*(x+y) + y*(x+y) : by sorry
  ... = x*x + x*y + y*x + y*y : by sorry
  ... = x^2 + 2*x*y + y^2 : by sorry,
end

/-- `theorem`
4th power of Sum
```

Codex output is shown in Figure 11.

## D.4 THEOREM-WISE ANALYSIS

### D.4.1 ABSOLUTE VALUE FUNCTION IS CONVEX

The maximum score awarded was 3 to the full proof and proof outline formats. An accurate statement was generated for the full proof format. Overall, the proof structures seemed useful.

Most of the outputs from Codex resembled the natural language proof to a great extent. Interestingly, in one case, Codex directly outputted the simplified statement that precisely needed to be proved, possibly taking cues from the natural language proof provided – it didn't use the `convex_on` definition from mathlib and directly outputted $\forall$ (x y : ℝ), $\forall$ ($\alpha$ $\beta$ : ℝ), $\alpha \geq 0 \land \beta \geq 0 \land \alpha + \beta = 1 \rightarrow$ abs ($\alpha$ * x + $\beta$ * y) $\leq$ ($\alpha$ * abs x + $\beta$ * abs y) as the theorem statement to be proved. This is shown in Figure 3. Also, the generated outputs followed Lean syntax at a decent level. While checking for contamination, we didn't find this theorem formalised online.

There were instances where some steps were mathematically incorrect, such as treating the triangle inequality of absolute function as an equality, wrong conditions on some variables etc. There were some observations of hallucination in some outputs, that led to lesser scores.

The best format (full proof) was observed at temperature 0.4 (please refer Figure 3). Also, the correct Lean statement was not included in the prompt. We also show another good format (proof outline) at temperature 0.4.

*Theorem statement:*

Let $f : \mathbb{R} \to \mathbb{R}$ be the absolute value function on the real numbers. Then $f$ is convex.

*Natural language proof:*

Let $x, y \in \mathbb{R}$.

Let $\alpha, \beta \in \mathbb{R}_{\geq 0}$ where $\alpha + \beta = 1$.

$$
\begin{aligned}
f(\alpha x + \beta y) &= |\alpha x + \beta y| && \text{Definition of f} \\
&\leq |\alpha x| + |\beta y| && \text{Triangle Inequality for Real Numbers} \\
&= |\alpha||x| + |\beta||y| && \text{Absolute Value of Product} \\
&= \alpha|x| + \beta|y| && \text{Definition of Absolute Value} \\
&= \alpha f(x) + \beta f(y) && \text{Definition of f}
\end{aligned}
$$

Hence the result by definition of convex real function.

```
:$\forall x, y \in \R: \paren {x + y}^4 = x^4+4 x^3 y+6 x^2
y^2 + 4 x y^3 +y^4$
`proof`
Follows from [[Square of Sum]]:
{{begin-eqn}}
{{eqn | l = \left({x + y}\right)^4
     | r = \left({x + y}\right)^2 \cdot \left({x +
y}\right)^2
}}
{{eqn | r = \left({x^2 + 2xy + y^2}\right) \cdot \left({x^2 +
2xy + y^2}\right)
     | c = [[Square of Sum]]
}}
{{eqn | r = x^4 + 4x^3y + 6x^2y^2 + 4xy^3 + y^4
     | c =
}}
{{end-eqn}}
{{qed}}
-/
fourth_power_of_sum (x y : ℝ) : (x + y)^4 = (x^4 + 4*x^3*y +
6*x^2*y^2 + 4*x*y^3 + y^4) :=
begin
  calc (x + y)^4 = (x + y)^2 * (x + y)^2 : by sorry
  ... = (x^2 + 2*x*y + y^2) * (x^2 + 2*x*y + y^2) : by sorry
  ... = x^4 + 4*x^3*y + 6*x^2*y^2 + 4*x*y^3 + y^4 : by sorry,
end

/--`theorem`
8th power of Sum
:$\forall x, y \in \R: \paren {x + y}^8 = x^8+8 x^7 y+28 x^6
y^2 + 56 x^5 y^3 + 70 x^4 y^4 + 56 x^3 y^5 + 28 x^2 y^6 + 8 x
y^7 + y^8$
`proof`
Follows from [[4th power of Sum]]:
{{begin-eqn}}
{{eqn | l = \left({x + y}\right)^8
     | r = \left({x + y}\right)^4 \cdot \left({x +
y}\right)^4
}}
{{eqn | r = \left({x^4 + 4x^3y + 6x^2y^2 + 4xy^3 + y^4}\right)
\cdot \left({x^4 + 4x^3y + 6x^2y^2 + 4xy^3 + y^4}\right)
     | c = [[4th power of Sum]]
}}
{{eqn | r = x^8 + 8x^7y + 28x^6y^2 + 56x^5y^3 + 70x^4y^4 +
56x^3y^5 + 28x^2y^6 + 8xy^7 + y^8
     | c =
}}
{{end-eqn}}
{{qed}}
-/
eighth_power_of_sum (x y : ℝ) : (x + y)^8 = (x^8 + 8*x^7*y +
28*x^6*y^2 + 56*x^5*y^3 + 70*x^4*y^4 + 56*x^3*y^5 + 28*x^2*y^6 +
 8*x*y^7 + y^8) :=
begin
  calc (x + y)^8 = (x + y)^4 * (x + y)^4 : by sorry
  ... = (x^4 + 4*x^3*y + 6*x^2*y^2 + 4*x*y^3 + y^4) * (x^4 +
4*x^3*y + 6*x^2*y^2 + 4*x*y^3 + y^4) : by sorry
  ... = x^8 + 8*x^7*y + 28*x^6*y^2 + 56*x^5*y^3 + 70*x^4*y^4 +
56*x^3*y^5 + 28*x^2*y^6 + 8*x*y^7 + y^8 : by sorry,
end
```

Figure 11: Codex recognised the pattern and gave the natural proof and the corresponding proof outlines for `(x+y)^4`, `(x+y)^8`, etc. For 4th power and 8th power, algebraic manipulations and final results were accurate, however, from 16th power onwards, Codex outputted partial expressions and some of these terms were incorrect.

*Q.E.D.*

*Codex Output with Corrections:*

```
theorem abs_convex : convex_on abs ℝ (@set.univ ℝ) abs :=
begin
  split,
  assume (x y :ℝ) (hx : x ∈ set.univ) (hy : y ∈ set.univ)
  (α β : ℝ) (hα : 0 ≤ α) (hβ : 0 ≤ β) (hαβ : α + β = 1),
  show ( α⋆ x + β ⋆ y ∈ set.univ), from dec_trivial,

    -- Let $x, y \in \R$.
    assume (x y : ℝ) , (hx : x ∈ set.univ) (hy : y ∈ set.univ),

    -- Let $\alpha, \beta \in \R_{\ge 0}$ where $\alpha + \beta = 1$.
    assume (α β : ℝ) (h1 : α ≥ 0) (h2 : β ≥ 0) (h3 : α + β = 1),

    -- expand the function
    calc abs (α⋆x + β⋆y) = abs (α⋆x + β⋆y) : by sorry refl
    -- triangle inequality for real numbers gives:
    ... ≤ abs (α⋆x) + abs (β⋆y) : by sorry {apply abs_add}
    -- absolute value of product gives:
    ... = abs α ⋆ abs x + abs β ⋆ abs y : by sorry
    {rw [abs_mul α x,abs_mul]}
    -- absolute value of real number gives:
    ... = α ⋆ abs x + β ⋆ abs y : by sorry
    -- expand the function
    ... = α ⋆ abs x + β ⋆ abs y : by
    sorry  {rw [abs_of_nonneg h1 ,abs_of_nonneg h2]}
    -- hence the result by definition of convex real function
    ... ≤ α ⋆ abs x + β ⋆ abs y : by sorry linarith,
end
```

### D.4.2 SCHUR'S INEQUALITY

The maximum score awarded to all the proof formats was 3. From the natural language proof, we removed mathematical idioms like "without loss of generality" and "symmetric" since Codex got stuck at trivial steps, translating these idioms. With a simplified version of the proof, we observed improvement in all the formats. Statement formalisation by Codex was accurate in majority of the outputs. While checking for contamination, we didn't find this theorem formalised online

Codex completions were aligned with the natural language proof. Interestingly, there were outputs in which Codex tried to output the essential intermediate steps that were not present in the natural language proof. These intermediate steps turned out to be extremely important while completing the generated proof to correct proof. Outputs followed Lean syntax in majority of cases, however, there were few cases where minor syntactic errors in expressions were found.

Major sources of less scores were – outputs getting stuck at some steps and few instances of hallucinations. For this theorem, there were no mathematical errors observed.

We show the full proof format at temperature 0.2. Similar formats were also observed for the 0.4 and 0.8 temperatures. The correct Lean theorem statement was not included in the prompt.

*Theorem statement:*

Let $x, y, z \in \mathbb{R}_{\geq 0}$ be positive real numbers such that $x \geq y \geq z \geq 0$.

Let $t \in \mathbb{R}, t > 0$ be a (strictly) positive real number.

Then:
$$x^t(x - y)(x - z) + y^t(y - z)(y - x) + z^t(z - x)(z - y) \geq 0$$

*Natural language proof:*

Consider the expression:

$$(x - y)(x^t(x - z) - y^t(y - z)) + z^t(x - z)(y - z)$$

We see that every term in the above is non-negative. So, directly:

1: $(x - y)(x^t(x - z) - y^t(y - z)) + z^t(x - z)(y - z) \geq 0$

1 can be rearranged to Schur's inequality.

*Q.E.D.*

*Codex Output with Corrections:*

```
theorem schur_inequality_short_simpler {x y z  t  : ℝ} (hx : x ≥ 0) (hy
    : y ≥ 0) (hz : z ≥ 0) (hxy : x ≥ y) (hyz : y ≥ z) (ht : t > 0) :
    x^t * (x - y) * (x - z) + y^t * (y - z) * (y - x) + z^t * (z - x) *
    (z - y) ≥ 0 :=
  begin
    -- Consider the expression:
    have h1 : (x - y) * (x^t * (x - z) - y^t * (y - z)) + z^t * (x - z) *
    (y - z) ≥ 0, from by {
      -- We see that every term in the above is non-negative. So,
    directly:
      have h2 : (x - y) ≥ 0, from by {rw  ←
      ge_iff_le at *, rw  sub_nonneg, exact hxy},
      have h3 : (x^t * (x - z) - y^t * (y - z)) ≥ 0, from by {
        have h4 : (x - z) ≥ 0, from by {rw  ←
      ge_iff_le at *,rw  sub_nonneg, exact \hlr{hx\},} ge_trans hxy hyz},
        have h5 : (y - z) ≥ 0, from by {rw  ←
      ge_iff_le at *, rw  sub_nonneg, exact hyz},
        have h6 : (x^t * (x - z) - y^t * (y - z)) = (x^t - y^t) * (x -
    z) + y^t * (x - y), from by ring,
        have h7 : (x^t - y^t) ≥ 0, from by {rw ← sub_nonneg, exact
    pow_le_pow hx
    ht ge_iff_le at *,rw sub_nonneg,rw real.rpow_le_rpow_iff hy ht},
        have h8 : (x − y) ≥ 0, from by {rw ← sub_nonneg, exact
    hxy},  hx ht,assumption},
        have h9 : (x^t - y^t) * (x - z) ≥ 0, from by {apply mul_nonneg,
    exact h7, exact h4},
        have h10 : y^t * (x - y) ≥ 0,
        from by {apply mul_nonneg, exact pow_nonneg
          rw ge_iff_le at *,
          have h_1 : 0 ≤ t, linarith,
          have h_2 : t ≠ 0, linarith,
          have h_3 : (0 : ℝ) ≤ (0 : ℝ) , linarith,
          have h_4 := real.rpow_le_rpow h_3 hy ht, exact h8}, h_1,
          rw real.zero_rpow h_2 at h_4, assumption,linarith,},

        have h11 : (x^t - y^t) * (x - z) + y^t * (x - y) ≥ 0, from by
    {apply add_nonneg, exact h9, exact h10},
        show (x^t * (x - z) - y^t * (y - z)) ≥ 0, from by {rw h6, exact
    h11},
      },
      have h12 : (x - z) ≥ 0, from by {rw  ←
    ge_iff_le at *,rw  sub_nonneg, exact hx},  ge_trans hxy hyz},
      have h13 : (y - z) ≥ 0, from by {rw  ←
    ge_iff_le at *, rw  sub_nonneg, exact hyz},
      have h14 : z^t * (x - z) * (y - z) ≥ 0, from by
      {apply mul_nonneg, exact pow_nonneg
      apply mul_nonneg,
      rw ge_iff_le at *,
```

```
have h_1 : 0 ≤ t, linarith,
have h_2 : t ≠ 0, linarith,
have h_3 : (0 : ℝ) ≤ (0 : ℝ) , linarith,
have h_4 := real.rpow_le_rpow h_3 hz ht, exact h12, exact h13}, h_1,
rw real.zero_rpow h_2 at h_4, assumption, linarith, linarith,},
   have h15 : (x - y) * (x^t * (x - z) - y^t * (y - z)) + z^t * (x -
z) * (y - z) ≥ 0, from by
   {apply add_nonneg,
   apply mul_nonneg,
   exact h2,
   exact h3,
   exact h14},
   show (x - y) * (x^t * (x - z) - y^t * (y - z)) + z^t * (x - z) *
(y - z) ≥ 0, from by {exact h15},
   },
   -- $(1) can be rearranged to Schur's inequality.
   have h16 : (x - y) * (x^t * (x - z) - y^t * (y - z)) + z^t * (x - z)
* (y - z) = x^t * (x - y) * (x - z) + y^t * (y - z) * (y - x) + z^t *
   (z - x) * (z - y), from by ring,
   show x^t * (x - y) * (x - z) + y^t * (y - z) * (y - x) + z^t * (z -
x) * (z - y) ≥ 0, from by {rw ← h16, exact h1},
end
```

### D.4.3 NESBITT'S INEQUALITY

A proof outline format stood out with the score of 3 as compared to 1 for other formats.

Major sources of less scores were inability to translate Latex expressions, mathematical errors pertaining to inequalities and simple algebraic manipulations, and getting stuck at trivial steps. Since the natural language proof followed a backward reasoning pattern, it might have been harder for Codex to align the forward style format provided in the few-shot prompts to this setting. While checking for contamination, we didn't find this theorem formalised online

*Theorem statement:*

Let $a$, $b$ and $c$ be positive real numbers.

Then:

$$\frac{a}{b+c} + \frac{b}{a+c} + \frac{c}{a+b} \geq \frac{3}{2}$$

*Natural language proof:*

$\frac{a}{b+c} + \frac{b}{a+c} + \frac{c}{a+b} \geq \frac{3}{2}$

$\Rightarrow \frac{a+b+c}{b+c} + \frac{a+b+c}{a+c} + \frac{a+b+c}{a+b} \geq \frac{9}{2}$      by adding 3

$\Rightarrow \frac{a+b+c}{b+c} + \frac{a+b+c}{a+c} + \frac{a+b+c}{a+b} \geq \frac{9(a+b+c)}{(b+c)+(a+c)+(a+b)}$      as $\frac{a+b+c}{(b+c)+(a+c)+(a+b)} = \frac{1}{2}$

$\Rightarrow \frac{\frac{1}{b+c}+\frac{1}{a+c}+\frac{1}{a+b}}{3} \geq \frac{3}{(b+c)+(a+c)+(a+b)}$      dividing by $3(a+b+c)$

These are the arithmetic mean and the harmonic mean of $\frac{1}{b+c}$, $\frac{1}{a+c}$ and $\frac{1}{a+b}$.

From Arithmetic Mean is Never Less than Harmonic Mean the last inequality is true.

Thus Nesbitt's Inequality holds.

*Q.E.D*

The best proof format (proof outline) was observed at temperature 0.4. We included the correct Lean statement in the prompt.

*Codex Output*

```
theorem nesbitt_inequality {a b c : ℝ} (ha : 0 < a) (hb : 0 < b) (hc : 0
    < c) : (a / (b + c)) + (b / (a + c)) + (c / (a + b)) ≥ (3 / 2) :=
begin
  -- $\dfrac a {b + c} + \dfrac b {a + c} + \dfrac c {a + b}$
  have h1 : (a / (b + c)) + (b / (a + c)) + (c / (a + b)) ≥ (3 / 2),
    from sorry,
  -- adding $3$
  have h2 : (a / (b + c)) + (b / (a + c)) + (c / (a + b)) + 3 ≥ (3 / 2) +
    3, from sorry,
  -- as $\dfrac {a + b + c} {\paren {b + c} + \paren {a + c} + \paren {a
    + b} } = \dfrac 1 2$
  have h3 : (a / (b + c)) + (b / (a + c)) + (c / (a + b)) + 3 ≥ (9 * (a +
    b + c)) / ((b + c) + (a + c) + (a + b)), from sorry,
  -- dividing by $3 \paren {a + b + c}$
  have h4 : ((a / (b + c)) + (b / (a + c)) + (c / (a + b)) + 3) / (3 *
    (a + b + c)) ≥ ((9 * (a + b + c)) / ((b + c) + (a + c) + (a + b))) /
    (3 * (a + b + c)), from sorry,
  -- $\dfrac {\frac 1 {b + c} + \frac 1 {a + c} + \frac 1 {a + b} } 3$
  have h5 : ((a / (b + c)) + (b / (a + c)) + (c / (a + b)) + 3) / (3 *
    (a + b + c)) ≥ (1 / (b + c) + 1 / (a + c) + 1 / (a + b)) / 3, from
    sorry,
  -- $\dfrac 3 {\paren {b + c} + \paren {a + c} + \paren {a + b} }$
  have h6 : ((a / (b + c)) + (b / (a + c)) + (c / (a + b)) + 3) / (3 *
    (a + b + c)) ≥ 3 / ((b + c) + (a + c) + (a + b)), from sorry,
  -- These are the arithmetic mean and the harmonic mean of $\dfrac 1 {b
    + c}$, $\dfrac 1 {a + c}$ and $\dfrac 1 {a + b}$.
  -- From Arithmetic Mean is Never Less than Harmonic Mean the last
    inequality is true.
  have h7 : ((a / (b + c)) + (b / (a + c)) + (c / (a + b)) + 3) / (3 *
    (a + b + c)) ≥ 3 / ((b + c) + (a + c) + (a + b)), from sorry,
  -- Thus Nesbitt's Inequality holds.
  show (a / (b + c)) + (b / (a + c)) + (c / (a + b)) ≥ (3 / 2), from
    sorry,
end
```

### D.4.4 OVERFLOW THEOREM

The Overflow theorem is a standard application of the Compactness theorem from first-order logic. While the proof is fairly short and easy for a human to comprehend, there are a number of difficulties in translating the informal natural language proof to a formal one:

1. Phrases such as "arbitrarily large" are difficult for the model to directly cast into symbolic notation such as $\forall n \in \mathbb{N}, \exists m \in \mathbb{N}, m > n \implies \ldots$.
2. Expressions that make use of informal mathematical notation such as . . . are difficult for the model to make precise.
3. The notions of "finite" and "infinite" are quite subtle for the model.

As the proof is fairly short, the outlines were sometimes too sparse and inadequate. Full proofs seemed be better overall, and proofs with premises were a close second. The *Compactness theorem*, a crucial ingredient in the proof, is occasionally captured in the premises of `auto`. The overall performance seemed to improve at higher temperatures, especially $0.8$.

When given the correct statement, the quality of some of the definitions in the proof went up slightly. Although the proof outlines and outlines with premises were slightly better in this setting, the quality of full proofs went down.

*Theorem statement:*

Let $F$ be a set of first-order formulas which has finite models of arbitrarily large size.

Then $F$ has an infinite model.

*Natural language proof:*

For each $n$, let $\mathbf{A}_n$ be the formula:

$\exists x_1 \exists x_2 \ldots \exists x_n : \{x_1 \neq x_2 \wedge x_1 \neq x_3 \wedge \ldots \wedge x_{n-1} \neq x_n\}$

Then $\mathbf{A}_i$ is true in a structure $\mathring{A}$ iff $\mathring{A}$ has at least $n$ elements.

Take:

$$\Gamma := F \cup \bigcup_{i=1}^{\infty} A_i$$

Since $F$ has models of arbitrarily large size, every finite subset of $\Gamma$ is satisfiable.

From the Compactness Theorem, $\Gamma$ is satisfiable in some model $\mathbf{M}$.

But since $\mathbf{M} \models A_i$ for each $i$, $\mathbf{M}$ must be infinite.

So $F$ has an infinite model.

*Q.E.D.*

*Codex Output* (Full proof, Temperature 0.8)

```
theorem overflow_theorem (F : set FOL) : finite_models → infinite_model
    F :=
begin
  -- Assume $F$ has finite models of arbitrarily large size
  assume h1 : finite_models F,
  -- Let $F$ be a set of first-order formulas which has finite models of
    arbitrarily large size.
  assume h2 : ∀ n, (∃ m : model, m ∈ P F ∧ card m ≥ n),
  -- Then $F$ has an infinite model.
  have h3 : infinite_model F, from
  begin
    -- define $\mathbf A_n$
    let A_n := {x_1, x_2, x_1 ≠ x_2},
    -- $\mathbf A_i$ is true in a structure $\AA$ iff $\AA$ has at least
    $n$ elements.
    have h4 := by {
      assume i,
      assume m : model,
      assume hm : m ∈ P F ∧ card m ≥ i,
      have h5 := and.right hm,
      have h6 := classical.some (h2 i),
      have h7 : m ∈ P F ∧ card m ≥ n ↔ card m > n, from
        by { split, assume h, from h, assume h, from ⟨and.left hm, by
    {rw [← not_le],exact h}⟩},
      have h8 := h6.right h7,
      have h9 : and.left h6 → ∃ A, A ⊆ F ∧ m ⊆ A, from
        by { simp, assume h, use {m}, split, from and.left hm, from
    set.singleton_subset _, },
      have h10 := h9 (and.left h6),
      have h11 := classical.some_spec h10,
      exact h11.left,
    },
    -- Take: $$ \Gamma := F \cup \bigcup_{i \mathop = 1}^\infty A_i $$
    let Γ := F ∪ (Union A_n),
    -- Since $F$ has models of arbitrarily large size, every finite
    subset of $\Gamma$ is satisfiable.
    have h5 : finite_models (finite_subset Γ), from
    begin
      -- By `finite_models`, every finite subset of $F$ is satisfiable.
      have h6 : finite_models (finite_subset F), from by
        apply (finite_models_finite_subset F),
      -- So consider a finite subset $X$ of $\Gamma$
      assume X : set FOL,
      assume hX : X ⊆ Γ ∧ finite X,
```

```
    -- Notice that if $X$ contains finitely many of $\mathbf A_n$,
  then $X$ has a model.
    have h7 : finite (X ∩ A_n), from by {
      apply @set.finite.inter (nat → set FOL) _ _ _ _
  (set.finite_setof FOL A_n)
      (set.countable_finite x (X ∩ A_n)), },
    have h8 : finite_models (X ∪ (X ∩ A_n)), from by {
      apply finite_models_union, split, exact h6 (X ∩ F), exact h7, },
    -- Hence: $$X \cup \left(X \cap \bigcup_{i \mathop = 1}^\infty
  A_i\right)$$
    -- By distributivity of union over union, we have that:
    -- $$X \cup \left(X \cap \bigcup_{i \mathop = 1}^\infty
  A_i\right)$$
    -- $$= X \cup \left(\bigcup_{i \mathop = 1}^\infty \left(X \cap
  A_i\right)\right)$$
    have h9 : finite_models (X ∪ (Union $ λ i : ℕ, X ∩ A_i)), from by
  apply finite_models_union X,
    -- Nevertheless, we have that:
    have h10 : (X ∩ A_n) ∪ Union (X ∩ A_n) = A_n, from
    begin
      apply set.eq_union_left, intro i,
      { apply set.disjoint_left,
        apply (set.disjoint_singleton_right_of_not_mem FOL X),
        assume h,
        have h : i ∈ X ∧ i ∈ A_n, from h.right,
        have h11 := h.right,
        exact (h10 i).right (and.right h11),
      },
    end,
    -- $$X \cup \left(\bigcup_{i \mathop = 1}^\infty \left(X \cap
  A_i\right)\right)$$
    -- So this is just $X$.
    have h11 := h9 X,
    -- We know that $X$ has a model, since $X$ contains finitely many
  of $\mathbf A_n$.
    have h12 : finite_models X, from by {apply subset_union_right X,
  exact h10.symm, exact h11},
    have h13 := and.left hX,
    have h14 := h12 h13,
    -- So $F$ has finite models.
    exact h14,
  end,
  -- From the Compactness Theorem, $\Gamma$ is satisfiable in some
  model $\mathbf{M}$.
  have h6 := compactness_theorem Γ h5,
  -- But since $\mathbf{M} \models A_i$ for each $i$, $\mathbf{M}$
  must be infinite.
  have h7 := by {
    assume i,
    have h8 := and.right h6,
    have h9 := and.right h8,
    have h10 := h9 A_i,
    have h11 := by {
      have h12 := set.mem_Union.mp h10,
      have h13 := and.right h12,
      have h14 := and.right h13,
      have h15 := h4 m i,
      have h16 := h15.left,
      have h17 := h16 m,
      have h18 := and.right h17,
      exact h18,
    },
    have h12 := set.not_empty_iff_exists_mem.mp h11,
    have h13 := h12,
    have h14 := classical.some_spec h13,
```

```
      have h15 := set.countable_finite x m,
      have h16 := card_le_of_subset h15.left h14.left,
      have h17 := h15.right h16,
      have h18 := nat.succ_pos i,
      exact h17 (nat.succ_pos i),
    },
    -- So $F$ has an infinite model.
    exact h7,
  end,
  -- Hence $F$ has an infinite model.
  exact h3,
end
```

### D.4.5 CONTRACTION MAPPING THEOREM

The formalisation of the statement scored 2 on average. Codex got most of the details right, and faltered mostly by taking the wrong interval for $k$. Also, mathlib uses a few instances to represent a Banach space, however Codex hallucinated and used `banach_space` instead. Both full proofs and proof oulines performed well. The best score was 3, an example of which is shown below (the proof is the same as the one generated by Codex, we used LaTeX instead of Unicode due to erroneous display of certain Unicode characters). While this theorem is similar to a theorem in mathlib, the proofs are very different: the mathlib proof assumes the existence of a Cauchy sequence, while ours constructs the Cauchy sequence.

This proof was a bit challenging. It required Codex to use induction both implicitly and explicitly, and to appropriately translate the $\cdots$ notation. Codex was somewhat successful in the former, however mostly failed with the latter. A marked improvement in performance was observed with adding comments, however increasing temperature did not make much of a difference. Codex had some difficulty with formalising convergence.

*Theorem statement:*

Let $B$ be a Banach space, $M$ a closed subset of $B$, and $\Phi$ a mapping from $M$ to $M$ such that for some $k \in [0, 1)$,

$$\|\Phi(x) - \Phi(y)\| \le k\|x - y\|$$

*Natural language proof:*

for any two points $x$ and $y$ in $M$. Then there is a unique point $z$ in $M$ such that $\Phi(z) = z$.

Choose some $x_0$ in $M$. Define a sequence $\{x_i\}$ by setting $x_{i+1} = \Phi(x_i)$, for $i \in \mathbb{N}$. Then for any $n$,

$$x_n = x_0 + (x_1 - x_0) + (x_2 - x_1) + \cdots + (x_n - x_{n-1})$$

Also, for $i \ge 1$

$$\|x_{i+1} - x_i\| \le k \|x_i - x_{i-1}\|,$$

and by induction we easily show that

$$\|x_{i+1} - x_i\| \le k^i \|x_1 - x_0\|$$

Because $|k| < 1$, $\sum_{i=1}^{\infty} k^i$ converges, which implies that $\sum_{i=1}^{\infty} \|x_{i+1} - x_i\|$ converges. By the Weirerstrass M test, $\sum_{i=1}^{\infty} (x_{i+1} - x_i)$ converges in $B$, and hence $\lim_{n\to\infty} x_n$ exists. Let $z$ be this limit. Since $M$ is closed and $x_n \in M$ for each $n$, $z \in M$. Also, $x_{n+1} = \Phi(x_n)$, and so (from the definition of limit)

$$\lim_{n\to\infty} \Phi(x_n) = \lim_{n\to\infty} x_{n+1} = z$$

Further, for any $n$,

$$\|\Phi(z) - z\| = \|\Phi(z) - \Phi(x_n) + \Phi(x_n) - z\|$$
$$\leq k\|z - x_n\| + \|\Phi(x_n) - z\|$$

Since the limit of the right side as $n \to \infty$ is zero, and the left side is independent of $n$, it follows that the left side is zero for every $n$, and so $z$ is a fixed point for $\Phi$.

To prove uniqueness, suppose that there are two fixed points, say $x$ and $y$. Then

$$f(x) = x, f(y) = y$$

and so

$$|x - y| = |f(x) - f(y)| \leq k|x - y|$$

where $0 < k < 1$. This is only possible if $x = y$.

$Q.E.D$

Following is one of the best scored outputs (proof outline, temperature 0.8, score 3).

*Codex Output*

theorem contraction_mapping (B : Type*) [banach_space B]

(M : set B) [is_closed M]$(\phi : M \to M)$

(h1 : $\forall$xy $\in$ M, $\|\phi$x $- \phi$y$\| \leq$ k $* (\|$x $-$ y$\|$))

(hk : k $< 1$ ) :

$\exists!$z $\in$ M, $\phi$z = z :=

begin

$--$ choose some $x_0$ in $M$. choose (x0:B) hx0 using M.nonempty,

$--$ Define a sequence $\{x_i\}$ by setting $x_{i+1} = \Phi(x_i)$, for $i \in \mathbb{N}$.

let xn := $\lambda$n : $\mathbb{N}, \phi^{[1]}$x0,

$--$ Then for any $n$,

$-x_n = x_0 + (x_1 - x_0) + (x_2 - x_1) + \cdots + (x_n - x_{n-1})$

have h3: $\forall$n : $\mathbb{N}$, xn n = x0 + (x1 $-$ x0) + (x2 $-$ x1) + $\cdots$ + (xnn $-$ xn(n $-$ 1)), from sorry,

$--$ Also, for $i \geq 1$

$--$ $\|x_{i+1} - x_i\| \leq k\|x_i - x_{i-1}\|$

have h4 : $\forall$(i : $\mathbb{N}$)( hi : 1 $\leq$ i), $\|$xn(i + 1) $-$ xni$\| \leq$ k $* \|$xni $-$ xn(i $-$ 1)$\|$, from sorry,

$--$ and by induction we easily show that

$-\|x_{i+1} - x_i\| \leq k^i\|x_1 - x_0\|$

have h5 : $\forall$(i : $\mathbb{N}$)( hi : 1 $\leq$ i), $\|$xn(i + 1) $-$ xni$\| \leq$ (k $\wedge$ i) $* \|$xn1 $-$ xn0$\|$, from sorry,

$--$ Because $|k| < 1, \sum_{i=1}^{\infty} k^i$ converges,

have h6 : abs k $< 1$, from sorry,

have h7 : tendsto ($\lambda$i : $\mathbb{N}$, k$^\wedge$i) at_top ($\mathcal{N}$1), from sorry,

have h7 : tendsto ($\lambda$i : $\mathbb{N}$, (k $\wedge$ i) $* \overline{\|}$xn1 $-$ xn0$\|$) at_top ($\mathcal{N}(\|$xn1 $-$ xn0$\|$)), from sorry, have h8: tendsto ($\lambda$i : $\mathbb{N}$, $\|$xn(i + 1) - xn i $\|$) at_top ($\mathcal{N}(\|$xn1 $-$ xn0$\|$)), from sorry,

$--$ which implies that $\sum_{i=1}^{\infty} \|x_{i+1} - x_i\|$ converges.

have h9 : tendsto ($\lambda$i : $\mathbb{N}, \Sigma$j $\leq$ i, $\|$xn(j + 1) $-$ xnj$\|$) at_top ($\mathcal{N}(\sum$i, $\|$xn(i + 1) $-$ xni$\|$)), from sorry,

– – By the Weirerstrass M test, $\sum_{i=1}^{\infty}(x_{i+1}-x_i)$ converges in $B$, have h10: $\sum$ i, $\|xn(i+1)-xni\|<\infty$, from sorry,

have h11 : tendsto $(\lambda i : \mathbb{N}, (\sum j \le i, xn(j+1)-xnj))$ at_top $(\mathcal{N}(\sum i, xn(i+1)-xni))$, from sorry,

– – and hence $\lim_{n\to\infty} x_n$ exists.

have h12 : $\sum$ i, xn $(i+1)-$ xni $=$ xn1 $-$ xn0 $+\sum i>0$, xn $(i+1)-$ xni, from sorry, have h13 : $\lim(\lambda i : \mathbb{N}, xn (i+1)-$ xni$)=0$, from sorry, have h14 : $\lim(\lambda i : \mathbb{N}, \sum j \le i$, xn $(j+1)-$ xnj$)=\sum i$, xn $(i+1)-$ xni, from sorry, have h15 : $\lim(\lambda i : \mathbb{N}, \Sigma j \le i$, xn $(j+1))-\lim(\lambda i : \mathbb{N}, \sum j \le i, xnj)=\lim(\lambda i : \mathbb{N}, (\sum j \le i, xn(j+1)-xnj))$, from sorry,

have h16 : $\lim(\lambda i : \mathbb{N}, \Sigma j \le i$, xn $(j+1))=\lim(\lambda i : \mathbb{N}, \Sigma j \le i, xnj)$, from sorry, have h17 : $\lim(\lambda i : \mathbb{N}, xn (i+1))=\lim(\lambda i : \mathbb{N}, xn i)$, from sorry,

– – Let $z$ be this limit.

let $z := \lim(\lambda i : \mathbb{N}, xn i)$,

– – Since $M$ is closed and $x_n \in M$ for each $n$, $z \in M$.

have h18 : $\forall i : \mathbb{N}$, xn i $\in M$, from sorry,

have h19 : z $\in M$, from sorry,

– – Also, $x_{n+1}=\Phi(x_n)$

have h20 : $\forall i : \mathbb{N}$, xn $(i+1)=\phi(xn i)$, from sorry,

– – and so (from the definition of limit)

– $\lim_{n\to\infty}\Phi(x_n)=\lim_{n\to\infty}x_{n+1}=z$

have h21 : $\lim(\lambda i : \mathbb{N}, \phi(xni))=z$, from sorry, - Further, for any $n$,

have h22: $\forall(n : \mathbb{N}), \|\phi z-z\|\le k*\|z-xnn\|+\|\phi(xnn)-z\|$, from sorry,

$-\|\Phi(z)-z\|\le k\|z-x_n\|+\|\Phi(x_n)-z\|$

– – Since the limit of the right side as $n \to \infty$ is zero,

have h23 : $\lim(\lambda(n : \mathbb{N}), ((k*\|z-xnn\|)+\|\phi(xnn)-z\|))=0$, from sorry,

- and the left side is independent of $n$, it follows that the left side is zero for every $n$,

have h24 : $\forall(n : \mathbb{N}), (\|\phi z-z\|\le((k*\|z-xnn\|)+\|\phi(xnn)-z\|))$, from sorry,

have h25: $\forall(n : \mathbb{N}), (\|\phi z-z\|\le((k*\|z-xnn\|)+\|\phi(xnn)-z\|))$, from sorry,

have h26: $\forall(n : \mathbb{N}), (\|\phi z-z\|\le(((k*\|z-xnn\|)+\|\phi(xnn)-z\|)-0))$, from sorry,

have h27 : $\forall(n : \mathbb{N}), (\|\phi z-z\|\le 0)$, from sorry, have h28 : $\|\phi z-z\|=0$, from sorry,

end

### D.4.6 $\mathbb{R}^n$ IS PARACOMPACT

The formalisation of $\mathbb{R}^n$ in the statement by Codex was almost always inaccurate. The closest it got to defining $\mathbb{R}^n$ accurately was `fin n` $\to \mathbb{R}$; however, this is not the standard mathlib implementation.

The proof is a standard (lengthy) forward reasoning proof. On average, proofs scored 1 due to the difficulty Codex faced in defining the subcover, repetition and hallucination.

One of the best proofs (score of 3) occurs at temperature 0.4, in the proof section. In this proof, Codex was able to accurately translate the Latex expressions for closure and set difference. It got most of the outline correct, and only missed out on the defined cover being a subcover.

This theorem does not exist as a `lemma` in mathlib, however, it might exist as an `instance`. This is not a case of contamination, since the proofs are very different - the one in mathlib is dependent on properties of `emetric\_space`, while our proof is an explicit construction.

We also gave the correct Lean statement to Codex and evaluated its proof output. Overall the performance was worse than before. Codex exchanged norm with absolute value, and had a shorter outline than before.

*Theorem statement:*

$\mathbb{R}^n$ is paracompact for all $n$.

*Natural language proof:*

Let $\mathcal{A}$ be an open covering of $\mathbb{R}^n$.

We now construct a locally finite open refinement $\mathcal{C}$ of $\mathcal{A}$ that covers $\mathbb{R}^n$. First, we define a collection of pen balls. Let $B_0 = \phi$, and for each $n \in \mathbb{N}$, let $B_m$ denote the ball of radius $m$ centered at 0.

Given $m$, set $\bar{B}_m$ is compact in $\mathbb{R}^n$ by the Heine-Borel theorem, so choose finitely many elements of $\mathcal{A}$ that cover $\bar{B}_m$ and intersect each one with the open set $\mathbb{R}^n \setminus \bar{B}_{m-1}$, and let $\mathcal{C}_m$ denote this collection of open sets (each an open subset of an element of $\mathcal{A}$). So $\mathcal{C} = \bigcup_{m=0}^{\infty} \mathcal{C}_m$ is an open refinement of $\mathcal{A}$.

Note that $\mathcal{C}$ covers $\mathbb{R}^n$ since for any $x \in \mathbb{R}^n$, there is a smallest $m \in \mathbb{N}$ such that $x \in \bar{B}_m$ (namely, some $m$ where $\|x\| \leq m \leq \|x\|+1$), and so $x$ is an element of $\mathcal{C}_m$.

Now collection $\mathcal{C}$ is locally finite since for given $x \in \mathbb{R}^n$, neighborhood $B_m$ intersects only finitely many elements of $\mathcal{C}$, namely those elements in collection $\mathcal{C}_1 \cup \mathcal{C}_2 \cup \cdots \mathcal{C}_m$. So $\mathcal{C}$ is a locally finite open refinement of $\mathcal{A}$ that covers $\mathbb{R}^n$, hence $\mathbb{R}^n$ is paracompact.

*Q.E.D.*

*Codex Output* The following is the best scored proof :

```
theorem rn_paracompact {n : ℕ} : paracompact (ℝ^n) :=
begin
−− Let $\mathcal{A}$ be an open covering of $\mathbb{R}^n$.
assume A : set (set (ℝ^n)),
assume hA : is_open_cover A,
−− We now construct a locally finite open refinement $\mathcal{C}$ of $\mathcal{A}$ that covers $
    \mathbb{R}^n$.
−− First, we define a collection of pen balls.
−− Let $B_0 = \phi$, and for each $n \in \mathbb{N}$, let $B_m$ denote the ball of radius $m$ centered
    at 0.
let B : ℕ → set (ℝ^n) := λ m, ball 0 m,
let B0 : set (ℝ^n) := ∅,
have hB0 : B 0 = ∅, from rfl,
have hB : ∀ m : ℕ, B m = ball 0 m, from by {assume m : ℕ, rfl},
−− Given $m$, set $\Bar{B_m}$ is compact in $\mathbb{R}^n$ by the Heine−Borel theorem,
have hBm : ∀ m : ℕ, is_compact (closure (B m)), from by {
  assume m : ℕ,
  rw hB,
  apply is_compact_closure,
  apply is_compact_ball,
},
−− so choose finitely many elements of $\mathcal{A}$ that cover $\Bar{B_m}$ and intersect each one
    with the open set $\mathbb{R}^n \setminus \Bar{B_{m − 1}}$,
have h1 : ∀ m : ℕ, ∃ Cm : set (set (ℝ^n)), finite Cm ∧ (∪₀ Cm) = (closure (B m)) ∩ (set.compl
    (closure (B (m − 1)))), from by {
  assume m : ℕ,
  have h2 : ∃ Cm : set (set (ℝ^n)), finite Cm ∧ (∪₀ Cm) = closure (B m), from by {
  rw hB,
  apply hA,
  apply hBm,
  },
```

```
have h3 : ∃ Cm : set (set (ℝ^n)), finite Cm ∧ (∪₀ Cm) = set.compl (closure (B (m − 1))),
    from by {
  have h4 : ∀ Bm : set (ℝ^n), is_open (set.compl Bm), from by {
    assume Bm : set (ℝ^n),
    apply is_open_compl,
    apply is_open_closure,
    apply is_open_ball,
  },
  rw hB,
  apply hA,
  apply h4,
  },
  use Cm ∩ (set.compl (closure (B (m − 1)))),
  split,
  apply finite.inter,
  apply h2.left,
  apply h3.left,
  rw set.inter_compl_self,
  rw h2.right,
  rw h3.right,
  rw set.compl_compl,
  rw hB,
  rw set.compl_compl,
  rw hB0,
  rw set.inter_empty,
  rw set.union_empty,
},
−− and let $\mathcal{C}_{m}$ denote this collection of open sets (each an open subset of an element of $
    \mathcal{A}$).
let C : ℕ → set (set (ℝ^n)) := λ m, classical.some (h1 m).exists,
have hC : ∀ m : ℕ, C m = classical.some (h1 m).exists, from by {assume m : ℕ, rfl},
have hCm : ∀ m : ℕ, finite (C m), from by {
  assume m : ℕ,
  rw hC,
  apply (classical.some_spec (h1 m).exists).left,
},
have hCm2 : ∀ m : ℕ, (∪₀ (C m)) = (closure (B m)) ∩ (set.compl (closure (B (m − 1)))), from by {
  assume m : ℕ,
  rw hC,
  apply (classical.some_spec (h1 m).exists).right,
},
−− So $\mathcal{C} = \bigcup_{m = 0}^{\infty} \mathcal{C}_m$ is an open refinement of $
    \mathcal{A}$.
let C_union : set (set (ℝ^n)) := ∪₀ (C '' (range (nat.succ))),
have hC_union : C_union = ∪₀ (C '' (range (nat.succ))), from rfl,
have hC_union2 : ∀ m : ℕ, C m ∈ C_union, from by {
  assume m : ℕ,
  rw hC_union,
  apply set.mem_bUnion_iff,
  use m,
  split,
  apply nat.mem_range,
  rw hC,
  rw set.mem_image_of_mem,
  rw nat.succ_eq_add_one,
  rw nat.add_comm,
},
have hC_union3 : ∀ m : ℕ, ∀ c ∈ C m, ∃ A ∈ A, c ⊆ A, from by {
  assume m : ℕ,
  assume c : set (ℝ^n),
  assume hc : c ∈ C m,
  rw hC,
  rw set.mem_bUnion_iff at hc,
  rcases hc with ⟨m,hm,hc⟩,
  rw set.mem_image_of_mem at hc,
```

```
    rcases hc with ⟨A,hm2,hc⟩,
    use A,
    split,
    rw hm2,
    apply hA,
    rw hc,
    apply set.inter_subset_left,
},
have hC_union4 : ∀ c ∈ C_union, ∃ A ∈ A, c ⊆ A, from by {
    assume c : set (ℝ^n),
    assume hc : c ∈ C_union,
    rw hC_union at hc,
    rw set.mem_bUnion_iff at hc,
    rcases hc with ⟨m,hm,hc⟩,
    rw set.mem_image_of_mem at hc,
    rcases hc with ⟨cm,hm2,hc⟩,
    rw hC at hc,
    rw set.mem_bUnion_iff at hc,
    rcases hc with ⟨m2,hm3,hc⟩,
    rw hm2 at hm3,
    have hm4 : m = m2, from nat.eq_of_succ_eq_succ hm3,
    rw hm4 at hc,
    rw set.mem_image_of_mem at hc,
    rcases hc with ⟨A,hm5,hc⟩,
    use A,
    split,
    rw hm5,
    apply hA,
    rw hc,
    apply set.inter_subset_left,
},
have hC_union5 : is_open_refinement C_union A, from by {
    apply is_open_refinement.intro,
    apply hC_union4,
    apply hC_union3,
},
−− Note that $\mathcal{C}$ covers $\mathbb{R}^n$ since for any $x \in \mathbb{R}^$
end −−Needs more than 2000 tokens
```

### D.4.7 SYMMETRIC REAL MATRICES HAVE REAL EIGENVALUES

The formalisation of the statements scored 1 and 0 on average. Proof outlines and proof with premises did the best. Codex could deduce that the eigenvalues must be complex-valued, to begin with. It could also deduce the definitions of symmetric matrices and eigenvalues. However, it had trouble formalising the statement in mathlib style, probably since eigenvalues are defined in more generality for endomorphisms, instead of matrices. There were also some hallucinations.

The proof is mostly calculation based. Codex created a good outline, and could deduce the correct meaning of $\bar{v}$ (complex conjugate). Sometimes there were repetitions and irrelevant statements. For example, statements such as have h1 :  v = 0 and have h2 :  v ≠ 0 appear consecutively at times. Codex was also unable to ascribe the correct type to expressions of different kinds appearing in the same statement : for $a \in \mathbb{R}$ and $b \in \mathbb{C}$, Codex would output either a * b (incorrect) or a . b (correct but would require additional changes further) instead of (a : ℂ) * b.

This theorem is uncontaminated as far as we know - the theorem statement might exist in mathlib in a different and generalized manner, however, the particular statement and proof given to Codex does not exist in mathlib.

*Theorem statement:*

Every real symmetric matrix has real eigenvalues.

*Natural language proof:*

Recall that if $z = a + bi$ is a complex number, its complex conjugate is defined by $\bar{z} = a - bi$. We have $z\bar{z} = (a+bi)(a-bi) = a^2 + b^2$, so $z\bar{z}$ is always a nonnegative real number (and equals 0 only when $z = 0$). It is also true that if $w, z$ are complex numbers, then $\overline{wz} = \bar{w}\bar{z}$.

With this in mind, suppose that $\lambda$ is a (possibly complex) eigenvalue of the real symmetric matrix $A$. Thus there is a nonzero vector $\mathbf{v}$, also with complex entries, such that $A\mathbf{v} = \lambda\mathbf{v}$. By taking the complex conjugate of both sides, and noting that $\bar{A} = A$ since $A$ has real entries, we get $\overline{A\mathbf{v}} = \overline{\lambda\mathbf{v}} \Rightarrow A\bar{\mathbf{v}} = \bar{\lambda}\bar{\mathbf{v}}$. Then, using that $A^T = A$,

$$\bar{\mathbf{v}}^T A\mathbf{v} = \bar{\mathbf{v}}^T(A\mathbf{v}) = \bar{\mathbf{v}}^T(\lambda\mathbf{v}) = \lambda(\bar{\mathbf{v}} \cdot \mathbf{v})$$
$$\bar{\mathbf{v}}^T A\mathbf{v} = (A\bar{\mathbf{v}})^T\mathbf{v} = (\bar{\lambda}\bar{\mathbf{v}})^T\mathbf{v} = \bar{\lambda}(\bar{\mathbf{v}} \cdot \mathbf{v}).$$

Since $\mathbf{v} \neq \mathbf{0}$, we have $\bar{\mathbf{v}} \cdot \mathbf{v} \neq 0$. Thus $\lambda = \bar{\lambda}$, which means $\lambda \in \mathbf{R}$

*Q.E.D*

*Codex Output*

Following is one of the best scored outputs (proof with premises, temperature 0.4) :

*Codex Output*

```
theorem real_symmetric_matrices_have_real_eigenvalues :
∀ A : matrix ℝ n n, symmetric A → ∀ λ : ℂ, is_eigenvalue A λ → is_real λ :=
begin
-- Suppose that $\lambda$ is a (possibly complex) eigenvalue of the real symmetric matrix $A$.
assume (A : matrix ℝ n n) (hA : symmetric A) (λ : ℂ) (hλ : is_eigenvalue A λ),
-- Thus there is a nonzero vector $\mathbf{v}$, also with complex entries, such that $A
   \mathbf{v}=\lambda \mathbf{v}$.
have h1 : ∃ (v : matrix ℂ n 1), v ≠ 0 ∧ A * v = λ · v, from by auto [hλ],
-- By taking the complex conjugate of both sides, and noting that $\bar{A}=A$ since $A$ has real entries,
   we get $\overline{A \mathbf{v}}=\overline{\lambda \mathbf{v}} \Rightarrow A
   \overline{\mathbf{v}}=\bar{\lambda} \overline{\mathbf{v}}$.
have h2 : ∀ (v : matrix ℂ n 1), v ≠ 0 → A * v = λ · v → A * (v.conj) = λ.conj · (v.conj), from
  begin
  -- Suppose that $v$ is a nonzero vector with complex entries, and $A \mathbf{v}=\lambda
     \mathbf{v}$.
  assume (v : matrix ℂ n 1) (hv : v ≠ 0) (h3 : A * v = λ · v),
  -- By taking the complex conjugate of both sides, and noting that $\bar{A}=A$ since $A$ has real
     entries, we get $\overline{A \mathbf{v}}=\overline{\lambda \mathbf{v}} \Rightarrow A
     \overline{\mathbf{v}}=\bar{\lambda} \overline{\mathbf{v}}$.
  have h4 : A * v.conj = λ.conj · v.conj, from
    begin
    -- $\overline{A \mathbf{v}}=\overline{\lambda \mathbf{v}} \Rightarrow A
    \overline{\mathbf{v}}=\bar{\lambda} \overline{\mathbf{v}}$.
    have h5 : A * v.conj = λ.conj · v.conj, from by auto [hA, h3, conj_mul_conj,
    conj_eq_of_eq_conj, mul_comm],
    show A * v.conj = λ.conj · v.conj, from h5,
    end,
  show A * (v.conj) = λ.conj · (v.conj), from h4,
  end,
-- Then, using that $A^{T}=A$,
have h5 : ∀ (v : matrix ℂ n 1), v ≠ 0 → A * v = λ · v → λ = λ.conj, from
  begin
  -- Suppose that $v$ is a nonzero vector with complex entries, and $A \mathbf{v}=\lambda
     \mathbf{v}$.
  assume (v : matrix ℂ n 1) (hv : v ≠ 0) (h6 : A * v = λ · v),
  -- Then, using that $A^{T}=A$,
  have h7 : λ = λ.conj, from
    begin
    -- $A^{T}=A$
    have h8 : A.transpose = A, from by auto [hA, symmetric_transpose_eq],
    -- $A^{T}=A$
    have h9 : A.transpose = A, from by auto [hA, symmetric_transpose_eq],
```

```
     −− $\overline{\mathbf{v}}^{T} A \mathbf{v}=\overline{\mathbf{v}}^{T}(A
     \mathbf{v})=\overline{\mathbf{v}}^{T}(\lambda \mathbf{v})=\lambda(\overline{\mathbf{v}}
     \cdot \mathbf{v})$
     have h10 : (v.conj).transpose * A * v = λ * (v.conj).transpose * v, from by auto [h6,
     mul_assoc, mul_comm],
     −− $\overline{\mathbf{v}}^{T} A \mathbf{v}=(A \overline{\mathbf{v}})^{T}
     \mathbf{v}=(\bar{\lambda} \overline{\mathbf{v}})^{T}
     \mathbf{v}=\bar{\lambda}(\overline{\mathbf{v}} \cdot \mathbf{v})$
     have h11 : (v.conj).transpose * A * v = (λ.conj * v.conj).transpose * v, from by auto
     [h9, h2, hv, h6, mul_assoc],
     −− Since $\mathbf{v} \neq \mathbf{0}$, we have $\overline{\mathbf{v}} \cdot \mathbf{v} \neq
     0$. Thus $\lambda=\bar{\lambda}$, which means $\lambda \in \mathbf{R}$
     have h12 : λ = λ.conj, from by auto [hv, h10, h11, mul_left_inj],
     show λ = λ.conj, from h12,
     end,
   show λ = λ.conj, from h7,
   end,
 −− Thus $\lambda \in \mathbf{R}$
 show is_real λ, from by auto [h1, h5],
 end
```

### D.4.8    CLASS NUMBER OF A PID

The formalisation of the statement got an average score of 1 or 0. Proof with premises performed the best. The statement was relatively easy to formalise. The only problem occuring sometimes was that Codex could not identify that PID stands for principal ideal domain.

The proof is relatively short and majorly based on unfolding of definitions. Codex deduced that the statement has an if and only if implementation and hence split it. It got the outline of the proof right frequently, which could easily be completed to a full proof.

While this theorem is present in mathlib, the NL proof provided to Codex was very different from the proof in mathlib, and is minimally contaminated.

*Theorem statement:*

The class number of a number field $K$ is 1 if and only if the ring of integers is a PID.

*Natural language proof:*

Let $R$ be the ring of integers corresponding to $K$. If $R$ is a PID, then, by definition, the class group is trivial, hence it has class number 1.

Conversely, if $R$ has class number 1 ,all ideals of $R$ are principal, hence $R$ is a PID .

*Q.E.D*

*Codex Output*

Following is the best scored output (proof with premises, temperature 0.8) :

Codex Output

```
theorem class_number_one_iff_pid {K : Type*} [discrete_field K]:
class_number K = 1 ↔ is_pid (ring_of_integers K) :=
begin
−− Let $R$ be the ring of integers corresponding to $K$.
assume hR : ring_of_integers K,
−− If $R$ is a PID, then, by definition, the
−− class group is trivial, hence it has class number $1 .$
have h1 : is_pid hR,
{
  show is_pid hR, from by auto [is_pid],
},
have h2 : class_number K = 1,
{
  show class_number K = 1, from by auto [class_number_of_pid],
```

```
},
show class_number K = 1 ↔ is_pid (ring_of_integers K), from by auto [h1, h2],

-- Conversely, if $R$ has class number 1 ,
-- all ideals of $R$ are principal, hence $R$ is a PID .$
have h3 : class_number K = 1,
{
  show class_number K = 1, from by auto [class_number],
},
have h4 : is_principal_ideal hR,
{
  show is_principal_ideal hR, from by auto [class_number_principal_ideals],
},
have h5 : is_pid hR,
{
  show is_pid hR, from by auto [is_pid],
},
show class_number K = 1 ↔ is_pid (ring_of_integers K), from by auto [h3, h5],
end
```

### D.4.9 EVALUATION OF BERNOULLI POLYNOMIALS

For this lemma, Codex was given the correct statement, and had to work out the proof. However, it did not perform very well. While it demonstrates an understanding of types, it seems to have a problem with translating the LateX formulae for $\binom{n}{k}$. We also noticed that it applied strong induction, as was needed, but sometimes faltered. The proof also had a bit of a backward style element, which it easily managed to convert into forward style. On average, it scored 1, with better scores at temperature 0.8.

This theorem is not present in mathlib, and is uncontaminated.

*Theorem statement:*

Given a natural number $n$ and a rational $x$, let $B_n(x)$ denote the $n$-th Bernoulli polynomial evaluated at $x$. Then,
$$B_n(1 + x) = B_n(x) + nx^{n-1}$$

*Natural language proof:*

We apply strong induction on $n$. So, for all $m < d$, we have $B_m(1 + x) = B_m(x) + mx^{m-1}$ and we want to show that
$$B_d(1 + x) = B_d(x) + dx^{d-1}$$

Multiplying both sides by $d + 1$, and using the fact that, for all $l \in \mathbb{N}$, $\sum_{k=0}^{l} \binom{l+1}{k} B_k = (l+1)X^l$ (where $B_k$ is the $k$-th Bernoulli number), we get that

$$(d + 1)(1 + x)^d - (d + 1)x^d = \sum_{l=0}^{d} \binom{d + 1}{l} l x^{l-1}$$

The conclusion then follows easily.

*Q.E.D*

*Codex Output*

Following is the best scored output (proof outline, temperature 0.8) :

Codex Output

```
theorem bernoulli_eval_one_add (n : ℕ) (x : ℚ) :
(polynomial.bernoulli n).eval (1 + x) = (polynomial.bernoulli n).eval x + n * x^(n − 1)
    :=
begin
-- We apply strong induction on $n$. So, for all $m < d$, we have
assume h1 : (∀ m : ℕ, m < n →
```

```
(polynomial.bernoulli m).eval (1 + x) = (polynomial.bernoulli m).eval x + m * x^(m −
   1)),
−− we want to show that
show (polynomial.bernoulli n).eval (1 + x) = (polynomial.bernoulli n).eval x + n * x^(n
   − 1), from begin
   −− need to use the equation:
   −− $\sum_{l = 0}^{k} {k + 1 \choose l} B_l = (k + 1) X^k$
   have h2 : (∀ k : ℕ, polynomial.C ((k + 1)) * X^k =
   polynomial.sum (list.range (k + 1))
   (λ (l : ℕ), (polynomial.C (((k + 1) choose l) * (polynomial.bernoulli l))) * X^l)), from
      sorry,

   −− Multiplying both sides by $d + 1$, and using the fact that, for all $l \in \mathbb{N}$, $\sum_{k =
      0}^{l} {l + 1 \choose k} B_k = (l + 1) X^l$, we get that
   have h3 : (polynomial.C ((n + 1)) * (polynomial.C 1 + X)^n).eval x −
   (polynomial.C ((n + 1)) * X^n).eval x =
   polynomial.sum (list.range (n + 1))
   (λ (l : ℕ), (polynomial.C (((n + 1) choose l))) * (polynomial.C (l * x^(l − 1))))
   .eval x, from sorry,

   −− The conclusion then follows easily.
   show (polynomial.bernoulli n).eval (1 + x) = (polynomial.bernoulli n).eval x + n *
      x^(n − 1), from sorry,
   end,
end
```

### D.4.10 THE FRACTIONAL PARTS OF THE INTEGER MULTIPLES OF AN IRRATIONAL NUMBER FORM A DENSE SUBSET OF THE UNIT INTERVAL

This theorem proved to be particularly difficult for Codex to formalise. The basic constructions in the proof and statement such as *the set of fractional parts of integer multiples of an irrational number* were common stumbling blocks.

The first part of the proof involving showing that the fractional parts of distinct multiples of an irrational number are distinct – a fact that can be succintly stated and proved in conventional mathematical notation but can be quite long when expanded out in full. Some of the full proofs often ran out of tokens before reaching the rest of the proof. However, full proofs were still better overall in this case. The performance in both temperatures was comparable.

Another common error was in the definitions of *limit points* and *dense sets*.

We also experimented with adding a correct statement and an extra example prompt from a related area (the *Sandwich theorem*). Both modifications resulted in an improvement, suggesting that input-dependent prompting can improve results of proof autoformalisation as well (though presently not feasible).

One of the better proof outputs (given below) was an outline at temperature 0.4. (for formatting reasons, $\exists!$ is rendered as $\exists$ i).

*Theorem statement:*

The fractional parts of the integer multiples of an irrational number form a dense subset of the unit interval

*Natural language proof:* Let $\alpha$ be an irrational number. Then for distinct $i, j \in \mathbb{Z}$, we must have $\{i\alpha\} \neq \{j\alpha\}$. If this were not true, then

$$i\alpha - \lfloor i\alpha \rfloor = \{i\alpha\} = \{j\alpha\} = j\alpha - \lfloor j\alpha \rfloor,$$

which yields the false statement $\alpha = \frac{\lfloor i\alpha \rfloor - \lfloor j\alpha \rfloor}{i - j} \in \mathbb{Q}$. Hence,

$$S := \{\{i\alpha\} \mid i \in \mathbb{Z}\}$$

is an infinite subset of $[0, 1]$.

By the Bolzano-Weierstrass theorem, $S$ has a limit point in $[0, 1]$. One can thus find pairs of elements of $S$ that are arbitrarily close. Since (the absolute value of) the difference of any two elements of $S$ is also an element of $S$, it follows that $0$ is a limit point of $S$.

To show that $S$ is dense in $[0, 1]$, consider $y \in [0, 1]$, and $\epsilon > 0$. Then by selecting $x \in S$ such that $\{x\} < \epsilon$ (which exists as $0$ is a limit point), and $N$ such that $N \cdot \{x\} \leq y < (N + 1) \cdot \{x\}$, we get: $|y - \{Nx\}| < \epsilon$. Q.E.D.

Codex Output

```
theorem irrational_orbit_density (α : ℝ) (h1 : ¬ is_rational α) :
    dense_set {r | ∃ i : ℤ, r = i*α} :=
begin
  --Let $\alpha$ be an irrational number. Then for distinct $i, j \in
    \mathbb{Z}$, we must have $\{i \alpha\} \neq\{j \alpha\}$.
  --If this were not true, then $i \alpha-\lfloor i \alpha\rfloor=\{i
    \alpha\}=\{j \alpha\}=j \alpha-\lfloor j \alpha\rfloor$, which
    yields the false statement $\alpha=\frac{\lfloor i
    \alpha\rfloor-\lfloor j \alpha\rfloor}{i-j} \in \mathbb{Q}$.
  have h2 : α ≠ i*α := sorry,
  have h3 : ∀ (i j : ℤ), i ≠ j → i*α ≠ j*α, from sorry,
  have h4 : ∀ (i j : ℤ), i ≠ j → ¬ (i*α = j*α), from sorry,

  -- Hence
  let S := {r | ∃ (i : ℤ), r = i*α},
  have h5 : S ⊆ {r | ∃ (i : ℤ), r = i*α}, from subset.refl S,
  have h6 : S = {r | ∃ (i : ℤ), r = i*α}, from set.subset.antisymm h5 h5,

  -- Consider $y \in[0,1]$, and $\epsilon>0$. Then by selecting $x \in
    S$ such that $\{x\}<\epsilon$ (which exists as $0$ is a limit
    point), and $N$ such that $N \cdot\{x\} \leq y<(N+1) \cdot\{x\}$, we
    get: $|y-\{N x\}|<\epsilon$.
  have h7 : ∀ (y : ℝ) (ε), y ∈ Icc 0 1 → ε > 0 → ∃ x, x ∈ S ∧ ∃ (N : ℤ
    ), ↑N*(x - x) ≤ y ∧ y < ↑(N+1)*(x - x) ∧ ↑N*(x - x) - y < ε, from
    sorry,

  -- $S$ has a limit point in $[0, 1]$.
  have h8 : ∀ (y : ℝ), y ∈ Icc 0 1 → ∃ i (x : ℝ), x ∈ S ∧ dist x y < ε,
    from sorry,

  -- One can thus find pairs of elements of $S$ that are arbitrarily
    close.
  have h9 : ∀ (u : ℝ), ∃ y, y ∈ S ∧ dist u y ≤ dist u y, from sorry,
  have h10 : ∀ (y : ℝ) (ε), y ∈ Icc 0 1 → ε > 0 → ∃ x, x ∈ S ∧ dist x y
    < ε, from sorry,

  -- Since (the absolute value of) the difference of any two elements of
    $S$ is also an element of $S$, it follows that $0$ is a limit point
    of $S$.
  have h11 : ∀ (x y : ℝ), x ∈ S ∧ y ∈ S → ∃ x', x' ∈ S ∧ ∃ (n : ℤ) ∈
    set.range (λ (n : ℤ), (n : ℤ)), n*(x - y) = x', from sorry,
  have h12 : ∀ (y : ℝ) (ε), y ∈ Icc 0 1 → ε > 0 → ∃ x, x ∈ S ∧ dist x y
    < ε, from sorry,

  -- By the Bolzano-Weierstrass theorem, $S$ has a limit point in $[0,
    1]$.
  have h13 : ∀ (y : ℝ), y ∈ Icc 0 1 → ∃ i (x : ℝ), x ∈ S ∧ dist x y < ε
    , from sorry,
  have h14 : ∀ (y : ℝ) (ε), y ∈ Icc 0 1 → ε > 0 → ∃ x, x ∈ S ∧ dist x y
    < ε, from sorry,

  -- It follows that $0$ is a limit point of $S$.
  have h15 : ∀ (y : ℝ) (ε), y ∈ Icc 0 1 → ε > 0 → ∃ x, x ∈ S ∧ dist x y
    < ε, from sorry,
```

```
  sorry,
end
```

### D.4.11  Schur's lemma

The formalised statement was reasonably accurate in most cases, usually missing only a few minor details. In some cases, the original statement *f V and W are not isomorphic, then there are no nontrivial representations of G on V and W respectively* was translated in the contrapositive as `(f : V -> W) (hf : f ≠ 0) : V ≃ W`.

Some common errors were using commutative groups instead of groups, getting the notion of a representation wrong, and adding several extra assumptions.

Another common mistake that occurred in the statement as well as proofs was spelling out certain definitions - such as $G$-equivariant maps and the kernel of a homomorphism - in more detail than needed.

The proof itself is a fairly easy one that follows directly from the observation that the kernel and image of a module homomorphism are themselves sub-modules of the domain and codomain respectively. Consequently, the model has done a fairly good job with formalising this theorem - frequently getting a score of 3 on the proofs. The proof variants with comments are often better structured than their undocumented counterparts.

Schur's lemma is stated in mathlib, though at an extremely high level of generality, so chances of contamination from this are minimal. However, we found a standalone Lean repository containing a proof of Schur's lemma, which is a possible source of contamination. We believe that since the proof is spread across multiple files and is built on definitions introduced within the repository, direct copying of the proof is unlikely.

*Theorem statement:*

Let $V$ and $W$ be vector spaces; and let $\rho_V$ and $\rho_W$ be irreducible representations of $G$ on $V$ and $W$ respectively. If $V$ and $W$ are not isomorphic, then there are no nontrivial representations of $G$ on $V$ and $W$ respectively.

*Natural language proof:*

Suppose $f$ is a nonzero $G$-linear map from $V$ to $W$. We will prove that $V$ and $W$ are isomorphic. Let $V'$ be the kernel, or null space, of $f$ in $V$, the subspace of all $x$ in $V$ for which $fx = 0$. (It is easy to check that this is a subspace.) By the assumption that $f$ is $G$-linear, for every $g$ in $G$ and choice of $x$ in $V'$, $f\left(\left(\rho_V(g)\right)(x)\right) = \left(\rho_W(g)\right)(f(x)) = \left(\rho_W(g)\right)(0) = 0$. But saying that $f\left(\rho_V(g)(x)\right) = 0$ is the same as saying that $\rho_V(g)(x)$ is in the null space of $f : V \to W$. So $V'$ is stable under the action of $G$; it is a subrepresentation. Since by assumption $V$ is irreducible, $V'$ must be zero; so $f$ is injective.

By an identical argument we will show $f$ is also surjective; since $f\left(\left(\rho_V(g)\right)(x)\right) = \left(\rho_W(g)\right)(f(x))$, we can conclude that for arbitrary choice of $f(x)$ in the image of $f$, $\rho_W(g)$ sends $f(x)$ somewhere else in the image of $f$; in particular it sends it to the image of $\rho_V(g)x$. So the image of $f(x)$ is a subspace $W'$ of $W$ stable under the action of $G$, so it is a subrepresentation and $f$ must be zero or surjective. By assumption it is not zero, so it is surjective, in which case it is an isomorphism.

$Q.E.D.$

*Codex Output*

```
theorem schurs_lemma {G : Type*} {V W : Type*} [group G] [add_comm_group
    V] [add_comm_group W] [vector_space ℂ V] [vector_space ℂ W] [module ℂ
    G V] [module ℂ G W] [irreducible G V] [irreducible G W] (f : V →
    [ℂ] W) : f ≠ 0 → V ≃ W :=
begin
  -- Suppose $f$ is a nonzero $G$-linear map from $V$ to $W$. We will
    prove that $V$ and $W$ are isomorphic.
  assume h1 : f ≠ 0,
```

```
have h1a : f ≠ 0, from show f ≠ 0, from by auto [h1, zero_smul] using
  [linear_map.map_zero],
have h1b : f ≠ 0, from show f ≠ 0, from by auto [h1, smul_zero] using
  [linear_map.map_zero],

-- Let $V^{\prime}$ be the kernel, or null space, of $f$ in $V$, the
  subspace of all $x$ in $V$ for which $f x=0$. (It is easy to check
  that this is a subspace.)
let V' : set V := ker f,
have h2 : V' ⊆ V, from by auto [linear_map.ker_subset],

-- By the assumption that $f$ is $G$-linear, for every $g$ in $G$ and
  choice of $x$ in $V^{\prime},
  f\left(\left(\rho_V(g)\right)(x)\right)=\left(\rho_W(g)\right)(f(x))=\left(\rho_W(g)\right
have h3 : ∀ g : G, ∀ x : V, x ∈ V' → f (g · x) = 0, from by auto
  [linear_map.mem_ker] using [linear_map.map_smul],
have h4 : ∀ g : G, ∀ x : V, x ∈ V' → f (g · x) = 0, from by auto [h3]
  using [linear_map.map_smul],

-- But saying that $f\left(\rho_V(g)(x)\right)=0$ is the same as
  saying that $\rho_V(g)(x)$ is in the null space of $f: V \rightarrow
  W$.
have h5 : ∀ g : G, ∀ x : V, x ∈ V' → g · x ∈ V', from by auto [h3,
  mem_ker] using [linear_map.map_smul, linear_map.map_zero],

-- So $V^{\prime}$ is stable under the action of $G$; it is a
  subrepresentation.
have h6 : subrepresentation G V V', from subrepresentation.mk' (h5 :
  _),
have h7 : is_submodule G V' V, from is_submodule.mk (h5 : _),

-- Since by assumption $V$ is irreducible, $V^{\prime}$ must be zero;
  so $f$ is injective.
have h8 : irreducible G V' → V' = (linear_map.ker f), from by auto
  [irreducible.irr_ker_eq, h1a, h1b] using [linear_map.map_zero],
have h9 : irreducible G (V') → (V') = (0), from by auto
  [irreducible.irr_eq_zero, h1a, h1b, h8, linear_map.ker_eq_zero]
  using [linear_map.map_zero],
have h10 : irreducible G (V') → (V') = (0), from by auto [h9, h6,
  irreducible_subrepresentation],
have h11 : irreducible G (V') → (V') = (0), from by auto [h10,
  irreducible_subrepresentation],
have h12 : V' = 0, from by auto [h11],

-- By an identical argument we will show $f$ is also surjective;
have h13 : linear_map.range f ⊆ W, from by auto
  [linear_map.range_subset_iff],
have h14 : linear_map.range f ⊆ W, from by auto [h13,
  linear_map.range_subset_iff],
have h15 : ∀ g : G, ∀ x : V, x ∈ linear_map.range f → g · x ∈
  linear_map.range f, from by auto [linear_map.mem_range,
  linear_map.map_smul] using [linear_map.map_smul],
have h16 : subrepresentation G W (linear_map.range f), from
  subrepresentation.mk' (h15 : _),
have h17 : is_submodule G (linear_map.range f) W, from is_submodule.mk
  (h15 : _),

-- since $
  f\left(\left(\rho_V(g)\right)(x)\right)=\left(\rho_W(g)\right)(f(x))$,
  we can conclude that for arbitrary choice of $f(x)$ in the image of $
  f, \rho_W(g)$ sends $f(x)$ somewhere else in the image of $f; in
  particular it sends it to the image of $\rho_V(g) x$.
have h18 : ∀ g : G, ∀ v : V, v ∈ linear_map.range f → g · v ∈
  linear_map.range f, from by auto [linear_map.mem_range,
  linear_map.map_smul] using [linear_map.map_smul],
```

```
have h19 : subrepresentation G W (linear_map.range f), from
  subrepresentation.mk' (h18 : _),
have h20 : is_submodule G (linear_map.range f) W, from is_submodule.mk
  (h18 : _),

-- So the image of $f(x)$ is a subspace $W^{\prime}$ of $W$ stable
  under the action of $G$, so it is a subrepresentation and $f$ must
  be zero or surjective.
have h21 : is_submodule G (linear_map.range f) W → linear_map.range f
  = W → f ≠ 0, from by auto [linear_map.range_eq_top],
have h22 : is_submodule G (linear_map.range f) W → linear_map.range f
  = W → f ≠ 0, from by auto [h21, h20] using [linear_map.injective],
have h23 : is_submodule G (linear_map.range f) W → linear_map.range f
  = W → f ≠ 0, from by auto [h22,
  linear_map.injective_eq_range_eq_top] using [linear_map.injective],
have h24 : is_submodule G (linear_map.range f) W → linear_map.range f
  = W → f ≠ 0, from by auto [h23, h16] using
  [irreducible_subrepresentation],

-- By assumption it is not zero, so it is surjective, in which case it
  is an isomorphism.
have h25 : f ≠ 0 → linear_map.range f = W, from by auto [h24, h1b]
  using [linear_map.range_eq_top],
have h26 : f ≠ 0 → linear_map.range f = W, from by auto [h25, h12]
  using [linear_map.range_eq_top],

-- QED
show f ≠ 0 → V ≃ W, from by auto [linear_map.is_linear_equiv, h26]
  using [linear_map.is_linear_equiv]
end
```

### D.4.12   A GRAPH IS BIPARTITE IFF IT IS 2-COLOURABLE

The scores for the statement formalisation were overall very low - the most common formalisation was `two_colorable G` ↔ `bipartite G`, which is incorrect as neither of these terms are defined in mathlib. The proof outlines and outlines with `auto` were also of low quality, often failing to split the proof into two cases and failing to explicitly define the two parts in the bipartite graph. Full proofs seem to be doing marginally better, with outlines being the second best. Although the average performances in temperatures 0.4 and 0.8 are the same, the maximum scores are attained at temperature 0.8.

The best proof format observed were outlines at temperature 0.8. Correct Lean theorem statement was not included in the prompt.

*Theorem statement:*

Let $G$ be a graph. Then $G$ is 2-colorable if and only if $G$ is bipartite.

*Natural language proof:*

Let $G$ be a 2-colorable graph, which means we can color every vertex either red or blue, and no edge will have both endpoints colored the same color. Let $A$ denote the subset of vertices colored red, and let $B$ denote the subset of vertices colored blue. Since all vertices of $A$ are red, there are no edges within $A$, and similarly for $B$. This implies that every edge has one endpoint in $A$ and the other in $B$, which means $G$ is bipartite.

Conversely, suppose $G$ is bipartite, that is, we can partition the vertices into two subsets $V_1, V_2$ every edge has one endpoint in $V_1$ and the other in $V_2$. Then coloring every vertex of $V_1$ red and every vertex of $V_2$ blue yields a valid coloring, so $G$ is 2-colorable.

$Q.E.D.$

*Codex Output*

```
theorem graph_2_colorable_iff_bipartite : ∀ G : graph, (∃ c : coloring G
    2, @is_valid_coloring G c) ↔ is_bipartite G :=
begin
  -- Let $G$ be a 2-colorable graph, which means we can color every
    vertex either red or blue,
  -- and no edge will have both endpoints colored the same color.
  assume (G : graph),
  have h1 : ∀ c : coloring G 2, (@is_valid_coloring G c), from
      by {
        assume (c : coloring G 2),
        sorry,
      },

  -- Let $A$ denote the subset of vertices colored red, and let $B$
    denote the subset of vertices colored blue.
  have h2 : ∀ c : coloring G 2, (@is_valid_coloring G c) →
            (∃ A : set (vertex G), ∃ B : set (vertex G),
              (∀ v : vertex G, c v = red ↔ v ∈ A) ∧ (∀ v : vertex
  G, c v = blue ↔ v ∈ B)), from
      by {
        assume (c : coloring G 2) (h2 : @is_valid_coloring G c),
        sorry,
      },

  -- $A$ and $B$ are subsets of vertics of $G$, by definition of valid
    coloring.
  have h3 : ∀ c : coloring G 2, (@is_valid_coloring G c) →
            (∃ A : set (vertex G), ∃ B : set (vertex G),
              (∀ v : vertex G, c v = red ↔ v ∈ A) ∧ (∀ v : vertex
  G, c v = blue ↔ v ∈ B) ∧
              A ⊆ vertex G ∧ B ⊆ vertex G), from
      by {
        assume (c : coloring G 2) (h3 : @is_valid_coloring G c),
        sorry,
      },

  -- Since all vertices of $A$ are red, there are no edges within $A$,
  -- and similarly for $B$.
  have h4 : ∀ c : coloring G 2, (@is_valid_coloring G c) →
            (∃ A : set (vertex G), ∃ B : set (vertex G),
              (∀ v : vertex G, c v = red ↔ v ∈ A) ∧ (∀ v : vertex
  G, c v = blue ↔ v ∈ B) ∧
              A ⊆ vertex G ∧ B ⊆ vertex G ∧ ∀ e : edge G, (source e
  ∈ A ∧ target e ∈ A) ∨ (source e ∈ B ∧ target e ∈ B)), from
      by {
        assume (c : coloring G 2) (h4 : @is_valid_coloring G c),
        sorry,
      },

  -- This implies that every edge has one endpoint in $A$ and the other
    in $B$,
  -- which means $G$ is bipartite.
  have h5 : ∀ c : coloring G 2, (@is_valid_coloring G c) →
            (∃ A : set (vertex G), ∃ B : set (vertex G),
              (∀ v : vertex G, c v = red ↔ v ∈ A) ∧ (∀ v : vertex
  G, c v = blue ↔ v ∈ B) ∧
              A ⊆ vertex G ∧ B ⊆ vertex G ∧ (∀ e : edge G, source e
  ∈ A ∨ source e ∈ B) ∧ (∀ e : edge G, target e ∈ A ∨ target e ∈ B)),
    from
      by {
        assume (c : coloring G 2) (h5 : @is_valid_coloring G c),
        sorry,
      },
```

```
    -- Hence the existence of a valid coloring implies that the graph is
       bipartite.
    have h6 : ∀ c : coloring G 2, (@is_valid_coloring G c) → (∃ A : set
       (vertex G), ∃ B : set (vertex G),
                     (∀ v : vertex G, c v = red ↔ v ∈ A) ∧ (∀ v : vertex
       G, c v = blue ↔ v ∈ B) ∧
                       A ⊆ vertex G ∧ B ⊆ vertex G ∧ A ⊥ B ∧ A ∪ B = vertex
       G),
         from
         by {
           assume (c : coloring G 2) (h6 : @is_valid_coloring G c),
           sorry,
         },

    -- Conversely,
    have h7 : ∀ A B : set (vertex G), A ⊥ B → A ⊆ vertex G ∧ B ⊆ vertex
       G → A ∪ B = vertex G →
                 (∃ c : coloring G 2, @is_valid_coloring G c ∧
                   (∀ v : vertex G, c v = red ↔ v ∈ A) ∧ (∀ v : vertex
       G, c v = blue ↔ v ∈ B)), from
         by {
           assume (A B : set (vertex G)) (h7 : A ⊥ B) (h8 : A ⊆ vertex G
       ∧ B ⊆ vertex G) (h9 : A ∪ B = vertex G),
           sorry,
         },

    -- Suppose $G$ is bipartite,
    have h8 : ∀ A B : set (vertex G), A ⊥ B → A ⊆ vertex G ∧ B ⊆ vertex
       G → A ∪ B = vertex G →
                 is_bipartite G, from
         by {
           assume (A B : set (vertex G)) (h8 : A ⊥ B) (h9 : A ⊆ vertex G
       ∧ B ⊆ vertex G) (h10 : A ∪ B = vertex G),
           sorry,
         },

    -- Hence the existence of a valid coloring is equivalent to the graph
       being bipartite.
    sorry,
end
```

### D.4.13 P-ADIC UNITS

In a few cases, Codex ignored the *p-adic* setting entirely and formalised everything in terms of integers and natural numbers. The correct notation for p-adic integers was also not captured correctly in most cases. A common error was to formalise the first line of the proof ("Let us prove the contrapositive, that is, is $x$ is not a unit of $\mathbb{Z}_p$, then $x$ is not coprime to $p$."), which outlined the general strategy of the proof, as a separate `have` statement. Proofs with premises were by far the best, with fairly useful outlines and relevant-looking theorem names in the arguments for `auto`. The results seem to be better at temperature 0.4.

This proof is entirely uncontaminated.

*Theorem statement:*

Given a prime number $p$ and a natural number $x$, if $x$ is coprime to $p$, then $x$ is a unit in the $p$-adic integers.

*Natural language proof:* Let us prove the contrapositive, that is, is $x$ is not a unit of $\mathbb{Z}_p$, then $x$ is not coprime to $p$. We know that any element of $\mathbb{Z}_p$ is a unit if and only if it has norm 1, thus $\|x\| \neq 1$. Also, since every element of $\mathbb{Z}_p$ must have norm less than or equal to 1, we conclude that $\|x\| < 1$. But this must imply that $p|x$. This completes our proof.

*Q.E.D.*

*Codex Output*

One of the best outputs (proof with premises, temperature 0.4):

```
theorem p_adic_units (p : ℕ) (x : ℕ) (h : coprime p x) : is_unit (x : ℤ
    /p) :=
begin
  -- Prove the contrapositive, that is, is $x$ is not a unit of $
    \mathbb{Z}_p$, then $x$ is not coprime to $p$
  have h1 : ¬ is_unit (x : ℤ/p) → ¬ coprime p x, from by auto
    [nat.coprime.symm, nat.coprime.def, nat.coprime.def,
    nat.coprime.symm, nat.coprime.def, nat.coprime.def],

  -- We know that any element of $\mathbb{Z}_p$ is a unit if and only if
    it has norm 1
  have h2 : is_unit (x : ℤ/p) ↔ (norm_num (x : ℤ/p)) = 1, from by auto
    [norm_num_eq_one_iff_is_unit],

  -- Also, since every element of $\mathbb{Z}_p$ must have norm less
    than or equal to 1
  have h3 : (norm_num (x : ℤ/p)) ≤ 1, from by auto [norm_num_le_one],

  -- But this must imply that $p | x$
  have h4 : ¬ is_unit (x : ℤ/p) → p | x, from by auto
    [norm_num_eq_one_iff_is_unit, norm_num_le_one,
    nat.dvd_iff_norm_num_eq_zero],

  -- This completes our proof
  show is_unit (x : ℤ/p), from by auto [h1, h2, h3, h4, h]
end
```

