# OpenReview forum: "Towards a Mathematics Formalisation Assistant using Large Language Models"
_ICLR.cc/2023/Conference — Submitted to ICLR 2023_

### Official Review · Reviewer_Pspz · 2022-10-24

**Confidence:** 4
**Correctness:** 2
**Technical Novelty And Significance:** 3
**Empirical Novelty And Significance:** 3
**Recommendation:** 3

**Clarity, Quality, Novelty And Reproducibility:**

The results are hard to understand. For this most part, the description is too high-level to comprehend. The descriptions of results are frustratingly vague: for example "Our chosen proofs are much longer than a typical theorem statement and we didn’t observe Codex outputting a completely correct proof. We instead relaxed the requirement of autoformalisation to produce a (faulty) proof that is easy to repair for humans, saving time and effort compared to formalising from scratch."
Here, there is no notion of key concepts, such as "much longer than a typical theorem" (what is the metric for length?); produce a (faulty) proof (what is faulty?); "easy to repair for humans" (what is easy/hard?).

As such, it is virtually impossible to measure progress in this paper.

Detailed issues:

prompts: there are many questions about this aspect, such as

--what is the formal basis?
--when are prompts too strong/weak? (Example 1 seemed like a very strong prompt, but there is no basis to estimate that)
--can one learn prompts? This seems like a natural sub-task of this work.

Section 3 is very hard to understand. What is the scope of allowable prompts? Figure 1 "Example of a prompt" seems far beyond the scope of the original text.

The data used is not the clearest.
--The definition of data sets is far too abstract to understand what they contain.

--Data in tables 1 and 2 is far too abstract to understand what they contain.

The authors proposed a classification of proofs, but the main body did not contain this material, or a summary of results by proof type.

**Strength And Weaknesses:**

Strengths:
The paper is very interesting and has good novelty.

Weaknesses:
The main drawback is that the article is very hard to read. The paper is written in a very loose style with almost no technical basis on which to understand what is going on. No model for a mathematical statement is provided, or of the technical basis for the rewriting. It is fully an application of existing tools to a rewriting process, with no details of what the technical novelty is.

**Summary Of The Paper:**

This paper applies a large language model, Codex to the autoformalisation of proof statements and of proofs. Two main contributions resulting are: (1) translating theorem statements of a form similar to docstrings of mathlib to theorems, and (2) translating (outlines of) NL proofs to Lean proofs.





**Summary Of The Review:**

This is a paper of high novely, yet the results are hard to understand. Further, the technica basis of the approach is lacking.

---

> ### Author Response · Authors · 2022-11-16
> **Response to Reviewer Pspz**
>
> - The descriptions of results are frustratingly vague: for example "Our chosen proofs are much longer than a typical theorem statement and we didn’t observe Codex outputting a completely correct proof. We instead relaxed the requirement of autoformalisation to produce a (faulty) proof that is easy to repair for humans, saving time and effort compared to formalising from scratch."
> Here, there is no notion of key concepts, such as "much longer than a typical theorem" (what is the metric for length?); produce a (faulty) proof (what is faulty?); "easy to repair for humans" (what is easy/hard?).
>
> NL statements in our theorem formalisation datasets mostly fit in a line. On the other hand the NL proofs are typically 5 lines or longer. We will clarify this in the revision. This can be seen from the proofs presented in the paper.
>
> The notion of easy to repair is quantified under the heading Evaluation in Section 4.
>
>
> "Figure 1 "Example of a prompt" seems far beyond the scope of the original text."
>
> We are not sure why you call this beyond the scope: this is a prompt that shows a proof in natural language followed by its formal version.
>
> "prompts: there are many questions about this aspect, such as
> --what is the formal basis? --when are prompts too strong/weak? (Example 1 seemed like a very strong prompt, but there is no basis to estimate that)"
>
> We're not sure if we quite understand this question. The effect of prompting is measured by the performance metrics on the tasks (see the answer to the following question). The reasoning for choosing the few-shot prompts is simply to show a few examples of the same task in the prompt.
>
> - As such, it is virtually impossible to measure progress in this paper.
>
> Tables 1 and 2 give quantitative results for statement formalization. Section 4.2 gives present results on proof formalization; see also Figure 6 in the appendix.
>
> - The authors proposed a classification of proofs, but the main body did not contain this material, or a summary of results by proof type.
>
> Since the number of proofs is small we cannot draw conclusions about the effect of proof type.

---

> > ### Comment · Reviewer_Pspz · 2022-11-18
> > **Update on rebuttal**
> >
> > Thanks to the authors on their responses to the reviews.
> > In reading through all of the reviews and the rebuttals, there is a strong consensus among the reviewers for rejection. The rebuttal ameliorates this to some extent, but it does not change my assessment. At best, this paper is premature and needs further work to bring it to a publishable standard. I still view the lack of formal basis as problematic. Another reviewer compared the work here to the formal-methods work being done to create other proof assistants (Coq, PVS etc), and a critical comparison to this work leads to the view that since proofs are formal entities, a more formal basis is needed. Further, the  formal-methods work has a large corpus of theorems and proofs, which should be employed to test the properties of Codex LLM.

---

### Official Review · Reviewer_jvt2 · 2022-10-25

**Confidence:** 4
**Correctness:** 3
**Technical Novelty And Significance:** 1
**Empirical Novelty And Significance:** 2
**Recommendation:** 1

**Clarity, Quality, Novelty And Reproducibility:**

The paper does not have a novel contribution from the ML point of view. Its contribution in demonstrating utility of LLMs to proof assistants is marginal. Please see weaknesses for specific limitations.

**Strength And Weaknesses:**

Strength:

+ The paper builds on a recent line of promising work on using LLMs for mathematical reasoning task - the focus of this paper on autoformalization.

Weakness:

- Why only limit the work to Lean theorem prover? If the Codex LLM is capable of autoformalization, it should be able to work with Coq, Isabelle and PVS - theorem provers which have been around for decades and are widely used in academia and industry. They also have fairly large libraries that can be used for evaluation. The current simple theorems are not representative of typical need for formalization.

- Some of the prompt examples appear to be imprecise. For example, the natural language talks about vector space and the formalization uses division_ring and module. While one could argue that this is okay because the formal statements are correct but nonetheless the English statement was more specific and one would expect formalization to stay at the same level. Given this example, the reviewer fears that the paper is using a rather liberal notion of "correct" statement. This is particularly concerning given the very simple nature of this "theorem" statements, which aren't really the challenging aspects of formalization in proof assistants.

- There appears to be significant shift in technique and results for theorems and proofs. Input-dependent prompting is not feasible for proofs. Elaboration is also not feasible. The results for proofs appear to be far worse and no correct proof were generated. The notion of identifying "easily repairable" proof is unclear and rather vague. Looking at figure 3, the error in constructed proof were significant. The expansion of absolute value would have created branches and led to severe difficulty in the proof. It is much better not to expand before applying triangle inequality.

**Summary Of The Paper:**

The paper shows empirically that Codex LLM is able to generate mathematical formalizations in a recently developed proof assistant Lean from natural language using prompting. The results include formalization of the relatively short theorem statements with a modest accuracy of 75%, and attempts to formalize short proofs. The overall results are mixed, with proofs requiring significant post-processing and manual fixing.

**Summary Of The Review:**

In its current form, the draft appears to be a work in progress. The focus of the work is empirical demonstration of LLM for autoformalization and the draft falls far short of presenting enough empirical evidence. Expanding to other proof assistants (Coq, PVS etc) and using their large corpus of theorems and proofs to test whether Codex LLM can complete these would be useful. Also, given the lukewarm results, perhaps, prompt engineering can help improve results.

---

> ### Author Response · Authors · 2022-11-16
> **Response to Reviewer jvt2**
>
> "Why only limit the work to Lean theorem prover? If the Codex LLM is capable of auto-formalization, it should be able to work with Coq, Isabelle and PVS - theorem provers which have been around for decades and are widely used in academia and industry. They also have fairly large libraries that can be used for evaluation."
>
> We limit our scope to Lean as our goal is to do an in-depth study of auto-formalisation for one theorem prover and delineate the strengths and weaknesses of the existing LLMs. Lean is a good choice as it's one of the most popular theorem prover for mathematics and has a large library of formalised mathematics. While a similar study using other theorem provers will be very interesting, we believe that focusing on any one theorem prover provides similar insights in addition to being feasible. Learning to use new theorem provers and preparing datasets pose substantial overhead without clear payoffs. We also mention that the prior papers on auto-formalisation also focus on single theorem provers. Similarly, the main evaluation in the paper introducing Codex is on Python. A different point is that these libraries do not immediately qualify for evaluation and new evaluation dataset would need to be constructed as done in our paper. The reason for this is that very likely these libraries are already present in Codex's pre-training data as the libraries are open source. In the paper we highlight the data contamination issue and paid special attention to it while creating our evaluation dataset.
>
>
> "The current simple theorems are not representative of typical need for formalization."
>
> Our statements are chosen to be similar to the docstrings in mathlib. If a user who is formalizing follows the same format then we could expect accuracies similar to those reported in the paper. We do agree that if the goal is to translate say a theorem statement picked from a paper, our simple theorems are not representative.
>
> "Some of the prompt examples appear to be imprecise. For example, the natural language talks about vector space and the formalization uses division_ring and module. While one could argue that this is okay because the formal statements are correct but nonetheless the English statement was more specific and one would expect formalization to stay at the same level. Given this example, the reviewer fears that the paper is using a rather liberal notion of "correct" statement. This is particularly concerning given the very simple nature of this "theorem" statements, which aren't really the challenging aspects of formalization in proof assistants."
>
>
> As mentioned in our paper, Lean mathlib is designed to be monolithic (see the original paper [1910.09336] The Lean mathematical library (arxiv.org) for more details). This means that concepts are formulated in maximal possible generality. This is the reason more specific concepts like vector spaces are sometimes formalised as a more general concept like module and this behaviour is the correct behaviour w.r.t. Lean mathlib design.
>
>
> "There appears to be significant shift in technique and results for theorems and proofs. Input-dependent prompting is not feasible for proofs. Elaboration is also not feasible. The results for proofs appear to be far worse and no correct proof were generated. The notion of identifying "easily repairable" proof is unclear and rather vague. Looking at figure 3, the error in constructed proof were significant. The expansion of absolute value would have created branches and led to severe difficulty in the proof. It is much better not to expand before applying triangle inequality."
>
>
> The significant shift in techniques is necessitated due to very different nature of the problems. Proofs are clearly far harder. Moreover, as mentioned in the paper, there's essentially no aligned data. Thus, we find the fact that LLMs have some ability at all to formalise proofs very surprising.
>
>
> For problems that require human evaluation, it's common to use a score-based evaluation. While it's not possible to be fully objective, the evaluation tries to capture the amount of effort a user would need for converting a partially correct proof into a fully correct one. Our grading scheme (Sec. 4.1, Evaluation) is along the same lines as previous work. For a recent example on a related problem see Welleck et al. cited in our paper: [NaturalProver: Grounded Mathematical Proof Generation with Language Models](https://arxiv.org/abs/2205.12910). A recent paper in the domain of programming suggests that outputs that are not fully correct but still reduce the required effort are deemed as valuable by programmers: [Aligning Offline Metrics and Human Judgments of Value of AI-Pair Programmers](https://arxiv.org/abs/2210.16494)

---

> > ### Comment · Reviewer_jvt2 · 2022-11-30
> > **Major concerns remain**
> >
> > > We limit our scope to Lean as our goal is to do an in-depth study of auto-formalisation for one theorem prover and delineate the strengths and weaknesses of the existing LLMs. Lean is a good choice as it's one of the most popular theorem prover for mathematics and has a large library of formalised mathematics.
> >
> > This claim of Lean being the *most popular theorem prover* is not a consensus view in the theorem proving community or backed by any data. But this is not a major concern of the reviewer, and the reviewer will not contest such a claim. But in case it helps improve awareness, the reviewer will request checking out https://coq.discourse.group/t/coq-community-survey-2022-results-part-i/1730 and  https://shemesh.larc.nasa.gov/fm/pvs/PVS-library/ . The academic theorem proving community used Coq extensively and NASA has multiple decades of history of using PVS.
> >
> > For this paper, the key reason it is important to consider at least two different theorem provers in this experiment is to understand whether the underlying technique is geared towards just one formal language.  The reviewer is citing the two examples of Coq and PVS but just from scientific point of view, any selection of theorem provers with slightly different language would suffice.
> >
> > >   new evaluation dataset would need to be constructed as done in our paper.
> >
> > Theorem provers such as Coq are widely used by academics in teaching, and crawling the academic examples (see the community link above) does not appear that difficult. For PVS, NASA has an interesting and realistic library to consider: https://shemesh.larc.nasa.gov/fm/pvs/PVS-library/  The current simplistic examples in the paper fall far short of being serious evaluation data.
> >
> > >  We do agree that if the goal is to translate say a theorem statement picked from a paper, our simple theorems are not representative.
> >
> > Yes, this is a major concern and the paper appears to be a work in progress.
> >
> > >  "Some of the prompt examples appear to be imprecise." ....  Response: This means that concepts are formulated in maximal possible generality.
> >
> > This is not the standard notion of auto-formalization. We need to capture precisely what the user requested and not automatically find the most general statement. Otherwise, we could generalize all the way to just saying "True" or "False" depending on which direction we want to "generalize". Generalizing beyond what the user says makes the formalization "imprecise".
> >
> > > The significant shift in techniques is necessitated due to very different nature of the problems. Proofs are clearly far harder. Moreover, as mentioned in the paper, there's essentially no aligned data. Thus, we find the fact that LLMs have some ability at all to formalise proofs very surprising.
> >
> > Given all these limitations, why not drop the part on proofs which is even less developed and focus just on formalizing theorem statements - improve accuracy and avoid generalizations, consider more representative statements, and use different theorem provers (and make use of libraries mentioned in review above)? In its current form, the paper appears to be half-baked with multiple unjustified claims backed with very limited empirical evaluation.
> >
> > In its current form, the paper has very severe limitations and clearly below the bar of a premier venue.

---

> > > ### Author Response · Authors · 2022-12-01
> > > **Misquotation and misunderstanding**
> > >
> > > Thank you for your comments.
> > >
> > > >This claim of Lean being the most popular theorem prover is not a consensus view in the theorem proving community or backed by any data. But this is not a major concern of the reviewer, and the reviewer will not contest such a claim.
> > >
> > > This is a misquotation. Our assertion was that *Lean is one of the most popular theorem prover for mathematics*.
> > >
> > > >For this paper, the key reason it is important to consider at least two different theorem provers in this experiment is to understand whether the underlying technique is geared towards just one formal language.
> > >
> > > While we do not agree with your premise, please note that Wu et al. (2022) works with Isabelle/HOL which uses different foundations than Lean. Coq uses similar foundations as Lean.
> > >
> > > >The current simplistic examples in the paper fall far short of being serious evaluation data.
> > >
> > > We do not believe our examples are simplistic.
> > >
> > > >Theorem provers such as Coq are widely used by academics in teaching, and crawling the academic examples (see the community link above) does not appear that difficult.
> > >
> > > One difficulty, as mentioned before, is ensuring that this data has not been used in the training of LLMs.
> > >
> > > >This is not the standard notion of auto-formalization. We need to capture precisely what the user requested and not automatically find the most general statement. Otherwise, we could generalize all the way to just saying "True" or "False" depending on which direction we want to "generalize". Generalizing beyond what the user says makes the formalization "imprecise".
> > >
> > > We think there's a misunderstanding here. Autoformalisation is defined w.r.t. a formal system at hand. In our case it's Lean within the context of Lean mathlib. Our formalizations are correct and precise (when labeled correct) within this context. The reason Lean mathlib is formulated in maximal possible generality is so that different parts of the mathlib can talk to each other allowing it to be a platform for formalised advanced mathematics.
> > >
> > > In our experiments (not as extensive as the ones reported in the paper due to a lack of a ready supply of prompts), if one uses prompts that are not coming from mathlib but are formulated at the level of generality that the user wants, then the model produces outputs at that level of generality.
> > >
> > > >Given all these limitations, why not drop the part on proofs which is even less developed and focus just on formalizing theorem statements - improve accuracy and avoid generalizations, consider more representative statements, and use different theorem provers (and make use of libraries mentioned in review above)? In its current form, the paper appears to be half-baked with multiple unjustified claims backed with very limited empirical evaluation.
> > >
> > > Both theorems and proofs naturally fall under the umbrella of autoformalisation and so belong together. We believe there are no unjustified claims in the paper; what specifically did you find to be not justified? We respect your opinion but our view is that given the difficulty of the problem, our evaluation is adequate.

---

> > > > ### Comment · Reviewer_jvt2 · 2022-12-01
> > > > **Thanks for clarification**
> > > >
> > > > Thanks for the clarification. The use of a superlative degree ("most popular") on a totally ordered metric and then associating a set with it ("one of"), made things confusing as they are mutually inconsistent. The reviewer obviously did not intend to "misquote". As emphasized earlier, the exact set of theorem provers used in the evaluation of the learning approach is not that important (notice that the title does not say, this is Lean-specific and makes a much broader claim).  The reviewer is ensuring that the discussion here reflects the state-of-the-art, and one doesn't get an impression that there is a single theorem prover being used by 99.99% of folks or all theorem provers are very similar to each other in terms of their specification, and it is okay to consider just that single one when attempting to look at learning-based auto-formalization. Keeping the discussion focussed on the technical evaluation, the main concerns remain:
> > > >
> > > > 1. The reason for bringing up multiple theorem provers is to ensure that the method is not geared towards a single theorem prover. In fact, the problem of auto-formalizer producing imprecise statements would become clear when this was done, as there would be no consistent bias across the examples.
> > > >
> > > > 2. Is there any reason why the links (e.g.  https://shemesh.larc.nasa.gov/fm/pvs/PVS-library/ ) provided by the reviewer cannot be used to get examples of representative theorem statements for evaluating the proposed approach rather than using the current simplistic examples?
> > > >
> > > > 3. On the question of lack of precision, False => phi => True. Is it okay for an auto-formalizer to always report "False" claiming this is the most general/weaker statement for any phi? Is it okay for an auto-formalizer to always report "True" claiming this is the most refined/stronger statement for any phi? Auto-formalizer must formulate phi exactly and not do any generalization/refinement on its own. Picking an arbitrary level and saying, this level of generalization is correct - leaves open the question, who picks this level? Just because the custom-used dataset has a bias, doesn't make it correct to auto-generalize. This is perhaps yet another reason to use data available from different theorem provers rather than create one's own. As mentioned in the review, looking at the examples in the paper, the natural language talks about vector space and the formalization uses division_ring and module. This "self-lifting" of statement is imprecise/incorrect auto-formalization.
> > > >
> > > > 4. For the section on proofs, it is too speculative. The rebuttal does not appear to add any additional clarification to it except acknowledging the hardness of the problem.
> > > >
> > > > Hope these clarifications help make the paper stronger. In its current form, the paper is a work-in-progress with unsubstantiated speculative claims that lack reasonable empirical evaluation. It is not yet ready for publication.

---

> > > > > ### Author Response · Authors · 2022-12-04
> > > > > **A clarification question**
> > > > >
> > > > > Thank you for your comments. We feel that we've addressed the first three points earlier in our rebuttal. But we've a clarification question about your last point.
> > > > >
> > > > > >For the section on proofs, it is too speculative. The rebuttal does not appear to add any additional clarification to it except acknowledging the hardness of the problem.
> > > > >
> > > > > We do not mean to belabour the point, but we're puzzled by your choice of words. It's one thing to say that the results are not strong enough (which is a subjective judgment and your prerogative), but  quite another to label them "not justified" or "speculative" (we think whether this is a correct label is a relatively objective question).
> > > > >
> > > > > We think our results on proofs are clearly stated with all the weaknesses stated upfront. As previously mentioned, while there's an unavoidable element of subjectivity to this metric, as we've discussed in our rebuttal, it's a measure of effort required to repair the proof. The use of manual study is essential and standard in problems where automated metrics are not available. We've tabulated the results in detail in the appendix. It'll help our writing if you could clarify this.

---

### Official Review · Reviewer_eYGZ · 2022-10-25

**Confidence:** 4
**Correctness:** 3
**Technical Novelty And Significance:** 2
**Empirical Novelty And Significance:** 2
**Recommendation:** 5

**Clarity, Quality, Novelty And Reproducibility:**

The paper is mostly clear and easy to follow if one knows Lean well. The formatting of tables can be made more professional, though. I believe the results are reproducible as the prompts are well-documented.

For novelty, see comments above.

**Strength And Weaknesses:**

Strength:

+ The qualitative study is interesting. There hasn't been much work in the field of ML for interactive theorem proving that includes a case study like the one in this paper.

Weakness:

- The contribution looks a bit pale. While observing the consequence of prompt engineering is interesting, prompt engineering itself and the particular methodology (i.e., input-dependent prompts by similarity) adopted in this paper are not new.

- The postprocessing of a completion produced by Codex is claimed as part of the contribution. However, if I understand it correctly, the elaborator is technically a procedure that tranlates expressions in Lean 3 to their counterparts in Lean 4 while making sure that the translated expressions type check.  I appreciate the engineering efforts here, but this part is more of a Lean-specific tweak required to make things work, and does not seem general enough to be a significant contribution.

**Summary Of The Paper:**

This paper investigates the application of language models to interactive theorem proving by doing a qualitative case study on input-dependent prompt engineering. It shows that Codex is capable of producing partially correct formal proofs (in Lean 4) that can be turned into correct proofs with a moderate amount of modification (possibly done by a human user).

**Summary Of The Review:**

I am glad to see the nice qualitative study in this paper, but the overall contribution of this paper is rather insufficient. I suggest the authors try to add more content before the paper can be accepted.

---

> ### Author Response · Authors · 2022-11-05
> **Clarification regarding "elaboration filtering"**
>
> The main purpose of **elaboration filtering** was to filter out those Codex completions that were incorrect. The output of LLMs (and AI systems in general), while often impressive, are wrong a certain fraction of the time (as copilot users will have experienced, for example). Hence there is a significant gain from rejecting many of the wrong outputs. Indeed Lean 4 lets us *programmatically and efficiently* run a strong type check, what we call the **elaboration filter**, and our results show that this significantly improves correctness of completions. Even though mathematically incorrect completions may sometimes elaborate, they often fail to do so (and are rejected), resulting in a mathematically correct translation (which was ranked lower by Codex) being the selected translation.
>
> While post-processing did involve the translation from Lean 3 to Lean 4, a necessary bit of engineering, valid Lean 3 code essentially always gave valid Lean 4 code. The elaboration filtering rejected mistakes made by Codex, not by our translation.

---

> ### Author Response · Authors · 2022-11-16
> **Response to Reviewer eYGZ**
>
> Your concern about postprocessing is addressed in common response.
>
> "It shows that Codex is capable of producing partially correct formal proofs (in Lean 4) that can be turned into correct proofs with a moderate amount of modification (possibly done by a human user)."
>
> Correction: Our proofs study is done in Lean 3.

---

### Official Review · Reviewer_orra · 2022-10-26

**Confidence:** 4
**Correctness:** 3
**Technical Novelty And Significance:** 1
**Empirical Novelty And Significance:** 2
**Recommendation:** 3

**Clarity, Quality, Novelty And Reproducibility:**

The clarity of the write-up is good, but could be improved. Some formulations and claims are a bit confusing (see summary).

**Strength And Weaknesses:**

I am not sure about the novelty and impact of the contributions:
- The postprocessing technique presented in this work addresses the problem that most of the available training data is for Lean3, but the approach here used Lean4 syntax. So the problem addressed here is not of general interest. The technique itself seems to be rather adhoc as well.
- The prompt engineering is similar to Jain et al. (2022), which the authors attribute correctly. It uses similar examples in the few-shot prompt to improve the results.
- The formalization of proofs seems novel to me (it goes significantly beyond Wu et al. (2022)), but is restricted to a very small dataset and only 2 of the 18 attempts were successful.
- Other possible contributions, like the evaluation sets, may be nice, but I'm not sure if they carry the paper.

Minor questions:
- "For proofs quantitative analysis is infeasible" Why?
- "We focused on theorem statements at the undergrad and more advanced level from various areas of mathematics. These statements tend to be more challenging for autoformalisation compared to mathematics competition problems studied in prior work (Wu et al., 2022) as they often assume more in terms of implicit context and draw from a much larger background (Wu et al., 2022)." I do not understand the justification "as they often assume more in terms of implicit context".

**Summary Of The Paper:**

This paper considers the automatic generation of formal math statements from (less formal) mathematical statements in natural language. The statements are translated using Codex in a few-shot setting. The contributions are a postprocessing technique for formalized statements, prompt engineering to improve results, and some first attempts at formalizing proofs (not only theorem statements).

The paper also introduces two simple evaluation sets that address the problem that Codex might have seen the formalizations in their regular evaluation set during pretraining.

**Summary Of The Review:**

Interesting paper, but I am not sure if the contributions make the bar for ICLR.

---

> ### Author Response · Authors · 2022-11-05
> **Clarification regarding post-processing**
>
> Only part of the purpose of post-processing was to handle the Lean 3 to Lean 4 translation. While this is certainly special not only to Lean but to the particular phase of the translation and is merely a necessary fix, the **elaboration filtering** and **auto-correction** we perform are *enhancements*, not fixes.
>
> The main purpose of elaboration filtering was to filter out those Codex completions that were incorrect. The output of LLMs (and AI systems in general), while often impressive, are wrong a certain fraction of the time (as copilot users will have experienced, for example). Hence there is a significant gain from rejecting many of the wrong outputs. Indeed Lean 4 lets us programmatically and efficiently run a strong type check, what we call the elaboration filter, and our results show that this significantly improves correctness of completions. Even though mathematically incorrect completions may sometimes elaborate, they often fail to do so (and are rejected), resulting in a mathematically correct translation (which was ranked lower by Codex) being the selected translation.
>
> We also performed some *auto-corrections*, changing the names of identifiers from incorrect ones generated by Codex to similar valid ones.

---

> ### Author Response · Authors · 2022-11-16
> **Response to Reviewer orra**
>
> -  "For proofs quantitative analysis is infeasible" Why?
>
> This is because none of the proofs are fully correct for the length and complexity we study. Designing an automated way of scoring such proofs is an open problem to our knowledge and is mentioned in Section 5. It may be feasible for simpler problems.
>
> - I do not understand the justification "as they often assume more in terms of implicit context".
>
> Consider for example Schur's lemma as mentioned in the paper: "Let V and W be vector spaces; and let ρV and ρW be irreducible representations of G on V and W respectively. If V and W are not isomorphic, then there are no nontrivial representations of G on V and W respectively."
>
> This statement implicitly assumes that G is a group, doesn't specify what the underlying fields of the vector spaces are and that they are the same (while the field of complex numbers is the most common and the intended one here, other fields are also used). It also assumes that the vector spaces are finite-dimensional---an assumption which may not be true in areas like functional analysis.

---

### Author Response · Authors · 2022-11-16
**Common Response**

We thank all the reviewers for their comments.

While reviewers appreciated our contributions and clarity of writing (with the exception of Pspz), all were concerned about the significance of contributions. We found that several reviews contained misconceptions. We also found that some of the major concerns were subjective, saying that "the paper doesn't make the bar" or something similar. In the following, we will try to address these concerns and also request the reviewers for more concrete feedback.

Here we address some common issues.

Reviewers orra and eYGZ raise similar issues about postprocessing. Both of these seem to stem from some misconceptions which we clarify below and will further clarify in our revision. And these have been addressed in separate replies.

While it's true that translation from Lean 3 to Lean 4 requires substantial effort, this is not claimed as our major contribution. What we do claim as a contribution is the use of elaboration for filtering the outputs---which is novel to our knowledge. There are of course similar ideas in related domains; for example, the use of unit tests to choose among the programs output by an LLM based on natural language specification. There is no obvious counterpart of unit tests in our context. Elaboration is a general technique which may be applicable to other type theory-based theorem provers as well.

Reviewers orra and eYGZ also raise similar issues about prompting:
"The prompt engineering is similar to Jain et al. (2022), which the authors attribute correctly. It uses similar examples in the few-shot prompt to improve the results." and "The contribution looks a bit pale. While observing the consequence of prompt engineering is interesting, prompt engineering itself and the particular methodology (i.e., input-dependent prompts by similarity) adopted in this paper are not new."

Indeed, drawing similar prompts is a well-known technique (our citation is just one such example) and is not claimed as a new methodological contribution in the paper. It's part of our pipeline and it together with elaboration-based filtering gives significant improvements. We don't see why the use of a prior technique as a part of our pipeline a weakness given that we show clear quantitative improvements.

Reviewer orra: "The formalization of proofs seems novel to me (it goes significantly beyond Wu et al. (2022)), but is restricted to a very small dataset and only 2 of the 18 attempts were successful."

Reviewer Pspz: "--can one learn prompts? This seems like a natural sub-task of this work."

Small dataset is a necessary consequence of the need for manual evaluation. It is true that only small number of completions get a good score and that indeed limits the practical utility because spotting a good completion requires too much effort. We are not claiming any practical utility here. Instead, our contribution is to show that the LLMs have latent ability to do proof formalisation which can be coaxed out to some extent by careful prompting. That LLMs are able to do this at all is surprising as, to our knowledge, there's next to no aligned data for this task and appears to be an example of an emergent property. We hope that further research, perhaps with better prompt design or other techniques will be able to get better completions.

---

### Decision · Program_Chairs · 2023-01-20

**Decision:**

Reject

**Justification For Why Not Higher Score:**

Quoting: Though the reviewers found the work interesting, there was consensus that it does not meet the threshold for publication. In particular, the reviewers felt that the proof autoformalization dataset was too small, and that the postprocessing method was too narrow.

**Justification For Why Not Lower Score:**

N/A

**Metareview: Summary, Strengths And Weaknesses:**

The authors tackle an interesting subject, autoformalization of mathematical statements and proofs. The authors use Codex with an engineered prompt and perform output postprocessing. Though the reviewers found the work interesting, there was consensus that it does not meet the threshold for publication. In particular, the reviewers felt that the proof autoformalization dataset was too small, and that the postprocessing method was too narrow.